# How to Evaluate Reward Models for RLHF

**Evan Frick    Tianle Li    Connor Chen    Wei-Lin Chiang    Anastasios N. Angelopoulos
Jiantao Jiao    Banghua Zhu    Joseph E. González    Ion Stoica**

UC Berkeley

## Abstract

We introduce a new benchmark for reward models that quantifies their ability to produce strong language models through RLHF (Reinforcement Learning from Human Feedback). The gold-standard approach is to run a full RLHF training pipeline and directly probe downstream LLM performance. However, this process is prohibitively expensive. To address this, we build a predictive model of downstream LLM performance by evaluating the reward model on proxy tasks. These proxy tasks consist of a large-scale human preference and a verifiable correctness preference dataset, in which we measure 12 metrics across 12 domains. To investigate which reward model metrics are most correlated to gold-standard RLHF outcomes, we launch an end-to-end RLHF experiment on a large-scale crowd-sourced human preference platform to view real reward model downstream performance as ground truth. Ultimately, we compile our data and findings into Preference Proxy Evaluations (PPE), the first reward model benchmark explicitly linked to post-RLHF real-world human preference performance, which we open-source for public use and further development at github.com/lmarena/PPE.

## 1 Introduction

The ultimate test of a reward model is as follows:

> Does the reward model lead to good post-RLHF language model performance?

In other words, because the reward model will be used as a reference signal for LLM training, in principle, only the downstream LLM performance matters. However, to evaluate downstream performance, we must train a new LLM using the reward model and evaluate the resulting LLM—a prohibitively expensive and time-consuming process (Figure 1). The long development-feedback cycle of reward models poses a significant challenge, limiting achievable reward model quality and, consequently, limiting the effectiveness of the entire RLHF process.

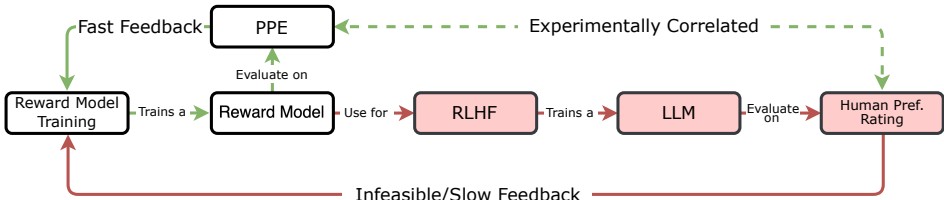

Figure 1: Overview of the RLHF pipeline. Reward models feed into the very beginning of the RLHF pipeline, making iterative improvements prohibitively slow. PPE enables a fast feedback loop that is correlated to downstream outcomes.

This paper introduces a cost-effective method for approximating the effect of a reward model on downstream LLM performance. Specifically, we measure reward model performance using a large-scale, crowdsourced pairwise human preference evaluation dataset, as well as a high-quality, programmatically verifiable correctness preference dataset. To avoid introducing bias, we do not utilize LLM judges or expert annotators to provide ground-truth references. Instead, we focus on real-world preference data that reflects organic LLM usage. Additionally, we aim our evaluation tasks

to closely resemble real-world RLHF training, making the assessment more aligned with practical use cases. Moreover, to bridge the existing knowledge gap between reward model evaluations and actual post-RLHF outcomes, we experimentally correlate our evaluation metrics with real human preferences on RLHF-ed LLMs. To achieve this, we used select reward models within a full RLHF training pipeline, each producing a fine-tuned LLM. These RLHF-tuned models are then deployed on a crowd-sourced human preference platform where we directly measure their downstream human preference scores. Through this end-to-end analysis, we identify which metrics across diverse domains show the strongest correlation with real-world post-RLHF performance. By validating this correlation, we ensure that iterative improvements on our evaluation will lead to tangible gains in downstream performance.

Additionally, we release PPE, a crowdsourced collection of 16,038 labeled human preference pairs containing responses from 20 different top LLMs and over 121 languages as well as a dataset of 2,555 prompts, each with 32 different sampled response options, totaling 81,760 responses across 4 different models, all grounded with verifiable correctness labels. PPE evaluates reward models on 12 different metrics and 12 different domains, such as their accuracy in selecting human-preferred or verifiably correct responses. Notably, PPE is the *only* reward model benchmark directly linked to downstream RLHF outcomes.

To summarize, our work makes the following contributions:

1. We analyze how reward model metrics correlate with real downstream human preference performance post-RLHF.

2. We fully open-source PPE, a comprehensive benchmark for reward models with metrics directly linked to downstream RLHF outcomes.

## 2 RELATED WORK

### 2.1 HUMAN PREFERENCE AND REWARD MODELS

Human preference has emerged as one of the gold standards for LLM training and evaluation. Several large-scale human preference datasets have been developed, including Stanford Human Preference (SHP) (Ethayarajh et al., 2022), Chatbot Arena (Chiang et al., 2024), and Anthropic HH (Bai et al., 2022a), among others. Researchers requiring human preference proxies have pursued two main approaches in this area. First, they have trained reward models based on real or synthetically generated human preference data to approximate human preferences for LLM training. Second, they have employed LLMs as judges for evaluating other LLMs.

For the training side, the line of work on Reinforcement Learning from Human Feedback (RLHF) focuses on the family of algorithms that first train a reward model as a proxy of human preferences, and then use the reward model as the signal to fine-tune the language model with reinforcement learning (Christiano et al., 2023; Bai et al., 2022a; Ouyang et al., 2022; Touvron et al., 2023; OpenAI, 2022; Bai et al., 2022b; Lee et al., 2023; OpenAI, 2023a;b; Zhu et al., 2024).

This paper studies one of the critical problems in the RLHF process: how do we evaluate reward models and select the best one for downstream performance?

### 2.2 REWARD MODEL BENCHMARKS

RewardBench is the first and only previous RLHF reward model benchmark (Lambert et al., 2024). RewardBench has 4 main tasks: Chat, Chat Hard, Safety, and Reasoning. The authors source considerable ground truth preference pairs from MT-Bench (Zheng et al., 2023) and AlpacaEval (Dubois et al., 2023), though preference labels are also hand-verified. RewardBench also uses adversarial examples from LLMBar (Zeng et al., 2024), coding example pairs with correct vs buggy implementations, and safety pairs with should-refusals and should-not-refusals. Overall, RewardBench is designed to evaluate across an array of tasks posited as relevant to RLHF. RewardBench takes a crucial first step toward reward model evaluations. However, the authors assert that more research must be done to understand how to correlate performance to RLHF success. In this paper, our experiments show that as reward models have improved, we now see a negative correlation between

| | Diverse Human Pref. | # Prompts | # Responses | Verified RLHF Outcomes |
|---|---|---|---|---|
| RewardBench[1] | No | 2,985 | 5,970 | No |
| PPE (Ours) | **Yes** | **18,593** | **113,836** | **Yes** |

Table 1: Comparison of PPE to existing work.

RewardBench evaluation score on top models and downstream RLHF performance. We aim to improve upon this gap with the our findings.

## 3 SOURCING GROUND TRUTH PREFERENCE LABELS

Previous work on sourcing preference ground truth labels often relies upon LLM judge preference labels in conjunction with manual verification from individuals, introducing potential preference biases. Alternatively, rejected responses are often curated synthetically by unnaturally perturbing the chosen output or modifying the prompt to produce forced errors, introducing bias on how errors look and occur. These preference pairs are not representative of the distribution of responses seen by reward models when providing learning signals for RLHF. We offer a brief comparison to previous work in Table 1.

Thus, we ground our preference labels with the following methodology: (1) Utilize crowdsourced diverse prompts and responses with human preference labels. (2) Utilize existing benchmarks with verifiable correctness checks on LLM-generated responses.

The methodology (1) provides an unbiased estimate of real-world human preference through the aggregation of many diverse human preferences. We use a large crowdsourced preference set of 16,038 preference labels to mitigate individual label noise and avoid over-fitting to any single individual's preference, details in subsection 4.1.

Methodology (2) curates an objective correctness signal naturally unbiased by response style. We use the second approach to label the correctness of many sampled responses from an LLM, mimicking rollouts or best-of-k exploration strategies seen in RLHF training processes. As a result, we draw preference pairs from more naturally occurring distributions (eg. real LLM responses and errors), better align with the expected environment reward models operate in.

For an overview of the curated benchmark datasets in PPE based on these two methodologies, please see Appendix A.1.

## 4 HUMAN PREFERENCE METRICS

To benchmark whether a reward model aligns with human preference directly, we utilize a human preference dataset collected from a large-scale human preference annotation platform that allows users to vote on pairwise comparisons between responses generated from two anonymized and randomly selected LLMs. Our human preference dataset contains human-labeled preferences for 16,038 pairwise comparisons between 20 selected top models[2]. These models were selected based on their strong performance on Chatbot Arena and overall popularity (Chiang et al., 2024). We emphasized selecting models that have already undergone some form of RLHF, anticipating that these models would be more challenging for reward models to evaluate.

Since the human preference set is crowd-sourced, we can repeat the collection process at any time to obtain an updated set that better reflects the current array of available models and any changes in human preference. Additionally, a newly updated human preference set would largely mitigate benchmark leakage that may have occurred with the previous set. Consequently, this human preference metric can remain consistently up-to-date with fresh, relevant data.

---

[1]RewardBench is currently the only other evaluation scheme for RLHF reward models (Lambert et al., 2024).

[2]mistral-large-2402, phi-3-medium-4k-instruct, gpt-4-1106-preview, claude-3-opus-20240229, gemini-1.5-pro-api-0514, gpt-4-0314, claude-3-haiku-20240307, gpt-4-0613, claude-3-sonnet-20240229, yi-1.5-34b-chat, llama-3-8b-instruct, gemini-1.5-flash-api-0514, llama-3-70b-instruct, gpt-4o-2024-05-13, command-r-plus, gpt-4-turbo-2024-04-09, qwen2-72b-instruct, command-r, qwen1.5-72b-chat, starling-lm-7b-beta

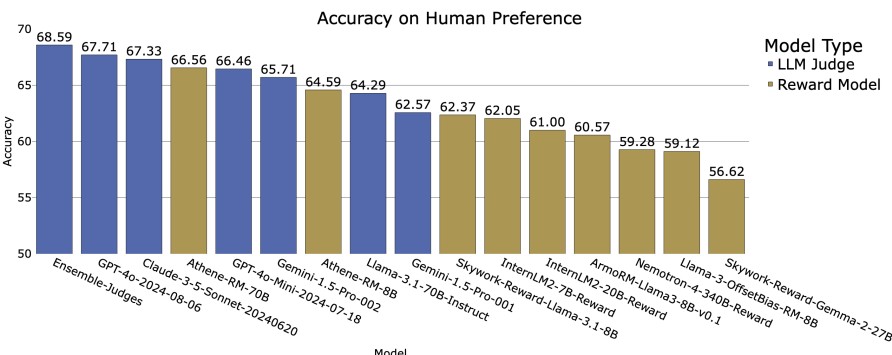

Figure 2: Model accuracies on predicting human preference labels on PPE's human preference benchmark dataset. Accuracies are measure on the "Overall" category.

## 4.1 CURATION

Specifically, we curate our human preference data from crowd-sourced battles. A "battle" consists of a user-provided prompt, two models and their responses to the prompt, and the user's preference vote for the responses. We perform a random sample weighted by model occurrence to obtain 50,000 collected battles between selected models such that models are represented at a uniform frequency, then de-duplicate and remove any samples containing P.I.I information using Azure AI. We use OpenAI's moderation API to flag and remove potentially harmful conversations from the sample. Finally, we subsample 16,038 pairs from the remaining battles to construct the human preference benchmark dataset.

The human preference benchmark dataset, at a glance: (1) Includes 4,583 instruction-following prompts, 5,195 hard prompts, 2,564 math prompts. Prompts may exist in multiple categories. (2) Includes user queries from over 121 languages. Top languages include English (8,842), Chinese (1,823), Russian (1,779), German (568), Korean (338), Japanese (321), etc. (3) Includes preferences crowdsourced from 6,120 individuals.

## 4.2 SCORING

We conduct several statistical metrics described below to evaluate different aspects of a given reward model.

1. **Accuracy.** We compute pairwise ranking accuracy against a human preference label for each reward model, excluding battles in which the human rater selected a "tie". This measures the granular case-by-case similarity to a real human preference signal. Figure 2 visualizes accuracy scores on the overall human preferences.

2. **Correlation.** Since each battle contains information on model identities, each reward model produces a ranking and a pairwise win-rate matrix for the 20 selected models. We compute Spearman and Kendall correlation between model ranking produced by each reward model against ground truth ranking. In addition, we compute row-wise Pearson Correlation between the win-rate matrix produced by each reward model against the ground truth win-rate matrix. We intuit that these aggregate correlation metrics measure overall similarity to real human preference.

3. **Confidence.** To weight stability in assigning preferences, we follow the metrics proposed in Arena-Hard-Auto (Li et al., 2024b), where we measure each reward models's Separability with Confidence Interval, Confidence Agreement, and Brier Score against ground truth ranking. These metrics are designed to measure uncertainties and over-confidence within a reward model.

Furthermore, we can calculate all the above scores conditioned on any subset of prompts in the evaluation data, specifically capturing 7 different domains. For example, we can observe these metrics on only math prompts or only instruction following prompts. We expect that strong reward models should score high regardless of the selected domain. Scores for all subsets are detailed in Appendix A.2. Score distribution statistics for each metric are detailed in Appendix A.2.1.

## 5 CORRECTNESS METRICS

To measure a reward model's ability to distinguish between different samples drawn from the same distribution, we utilize correctness metrics on established, reputable benchmarks with verifiable ground truths (e.g. Austin et al. (2021)'s MBPP-Plus). We construct a benchmark dataset wherein each prompt is associated with 32 different responses sampled from the same LLM. Additionally, since we use benchmarks with verifiable ground truths, we can score the correctness (a binary label) of each response according to the original static benchmark's verification function (e.g. code unit tests or Regex matching).

To assess the performance of reward models (and LLMs-as-judges), we obtain rewards/preferences for the sampled responses and evaluate how well these align with the verifiable correctness signal, with the general assumption that expert humans would always prefer correct answers over incorrect ones. Our response sampling strategy ensures that the preference labeler must disentangle the correctness signal from potentially very similar or even adversarial outputs, thereby increasing task difficulty. Moreover, this method naturally samples "unforced" errors as they would appear in real training or evaluation schemes, rather than synthetically constructing preference pairs that may contain underlying confounding biases.

### 5.1 CURATION

For the correctness metrics, we selected standard, widely used, reputable, and verifiable benchmarks: MMLU Pro (Wang et al., 2024b), MATH (Hendrycks et al., 2021), GPQA (Rein et al., 2023), MBPP Plus (Austin et al., 2021), and IFEval (Zhou et al., 2023). Each benchmark covers a different domain: general knowledge, mathematics, STEM, coding, and instruction following, respectively. While we initially curate PPE with these five benchmarks, it should be noted that any desired verifiable benchmark can be added to the correctness measurement paradigm by repeating the process outlined below, thereby providing a framework for customization towards specific evaluation needs.

For each benchmark, we sample LLM responses for 500 randomly selected prompts, each 32 times, for a total of 16,000 completions. If a benchmark has fewer than 500 prompts, we use all available prompts. We choose a large K of 32 to allow models to generate more diverse responses, covering a larger input domain for the human preference proxy and testing greater robustness to over-optimization. We note that this sampling strategy actually yields very similar KL-Divergence shifts as would be seen in RLHF training methods such as Proximal Policy Optimization (PPO) (Gao et al., 2022; Schulman et al., 2017).

We repeat this process for four different models: Llama-3-8B-Instruct, Gemma-2-9b-it, Claude-3-Haiku, and GPT-4o-mini-2024-07-18 (AI@Meta, 2024; Team et al., 2024; Anthropic, 2024; OpenAI, 2024). Each model samples prompts randomly with different seeds. We reason that different model response distributions may have different difficulties. For example, an already extremely high-performing model like GPT-4o-mini-2024-07-18 may be more challenging for reward models to evaluate correctness.

We then score all responses using the benchmark's verification methods. Using the correctness labels for all responses, we discard any rows in which the model got every single response wrong or every single response right, as it is impossible for the reward model to select a better generation in these cases. Additionally, we discard any row where less than 10% or greater than 90% of the responses were correct, with exceptions made for benchmarks with very few valid options. This step helps avoid vacuously correct responses, such as an LLM randomly guessing the correct multiple-choice answer with completely nonsensical reasoning, as well as prompts that are too easy.

From the remaining data, we randomly sample 128 responses from each model, totaling 512 samples. If a benchmark is too small and some models have fewer than 128 viable samples, we adjust the sampling accordingly. More details on curation can be found in Appendix A.3.1.

### 5.2 SCORING

We score the reward models on the correctness metrics in ways that target a reward model's robustness, granularity, and theoretical roof-line performance. Details on reward model and llm-judge scores can be found in Appendix A.3.3. Score distribution statistics can be found in Appendix A.3.4.

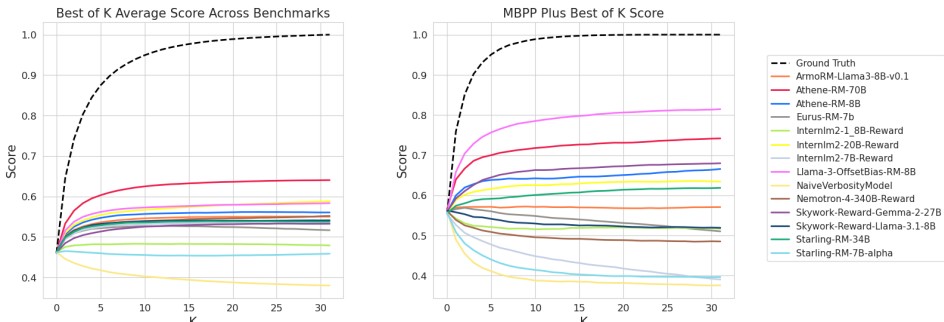

Figure 3: Best of K curves showing reward model score vs K. The blacked dashed line is the theoretical optimal curve, closer to this curve implies a better score. The left graph shows each reward model's curve averaged across all correctness PPE benchmarks. The right graph shows each reward model's curve on just the MBPP-Plus set where over-optimization behavior is seen in some reward models, characterized by curves that decrease with respect to increases in K.

| Reward Model | MMLU-Pro | MATH | GPQA | MBPP-Plus | IFEval | Mean |
|---|---|---|---|---|---|---|
| Athene-RM-70B | 0.77 | 0.79 | 0.59 | 0.68 | 0.62 | **0.69** |
| Claude 3.5 (ArenaHard)[†] | **0.81** | **0.86** | **0.63** | 0.54 | 0.58 | 0.68 |
| Llama-3-OffsetBias-RM-8B | 0.62 | 0.68 | 0.55 | **0.74** | 0.62 | 0.64 |
| GPT-4o-mini (ArenaHard)[†] | 0.71 | 0.81 | 0.57 | 0.54 | 0.56 | 0.63 |
| Llama-3.1-70B (ArenaHard)[†] | 0.73 | 0.73 | 0.56 | 0.58 | 0.56 | 0.63 |
| internLM2-20B-Reward | 0.68 | 0.70 | 0.57 | 0.58 | 0.62 | 0.63 |
| Athene-RM-8B | 0.68 | 0.71 | 0.55 | 0.62 | 0.57 | 0.62 |
| ArmoRM-Llama3-8B-v0.1 | 0.66 | 0.71 | 0.57 | 0.54 | 0.58 | 0.61 |
| Skywork-Reward-Llama-3.1-8B | 0.64 | 0.70 | 0.57 | 0.52 | 0.61 | 0.61 |
| Nemotron-4-340B-Reward | 0.70 | 0.65 | 0.57 | 0.49 | 0.63 | 0.61 |
| internLM2-7B-Reward | 0.67 | 0.73 | 0.55 | 0.44 | **0.64** | 0.60 |
| Llama-3.1-70B (Alpaca)[†] | 0.66 | 0.66 | 0.56 | 0.52 | 0.56 | 0.59 |
| Claude 3.5 (Alpaca)[†] | 0.66 | 0.63 | 0.56 | 0.52 | 0.57 | 0.59 |
| Skywork-Reward-Gemma-2-27B | 0.54 | 0.63 | 0.53 | 0.59 | 0.54 | 0.56 |
| GPT-4o-mini (Alpaca)[†] | 0.57 | 0.64 | 0.53 | 0.52 | 0.56 | 0.56 |
| NaiveVerbosityModel | 0.48 | 0.50 | 0.48 | 0.31 | 0.52 | 0.46 |

Table 2: Reward model and LLM-as-a-judge scores on the correctness accuracy metric. LLM-as-a-judge is marked with †.

### 5.2.1 BEST OF K CURVES

A best of K curve shows on average how the reward model's selected "best" answer's ground truth score changes vs K. When plotted against the ground truth curve, we can observe the gap between the reward model's ability to select the "best" answer given a set of K responses, and the "gold standard" best score. More formally, let $S_K$ be a size $K$ random sample of responses from a model, $g : S_K \to \{0, 1\}$ be the ground truth scoring function, and $\hat{R} : S_K \to \mathbb{R}$ be the reward model proxy score. Then, $\mathbb{E}_{S_K}[g(\arg\max_{s \in S_K} \hat{R}(s))]$ is the expected ground truth score of the select response by the reward model given K sampled responses. We then sweep across K = 1,..., 32 to obtain a curve. More details on these curves and derived metrics can be found in Appendix A.3.2. Best of K scores for various reward models are detailed in Appendix Table 30.

### 5.2.2 AREA UNDER RECEIVER OPERATOR CHARACTERISTICS (ROC) CURVE

Since the ground truth verification outputs a binary label, we can check each reward model's strength as a binary correctness classifier by calculating the area under the ROC curve. We first normalize the scores in each row with min-max normalization. Then we calculate the binary classification ROC curve using the normalized scores as "probabilities". AUC scores are detailed in Appendix Table 31.

### 5.2.3 Accuracy

Since LLM-as-a-judge cannot easily scale 32-wise judgments, we create a supplemental pairwise task to evaluate correctness preference accuracy compatible with both reward models and LLM-as-a-judge. For each row of best of K data, we simply sample 5 pairs of responses such that in each pair, there is one correct response and one incorrect response. Then, after randomizing positions, the LLM-as-a-judge picks the preferred response. We then measure the accuracy as the rate in which the correct response is preferred over the incorrect result. The accuracies for reward models are also collected for comparison. All scores are documented in Appendix Table 2.

## 6 Validating PPE on Post-RLHF Outcomes

By testing a reward model performance on a benchmark, we hope to glean insight towards downstream performance on an LLM RLHF-ed using a given reward model. To measure how well different metrics in PPE correlate to post-RLHF LLM performance on real-world human preference, we conduct an experiment in which we RLHF a given base LLM using different reward models. We then measure the real-world human preference scores of the resulting LLMs to understand the true performance of the original reward models.

For our experimental setup, we use each reward model to individually RLHF Llama-3.1-8B-Instruct through Direct Preference Optimization (DPO) (Rafailov et al., 2023). This way, we can compare LLMs tuned on identical RLHF pipelines, except for the reward model being measured. Then, these RLHF-ed LLMs are deployed to to a crowd-sourced annotation platform to collect real-world human pairwise preferences between model answers. Overall, 12,190 human votes were collected and compiled into relative rankings between these RLHF-ed LLMs. Under this controlled RLHF experiment, the non-noise variance in final human preference rankings attained by these models is dependent only on the reward model choice, effectively measuring the downstream performance of these reward models, albeit on a single model base model undergoing off-policy DPO RL training.

### 6.1 Training Procedure

Nine[3] reward models were selected to act as preference labels in a full RLHF training pipeline in which the resulting models were evaluated on real human preference. We constrained this experiment to nine models for cost reasons– the RLHF and human preference evaluation process is exceedingly expensive. We selected popular, newer, and high-performing reward models from RewardBench. We reason these will be the most difficult reward models to differentiate. We also require the selected reward models to be general-purpose reward models, and not specifically tuned to any single domain or task.

We create a training dataset by first including 7,000 prompts sampled from the original 50,000 human preference votes after PII removal, unsafe prompt removal, and de-duplication. We then add 500 random prompts from MMLU-Pro that are not in PPE, and another 500 prompts from MATH train set (also mutually exclusive from PPE). For each prompt, we sample 16 responses from the base model, Llama-3.1-8B-Instruct, randomizing the temperature for each generation, drawing from a triangular distribution ($a = 0.0, b = 1.0, c = 1.3$) to promote more diverse exploration. This process yields 8,000 total prompts, each with 16 different responses, totaling 128,000 responses.

Each reward model then constructs its own preference dataset. First, the reward model gives scores for each of the 16 responses for each prompt. The "chosen" response is set as the maximum scoring response. The "rejected" response is sampled as the rank $n$ response, where $n$ is sampled uniformly. Note that the sample for $n$ is seeded such that it is the same for each across reward models. This process yields a dataset of 8,000 rows, each with a prompt, a chosen response, and a rejected response where both responses are in-distribution for the base model– a requirement for using DPO.

---

[3]Selected: Athene-RM-70B and Athene-RM-8B, InternLM2-20B-Reward, InternLM2-7B-Reward, Llama-3-OffsetBias-RM-8B, ArmoRM-Llama3-8B-v0.1, Skywork-Reward-Gemma-2-27B, Skywork-Reward-Llama-3.1-8B, Nemotron-4-340B-Reward (Frick et al., 2024; Cai et al., 2024; Park et al., 2024; Wang et al., 2024a; Liu & Zeng, 2024; Wang et al., 2024c). Evaluated on Preference Proxy Evaluations (PPE), but not selected: Starling-RM-34B, Starling-RM-7B-Alpha, Eurus-RM-7B, InternLM2-1.8B-Reward, and NaiveVerbosityModel (Zhu et al., 2023a; Yuan et al., 2024; Cai et al., 2024).

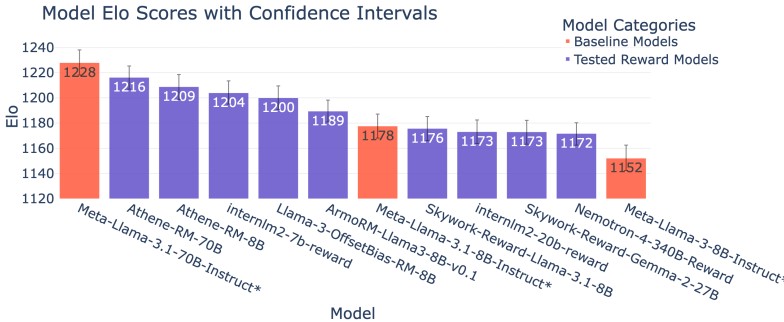

Figure 4: Post DPO performance on real human preference the Overall Category. "Model" is the reward model used to train the base model unless specified as a reference base model.

We then train Llama-3.1-8B-Instruct on each dataset using DPO producing an LLM associated with each selected reward model for real-world downstream human preference testing.

## 6.2 EVALUATION ON REAL-WORLD HUMAN PREFERENCE

We deploy the trained models to a crowd-sourced human preference platform to undergo blind evaluation from real users. We set up a cohort of 13 models which include the trained DPO models as well as Llama-3.1-8B-Instruct, Llama-3.1-70b-Instruct, and Llama-3-8B-Instruct. All models used temperature 0.2 (excluding Llama-3-8B-Instruct at temperature 0.7). Model pairs were sampled evenly with only each other for battles. Battles were collected over a six day period, from September 10th, 2024 to September 16th, 2024. In all battles, the receiving user was selected randomly. Additionally, the model names (labeled `llama-3.1-8b-dpo-test-{1,2...,9}`) were not revealed to the user until after the vote was given.

Overall, 12,190 human preference votes were collected, with an average of 2,032 battles per model, and an average of 190 battles per unique model pair. More details on battle statistics and be found in Appendix Table 39 of Appendix A.5. The resulting preference rankings are detailed in Figure 4. The preference rankings are calculated using the Bradley-Terry model, as proposed in Chiang et al. (2024).

## 7 STUDYING CORRELATION WITH DOWNSTREAM PERFORMANCE

In this section, we analyze how different metrics correlate with post-RLHF human preference scores (experimental setup detailed in Section 6.2). Our main results are displayed in Figure 5, which shows the correlations of our offline reward model evaluations against the real-world human-preference ranking from the crowdsourced platform.

On correctness metrics (left plot in Figure 5) we make several observations: (1) Mean across all domains is well correlated across all metrics, but exhibits higher correlation with AUC and Accuracy scores. (2) Math is the best individual benchmark domain in terms of predictive power. (3) ROC AUC score draws higher correlation across all benchmarks, even on benchmarks that are otherwise uncorrelated.

Turning to the right-hand side of Figure 5, the accuracy of the reward model is the best predictor of the fine-tuned LLM's preference score. Row-wise Pearson Correlation, Confidence Agreement, and Separability show some correlative power to downstream human preference rating but do not exceed accuracy. Meanwhile, metrics like the Spearman correlation and Kendall correlation have nearly zero correlation with the final human preference rating achieved by the post-DPO models. One possible reason for this trend is that accuracy measures expected preference correctness per preference pair— a much more granular scale. Other metrics involve aggregating reward model signals over higher-order preferences, such as preference for each model, as measured by correlation metrics. We consider these metrics as low granularity. Medium granularity metrics, such as Row-wise Pearson Correlation aggregate reward model signal, but do so over smaller subsets of preferences.

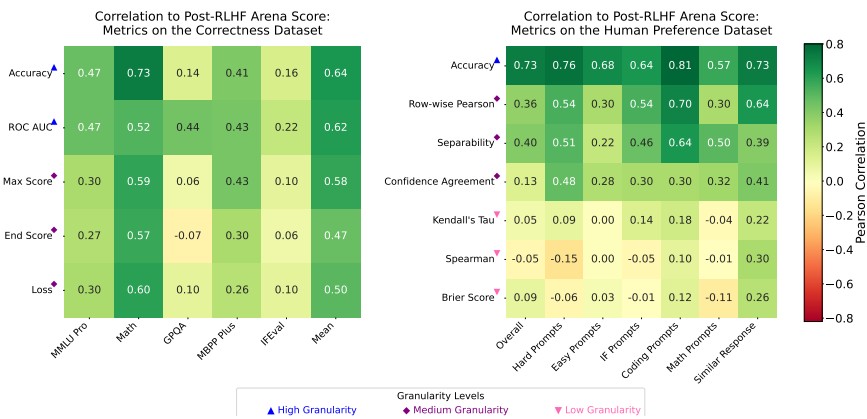

Figure 5: Pearson correlations of different metrics toward downstream human preference scores. Left: Pearson correlation between the ranking of models on 5 specific benchmarks and 5 different metrics and their respective post-DPO rankings on real human preference. Right: Pearson correlation between the ranking of models on 7 categories and 7 metrics on the Human Preference Dataset. A similar version using style controlled human preference as reference is shown in Appendix Figure 11.

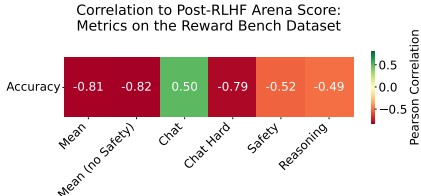

Figure 6: Pearson correlation between the ranking of models in RewardBench and their respective post-DPO rankings on real human preference. Style controlled version in Appendix Figure 12. Comments on these correlations can be found in Appendix A.6.1.

Overall, accuracy on the human preference dataset is more correlated than the correctness metrics. This is because correctness and human preference do not necessarily align. Moreover, the information contained in Loss, Max score, and End score may not prove relevant in DPO, which is off-policy. Those employing RLHF algorithms that have a higher risk of over-optimization may find these alternative measures helpful. However, when calculating correlation against style controlled ratings[4] we notice a slight decrease in correlations on the human preference dataset. Notably, the correctness preference measurements show no change, suggesting correctness preference may be more robust towards reward model preference quality, response style aside. We leave details for Appendix A.6.2.

Additionally, we observe that measuring the lower bound score may correlate more to downstream RLHF performance than the average score or upper bound score. In Figure 7, we first re-scale each category's scores to be mean 0 and SD 1, then we vary the quantile of the aggregation strategy across human preference dataset categories seen in Appendix Table 4 (Hard Prompts, Easy Prompts, etc). In this case, the 0 quantile is the minimum, and the 1 quantile is the maximum. We find that in nearly every metric, decreasing the quantile increases correlation with downstream ratings. We posit that the increase in correlation to downstream when using low quantile aggregation across metrics is because this strategy closer measures the robustness of the reward model. This is in line with previous theoretical work has suggest that pessimistic measures on reward model performance may be useful (Zhu et al., 2023b; Li et al., 2023). See Appendix A.6 for more details.

Recommendations for PPE based on these findings can be found in Appendix A.7.

---

[4]Style controlled ratings are calculated as detailed in Li et al. (2024a).

Performance Correlation vs Human Preference Category Score Quantile

Figure 7: Spearman Correlation, Confidence Agreement, and Accuracy metrics: For each metric, we take the quantiles of category scores (Hard, Easy, Instruction Following, Coding, Math, and Similar). The Pearson Correlation is calculated relative to Post-RLHF Human Preference ratings for each quantile. Notably, accuracy peaks at 0.80 correlation at low quantile aggregation.

## 8 LIMITATIONS

### 8.1 BENCHMARK LEAKAGE

We acknowledge that benchmark leakage is a very real possibility. We also consider two factors that help mitigate this issue: (1) The human preference dataset can be updated with new crowdsourced preference data at any time. This includes adapting to the most recent prompt and response distributions. (2) The correctness preference datasets can be extended to any other benchmark that becomes standard enough to be widely used.

### 8.2 LIMITS ON TESTING DOWNSTREAM PERFORMANCE

Unfortunately, end-to-end evaluation of reward models via post-RLHF LLM performance on human preference is extremely expensive and time-consuming. As such, we are limited to testing the performance of nine select models, rather than all reward models. In addition, we use DPO, an offline RL algorithm over PPO, an online algorithm, which may play more into over-optimization issues or may have different reward model requirements altogether. We encourage future work to study downstream outcomes under online RL algorithms. Moreover, we note that resource constraints necessitated experimenting with just Llama-3.1-8B-Instruct as the base policy model; additional exploration on a diverse set of base models may yield additional novel insights. With these considerations, we note that the downstream performance measured in our work is in the context of the base model and RLHF learning algorithm used, and is not a unilateral measurement of downstream outcomes in all possible configurations. Future work should experimentally verify the desired reward model behavior of other RLHF configurations.

## 9 CONCLUSION

We present PPE, a reward model benchmark explicitly tied to post-RLHF outcomes based on real human preferences. Our experiment aims to identify which metrics, applied to specific tasks, correlate most strongly with downstream performance. We find that across the board, granular measurements, such as accuracy, are the best predictors. Additionally, our results suggest that measuring lower bound performance may be more indicative of expected reward model performance in the RLHF pipeline. Overall, our evaluations achieve a 77% Pearson correlation with downstream performance, significantly improving upon previous work. Based on these results, we encourage future research to further investigate reward model quality and downstream RLHF performance under broader conditions. We fully open-source dataset creation, experimental validation, and reward model evaluation code and methods. We anticipate that the high-quality preference evaluation in PPE, combined with our post-RLHF analysis of metric predictive power, will significantly advance vital research into reward models and RLHF.

ACKNOWLEDGMENTS

This work is funded by the Sky Computing Lab at UC Berkeley. This project was also partially supported by NSF grants IIS-1901252 and CCF-2211209. We are grateful for the generous support for compute resources from RunPod.

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

# A APPENDIX

## A.1 OVERVIEW OF PPE BENCHMARK DATASETS

| Name | Num Prompts | Response per Prompt | Preference Type |
|------|-------------|---------------------|-----------------|
| Human Preference V1 | 16,038 | 2 | Real Human |
| MMLU Pro | 512 | 32 | Correctness |
| MATH | 512 | 32 | Correctness |
| GPQA | 512 | 32 | Correctness |
| IFEval | 512 | 32 | Correctness |
| MBPP Plus | 507 | 32 | Correctness |

Table 3: Released benchmarking datasets in PPE.

## A.2 DETAILED SCORES FOR THE HUMAN PREFERENCE EVALUATION DATASET

You may include other additional sections here.

| Reward Model | Accuracy | R.W. Pearson | Separability | Conf. Agree. | Kendalltau | Spearmanr | Brier Score |
|------|------|------|------|------|------|------|------|
| Ensemble-Judges (ArenaHard)[†] | 68.59 | 82.49 | 84.21 | 96.21 | 87.37 | 96.54 | 0.05 |
| Ensemble-Judges (AlpacaEval)[†] | 68.52 | 81.25 | 79.47 | 93.94 | 85.26 | 95.04 | 0.07 |
| GPT-4o-2024-08-06 (ArenaHard)[†] | 67.71 | 81.07 | 80.53 | 94.70 | 86.32 | 96.24 | 0.06 |
| Claude-3-5-Sonnet-20240620 (ArenaHard)[†] | 67.33 | 80.65 | 79.47 | 94.70 | 88.42 | 96.69 | 0.06 |
| GPT-4o-2024-08-06 (AlpacaEval)[†] | 67.13 | 77.92 | 76.32 | 90.91 | 84.21 | 93.23 | 0.07 |
| Athene-RM-70B | 66.56 | 80.69 | 84.74 | 93.94 | 82.11 | 93.23 | 0.07 |
| GPT-4o-Mini-2024-07-18 (ArenaHard)[†] | 66.46 | 78.42 | 75.26 | 92.42 | 83.16 | 93.08 | 0.07 |
| Gemini-1.5-Pro-002 (AlpacaEval)[†] | 66.09 | 82.63 | 83.16 | 96.21 | 86.32 | 95.19 | 0.05 |
| Gemini-1.5-Pro-002 (ArenaHard)[†] | 65.71 | 82.23 | 83.16 | 94.70 | 90.53 | 96.99 | 0.04 |
| Claude-3-5-Sonnet-20240620 (AlpacaEval)[†] | 65.34 | 73.91 | 74.21 | 85.61 | 71.58 | 85.26 | 0.11 |
| Llama-3.1-70B-Instruct (AlpacaEval)[†] | 65.27 | 74.81 | 79.47 | 87.88 | 72.63 | 85.56 | 0.12 |
| Gemini-1.5-Flash-002 (AlpacaEval)[†] | 65.04 | 74.29 | 78.95 | 88.64 | 74.74 | 88.72 | 0.11 |
| Athene-RM-8B | 64.59 | 76.85 | 83.68 | 91.67 | 77.89 | 90.53 | 0.10 |
| Llama-3.1-70B-Instruct (ArenaHard)[†] | 64.29 | 74.77 | 75.79 | 85.61 | 70.53 | 87.07 | 0.12 |
| Gemini-1.5-Flash-002 (ArenaHard)[†] | 63.01 | 76.12 | 76.32 | 90.91 | 76.84 | 90.23 | 0.10 |
| Starling-RM-34B | 62.92 | 70.47 | 77.37 | 78.79 | 67.37 | 81.20 | 0.15 |
| GPT-4o-Mini-2024-07-18 (AlpacaEval)[†] | 62.75 | 68.86 | 70.53 | 84.09 | 75.79 | 88.12 | 0.10 |
| Gemini-1.5-Pro-001 (ArenaHard)[†] | 62.57 | 75.92 | 81.05 | 93.18 | 85.26 | 94.44 | 0.07 |
| Skywork-Reward-Llama-3.1-8B | 62.37 | 75.51 | 78.95 | 87.88 | 71.58 | 88.12 | 0.11 |
| InternLM2-7B-Reward | 62.05 | 68.03 | 78.42 | 69.70 | 56.84 | 76.09 | 0.20 |
| Eurus-RM-7B | 62.02 | 60.37 | 75.26 | 64.39 | 51.58 | 65.26 | 0.22 |
| InternLM2-20B-Reward | 61.00 | 66.66 | 74.74 | 70.45 | 55.79 | 76.39 | 0.20 |
| ArmoRM-Llama3-8B-v0.1 | 60.57 | 71.85 | 76.84 | 84.85 | 76.84 | 89.17 | 0.10 |
| NaiveVerbosityModel | 59.81 | 32.03 | 76.32 | 35.61 | 29.47 | 33.53 | 0.33 |
| Nemotron-4-340B-Reward | 59.28 | 66.96 | 78.95 | 78.79 | 68.42 | 86.02 | 0.14 |
| Llama-3-OffsetBias-RM-8B | 59.12 | 58.86 | 65.79 | 61.36 | 51.58 | 69.02 | 0.20 |
| Starling-RM-7B-Alpha | 58.93 | 58.42 | 70.00 | 67.42 | 50.53 | 64.66 | 0.22 |
| InternLM2-1.8B-Reward | 57.22 | 47.11 | 69.47 | 41.67 | 36.84 | 54.14 | 0.28 |
| Skywork-Reward-Gemma-2-27B | 56.62 | 69.99 | 69.47 | 87.88 | 84.21 | 95.49 | 0.07 |

Table 4: Reward model and LLM judge performance on Overall subset of the human preference dataset. LLM-as-a-judge are labeled with system prompt source, and marked with †.

| Reward Model | Accuracy | R.W. Pearson | Separability | Conf. Agree. | Kendalltau | Spearmanr | Brier Score |
|---|---|---|---|---|---|---|---|
| Ensemble-Judges (ArenaHard)[†] | 69.46 | 67.05 | 74.21 | 96.88 | 83.16 | 94.44 | 0.06 |
| Claude-3-5-Sonnet-20240620 (ArenaHard)[†] | 69.25 | 67.96 | 72.11 | 97.92 | 86.32 | 95.49 | 0.06 |
| GPT-4o-2024-08-06 (ArenaHard)[†] | 68.50 | 68.17 | 71.05 | 97.92 | 85.26 | 95.94 | 0.06 |
| Ensemble-Judges (AlpacaEval)[†] | 68.32 | 66.01 | 75.26 | 96.88 | 83.16 | 94.59 | 0.07 |
| GPT-4o-Mini-2024-07-18 (ArenaHard)[†] | 66.63 | 63.55 | 71.05 | 95.83 | 82.11 | 94.29 | 0.08 |
| Gemini-1.5-Pro-002 (AlpacaEval)[†] | 66.53 | 66.85 | 72.63 | 96.88 | 84.21 | 95.49 | 0.06 |
| Athene-RM-70B | 66.43 | 67.01 | 76.84 | 96.88 | 78.95 | 92.93 | 0.08 |
| GPT-4o-2024-08-06 (AlpacaEval)[†] | 66.30 | 62.68 | 69.47 | 96.88 | 78.95 | 93.23 | 0.09 |
| Gemini-1.5-Pro-002 (ArenaHard)[†] | 65.70 | 68.57 | 68.42 | 95.83 | 83.16 | 94.44 | 0.07 |
| Llama-3.1-70B-Instruct (AlpacaEval)[†] | 64.96 | 65.76 | 65.26 | 90.62 | 70.53 | 87.82 | 0.11 |
| Llama-3.1-70B-Instruct (ArenaHard)[†] | 64.74 | 60.00 | 64.21 | 89.58 | 73.68 | 89.02 | 0.10 |
| Athene-RM-8B | 64.41 | 62.44 | 74.21 | 96.88 | 74.74 | 87.97 | 0.11 |
| Gemini-1.5-Flash-002 (AlpacaEval)[†] | 64.35 | 62.30 | 65.79 | 94.79 | 77.89 | 91.43 | 0.09 |
| Gemini-1.5-Flash-002 (ArenaHard)[†] | 64.18 | 60.68 | 67.37 | 94.79 | 81.05 | 92.18 | 0.08 |
| Claude-3-5-Sonnet-20240620 (AlpacaEval)[†] | 64.14 | 56.81 | 65.26 | 90.62 | 73.68 | 88.42 | 0.11 |
| Starling-RM-34B | 63.87 | 59.33 | 71.58 | 89.58 | 65.26 | 82.41 | 0.14 |
| Gemini-1.5-Pro-001 (ArenaHard)[†] | 63.53 | 67.93 | 68.42 | 96.88 | 85.26 | 95.19 | 0.05 |
| Eurus-RM-7B | 62.75 | 58.07 | 69.47 | 75.00 | 58.95 | 72.78 | 0.19 |
| InternLM2-7B-Reward | 62.14 | 60.77 | 67.37 | 85.42 | 65.26 | 83.16 | 0.14 |
| InternLM2-20B-Reward | 61.56 | 59.94 | 67.37 | 83.33 | 71.58 | 88.87 | 0.12 |
| GPT-4o-Mini-2024-07-18 (AlpacaEval)[†] | 61.56 | 50.96 | 59.47 | 90.62 | 72.63 | 89.02 | 0.11 |
| Skywork-Reward-Llama-3.1-8B | 61.15 | 62.46 | 68.42 | 88.54 | 70.53 | 86.62 | 0.11 |
| ArmoRM-Llama3-8B-v0.1 | 60.99 | 61.81 | 61.58 | 89.58 | 70.53 | 87.22 | 0.11 |
| NaiveVerbosityModel | 59.67 | 37.71 | 66.84 | 66.67 | 44.21 | 58.65 | 0.25 |
| Llama-3-OffsetBias-RM-8B | 59.42 | 56.03 | 59.47 | 73.96 | 62.11 | 80.15 | 0.16 |
| Nemotron-4-340B-Reward | 59.06 | 55.82 | 67.37 | 87.50 | 73.68 | 90.38 | 0.10 |
| InternLM2-1.8B-Reward | 58.49 | 52.40 | 61.58 | 63.54 | 48.42 | 63.91 | 0.21 |
| Starling-RM-7B-Alpha | 57.59 | 51.48 | 60.53 | 80.21 | 61.05 | 81.05 | 0.16 |
| Skywork-Reward-Gemma-2-27B | 56.21 | 40.13 | 38.42 | 63.54 | 70.53 | 89.02 | 0.11 |

Table 5: Reward model and LLM judge performance on Hard prompt subset of the human preference dataset. LLM-as-a-judge are labeled with system prompt source, and marked with †.

| Reward Model | Accuracy | R.W. Pearson | Separability | Conf. Agree. | Kendalltau | Spearmanr | Brier Score |
|---|---|---|---|---|---|---|---|
| Ensemble-Judges (AlpacaEval)[†] | 70.15 | 52.24 | 52.10 | 83.33 | 75.79 | 91.58 | 0.09 |
| GPT-4o-2024-08-06 (AlpacaEval)[†] | 69.97 | 52.01 | 47.37 | 83.33 | 72.63 | 90.08 | 0.09 |
| Ensemble-Judges (ArenaHard)[†] | 69.59 | 57.24 | 63.16 | 83.33 | 83.16 | 94.74 | 0.08 |
| GPT-4o-2024-08-06 (ArenaHard)[†] | 68.54 | 56.01 | 52.10 | 81.25 | 77.89 | 93.53 | 0.08 |
| GPT-4o-Mini-2024-07-18 (ArenaHard)[†] | 67.50 | 50.08 | 46.32 | 78.12 | 72.63 | 88.72 | 0.09 |
| Llama-3.1-70B-Instruct (AlpacaEval)[†] | 67.40 | 46.25 | 46.32 | 68.75 | 60.00 | 80.60 | 0.14 |
| Gemini-1.5-Pro-002 (ArenaHard)[†] | 67.08 | 55.16 | 57.37 | 90.62 | 82.11 | 94.89 | 0.06 |
| Claude-3-5-Sonnet-20240620 (AlpacaEval)[†] | 66.98 | 44.87 | 35.26 | 61.46 | 67.37 | 84.51 | 0.12 |
| Claude-3-5-Sonnet-20240620 (ArenaHard)[†] | 66.95 | 55.98 | 58.42 | 87.50 | 72.63 | 90.53 | 0.09 |
| Gemini-1.5-Flash-002 (AlpacaEval)[†] | 66.92 | 45.52 | 48.95 | 76.04 | 72.63 | 88.42 | 0.10 |
| Athene-RM-70B | 66.90 | 58.55 | 64.21 | 93.75 | 77.89 | 92.48 | 0.08 |
| Gemini-1.5-Pro-002 (AlpacaEval)[†] | 65.96 | 51.60 | 53.68 | 84.38 | 81.05 | 93.23 | 0.06 |
| GPT-4o-Mini-2024-07-18 (AlpacaEval)[†] | 65.39 | 42.05 | 25.79 | 46.88 | 69.47 | 85.71 | 0.12 |
| Athene-RM-8B | 64.49 | 53.01 | 58.95 | 83.33 | 64.21 | 83.16 | 0.13 |
| Llama-3.1-70B-Instruct (ArenaHard)[†] | 64.10 | 48.06 | 40.53 | 68.75 | 64.21 | 82.71 | 0.12 |
| Skywork-Reward-Llama-3.1-8B | 63.24 | 42.44 | 46.32 | 56.25 | 62.11 | 78.80 | 0.15 |
| Gemini-1.5-Pro-001 (ArenaHard)[†] | 62.65 | 40.53 | 54.21 | 78.12 | 80.00 | 93.68 | 0.09 |
| Eurus-RM-7B | 61.82 | 34.66 | 41.05 | 31.25 | 36.84 | 45.71 | 0.27 |
| InternLM2-7B-Reward | 61.70 | 32.69 | 34.74 | 45.83 | 45.26 | 60.60 | 0.23 |
| Starling-RM-34B | 61.41 | 33.87 | 35.79 | 41.67 | 44.21 | 60.75 | 0.22 |
| Gemini-1.5-Flash-002 (ArenaHard)[†] | 61.01 | 42.41 | 46.84 | 77.08 | 68.42 | 87.52 | 0.10 |
| InternLM2-20B-Reward | 60.37 | 40.89 | 42.63 | 51.04 | 42.11 | 57.29 | 0.23 |
| ArmoRM-Llama3-8B-v0.1 | 60.28 | 34.56 | 40.53 | 53.12 | 58.95 | 73.08 | 0.17 |
| Nemotron-4-340B-Reward | 59.58 | 45.52 | 56.32 | 68.75 | 67.37 | 84.06 | 0.13 |
| NaiveVerbosityModel | 59.24 | 12.01 | 45.79 | 5.21 | 6.32 | 8.57 | 0.40 |
| Starling-RM-7B-Alpha | 58.70 | 27.17 | 38.95 | 29.17 | 28.42 | 39.25 | 0.30 |
| Llama-3-OffsetBias-RM-8B | 58.66 | 35.23 | 29.47 | 29.17 | 43.16 | 55.49 | 0.23 |
| Skywork-Reward-Gemma-2-27B | 56.74 | 45.42 | 40.00 | 66.67 | 77.89 | 92.18 | 0.09 |
| InternLM2-1.8B-Reward | 55.54 | 30.02 | 27.89 | 15.62 | 22.11 | 29.32 | 0.30 |

Table 6: Reward model and LLM judge performance on Easy prompt subset of the human preference dataset. LLM-as-a-judge are labeled with system prompt source, and marked with †.

| Reward Model | Accuracy | R.W. Pearson | Separability | Conf. Agree. | Kendalltau | Spearmanr | Brier Score |
|---|---|---|---|---|---|---|---|
| Ensemble-Judges (ArenaHard)[†] | 69.77 | 66.89 | 70.00 | 97.09 | 83.16 | 93.68 | 0.07 |
| Claude-3-5-Sonnet-20240620 (ArenaHard)[†] | 68.38 | 70.13 | 64.74 | 92.23 | 80.00 | 91.88 | 0.07 |
| Ensemble-Judges (AlpacaEval)[†] | 67.86 | 69.18 | 70.00 | 96.12 | 86.32 | 95.04 | 0.05 |
| GPT-4o-2024-08-06 (ArenaHard)[†] | 67.51 | 60.99 | 66.84 | 96.12 | 78.95 | 92.93 | 0.08 |
| Gemini-1.5-Pro-002 (AlpacaEval)[†] | 66.78 | 68.61 | 73.16 | 97.09 | 88.42 | 96.54 | 0.04 |
| Gemini-1.5-Pro-002 (ArenaHard)[†] | 66.70 | 69.92 | 68.42 | 97.09 | 82.11 | 93.83 | 0.06 |
| Athene-RM-70B | 66.50 | 63.79 | 75.26 | 95.15 | 77.89 | 90.98 | 0.09 |
| GPT-4o-2024-08-06 (AlpacaEval)[†] | 66.09 | 64.39 | 65.26 | 92.23 | 82.11 | 93.98 | 0.06 |
| GPT-4o-Mini-2024-07-18 (ArenaHard)[†] | 65.75 | 62.88 | 73.16 | 92.23 | 76.84 | 90.53 | 0.09 |
| Gemini-1.5-Flash-002 (AlpacaEval)[†] | 65.43 | 64.33 | 65.79 | 89.32 | 82.11 | 93.38 | 0.07 |
| Athene-RM-8B | 64.77 | 60.56 | 68.42 | 90.29 | 76.84 | 89.32 | 0.09 |
| Llama-3.1-70B-Instruct (AlpacaEval)[†] | 63.68 | 63.11 | 63.16 | 79.61 | 75.79 | 88.57 | 0.10 |
| Claude-3-5-Sonnet-20240620 (AlpacaEval)[†] | 63.42 | 57.93 | 59.47 | 81.55 | 71.58 | 87.97 | 0.10 |
| Gemini-1.5-Pro-001 (ArenaHard)[†] | 63.25 | 66.39 | 62.63 | 88.35 | 80.00 | 91.13 | 0.08 |
| Llama-3.1-70B-Instruct (ArenaHard)[†] | 63.04 | 59.85 | 62.10 | 83.50 | 76.84 | 90.83 | 0.08 |
| Gemini-1.5-Flash-002 (ArenaHard)[†] | 62.66 | 60.73 | 61.05 | 87.38 | 75.79 | 89.77 | 0.09 |
| Nemotron-4-340B-Reward | 61.89 | 56.91 | 63.16 | 86.41 | 71.58 | 86.92 | 0.11 |
| InternLM2-20B-Reward | 61.89 | 57.38 | 64.74 | 79.61 | 64.21 | 83.76 | 0.15 |
| Skywork-Reward-Llama-3.1-8B | 61.41 | 57.88 | 66.32 | 81.55 | 74.74 | 88.12 | 0.10 |
| InternLM2-7B-Reward | 61.41 | 55.07 | 64.74 | 66.99 | 63.16 | 80.45 | 0.16 |
| Starling-RM-34B | 61.11 | 52.85 | 61.05 | 77.67 | 65.26 | 82.41 | 0.13 |
| GPT-4o-Mini-2024-07-18 (AlpacaEval)[†] | 61.10 | 50.62 | 43.16 | 66.99 | 72.63 | 87.82 | 0.10 |
| Eurus-RM-7B | 60.90 | 51.96 | 59.47 | 65.05 | 51.58 | 65.26 | 0.20 |
| ArmoRM-Llama3-8B-v0.1 | 60.87 | 55.71 | 56.32 | 78.64 | 76.84 | 90.53 | 0.10 |
| Llama-3-OffsetBias-RM-8B | 60.22 | 55.63 | 51.05 | 65.05 | 68.42 | 83.01 | 0.15 |
| InternLM2-1.8B-Reward | 57.27 | 38.46 | 55.79 | 39.81 | 42.11 | 59.55 | 0.23 |
| NaiveVerbosityModel | 57.07 | 31.21 | 56.84 | 32.04 | 33.68 | 47.67 | 0.29 |
| Skywork-Reward-Gemma-2-27B | 56.43 | 43.85 | 32.63 | 54.37 | 75.79 | 91.43 | 0.09 |
| Starling-RM-7B-Alpha | 55.71 | 40.10 | 48.42 | 52.43 | 44.21 | 58.20 | 0.22 |

Table 7: Reward model and LLM judge performance on If prompt subset of the human preference dataset. LLM-as-a-judge are labeled with system prompt source, and marked with †.

| Reward Model | Accuracy | R.W. Pearson | Separability | Conf. Agree. | Kendalltau | Spearmanr | Brier Score |
|---|---|---|---|---|---|---|---|
| Claude-3-5-Sonnet-20240620 (ArenaHard)[†] | 68.06 | 57.64 | 62.63 | 97.22 | 88.42 | 97.74 | 0.04 |
| Ensemble-Judges (ArenaHard)[†] | 67.98 | 58.22 | 71.58 | 91.67 | 84.21 | 96.09 | 0.05 |
| GPT-4o-2024-08-06 (ArenaHard)[†] | 67.66 | 58.16 | 65.79 | 97.22 | 88.42 | 97.29 | 0.04 |
| Ensemble-Judges (AlpacaEval)[†] | 67.47 | 55.98 | 72.11 | 94.44 | 82.11 | 94.14 | 0.06 |
| Athene-RM-70B | 66.87 | 57.57 | 70.53 | 94.44 | 81.05 | 93.23 | 0.07 |
| GPT-4o-Mini-2024-07-18 (ArenaHard)[†] | 66.08 | 53.90 | 67.90 | 100.00 | 85.26 | 96.24 | 0.05 |
| Claude-3-5-Sonnet-20240620 (AlpacaEval)[†] | 65.92 | 45.70 | 60.00 | 97.22 | 81.05 | 94.44 | 0.08 |
| Gemini-1.5-Pro-002 (AlpacaEval)[†] | 65.57 | 56.07 | 65.79 | 91.67 | 76.84 | 91.88 | 0.08 |
| GPT-4o-2024-08-06 (AlpacaEval)[†] | 65.50 | 55.66 | 62.10 | 94.44 | 86.32 | 95.94 | 0.05 |
| Athene-RM-8B | 65.22 | 57.37 | 70.00 | 94.44 | 76.84 | 92.18 | 0.09 |
| Llama-3.1-70B-Instruct (AlpacaEval)[†] | 64.40 | 54.30 | 62.10 | 94.44 | 75.79 | 92.03 | 0.09 |
| Llama-3.1-70B-Instruct (ArenaHard)[†] | 64.37 | 47.58 | 58.42 | 97.22 | 78.95 | 94.14 | 0.07 |
| Gemini-1.5-Flash-002 (AlpacaEval)[†] | 64.36 | 42.96 | 57.37 | 88.89 | 72.63 | 89.92 | 0.11 |
| Starling-RM-34B | 64.29 | 56.23 | 66.84 | 88.89 | 74.74 | 89.32 | 0.10 |
| Gemini-1.5-Pro-002 (ArenaHard)[†] | 64.18 | 54.06 | 66.32 | 90.28 | 77.89 | 92.78 | 0.08 |
| InternLM2-7B-Reward | 63.53 | 46.74 | 65.26 | 84.72 | 68.42 | 86.47 | 0.12 |
| Eurus-RM-7B | 62.98 | 57.01 | 66.32 | 81.94 | 62.11 | 78.05 | 0.16 |
| Gemini-1.5-Flash-002 (ArenaHard)[†] | 62.65 | 56.60 | 54.74 | 95.83 | 80.00 | 93.68 | 0.07 |
| InternLM2-20B-Reward | 62.10 | 47.74 | 58.95 | 90.28 | 75.79 | 91.13 | 0.09 |
| GPT-4o-Mini-2024-07-18 (AlpacaEval)[†] | 61.77 | 37.46 | 44.74 | 83.33 | 77.89 | 93.68 | 0.08 |
| Gemini-1.5-Pro-001 (ArenaHard)[†] | 61.55 | 46.75 | 56.32 | 94.44 | 75.79 | 91.43 | 0.08 |
| NaiveVerbosityModel | 61.39 | 41.83 | 63.68 | 79.17 | 48.42 | 66.02 | 0.22 |
| ArmoRM-Llama3-8B-v0.1 | 61.01 | 49.40 | 51.05 | 93.06 | 81.05 | 93.83 | 0.08 |
| Skywork-Reward-Llama-3.1-8B | 61.01 | 50.02 | 61.05 | 93.06 | 76.84 | 91.58 | 0.10 |
| Llama-3-OffsetBias-RM-8B | 59.80 | 45.80 | 48.95 | 62.50 | 64.21 | 83.01 | 0.14 |
| InternLM2-1.8B-Reward | 58.76 | 45.07 | 58.42 | 62.50 | 54.74 | 71.28 | 0.19 |
| Starling-RM-7B-Alpha | 58.71 | 46.85 | 56.32 | 76.39 | 64.21 | 78.80 | 0.15 |
| Nemotron-4-340B-Reward | 57.94 | 35.96 | 51.05 | 79.17 | 72.63 | 89.62 | 0.10 |
| Skywork-Reward-Gemma-2-27B | 56.41 | 25.46 | 26.84 | 54.17 | 64.21 | 84.51 | 0.13 |

Table 8: Reward model and LLM judge performance on Is code subset of the human preference dataset. LLM-as-a-judge are labeled with system prompt source, and marked with †.

| Reward Model | Accuracy | R.W. Pearson | Separability | Conf. Agree. | Kendalltau | Spearmanr | Brier Score |
|---|---|---|---|---|---|---|---|
| Ensemble-Judges (ArenaHard)[†] | 73.58 | 54.87 | 65.79 | 88.73 | 80.00 | 94.44 | 0.07 |
| GPT-4o-2024-08-06 (ArenaHard)[†] | 72.57 | 56.46 | 63.16 | 88.73 | 82.11 | 94.89 | 0.06 |
| Claude-3-5-Sonnet-20240620 (ArenaHard)[†] | 71.79 | 49.92 | 60.53 | 88.73 | 78.95 | 93.38 | 0.08 |
| GPT-4o-Mini-2024-07-18 (ArenaHard)[†] | 70.20 | 50.30 | 55.26 | 87.32 | 71.58 | 87.97 | 0.11 |
| Gemini-1.5-Pro-002 (ArenaHard)[†] | 69.61 | 60.91 | 58.42 | 84.51 | 77.89 | 92.63 | 0.08 |
| Ensemble-Judges (AlpacaEval)[†] | 69.09 | 52.15 | 62.10 | 91.55 | 74.74 | 91.13 | 0.09 |
| Llama-3.1-70B-Instruct (ArenaHard)[†] | 68.93 | 46.05 | 54.74 | 84.51 | 72.63 | 87.82 | 0.10 |
| Athene-RM-70B | 68.58 | 57.39 | 67.37 | 85.92 | 77.89 | 92.33 | 0.09 |
| GPT-4o-2024-08-06 (AlpacaEval)[†] | 68.21 | 53.79 | 56.84 | 88.73 | 77.89 | 92.93 | 0.08 |
| Gemini-1.5-Pro-002 (AlpacaEval)[†] | 67.25 | 55.63 | 59.47 | 88.73 | 84.21 | 95.04 | 0.07 |
| Claude-3-5-Sonnet-20240620 (AlpacaEval)[†] | 66.67 | 46.28 | 54.21 | 84.51 | 58.95 | 78.95 | 0.16 |
| Llama-3.1-70B-Instruct (AlpacaEval)[†] | 65.12 | 46.95 | 56.84 | 83.10 | 57.89 | 79.55 | 0.14 |
| Gemini-1.5-Pro-001 (ArenaHard)[†] | 64.70 | 47.86 | 51.58 | 84.51 | 77.89 | 92.63 | 0.08 |
| Gemini-1.5-Flash-002 (ArenaHard)[†] | 64.62 | 45.11 | 53.68 | 85.92 | 71.58 | 87.22 | 0.09 |
| Starling-RM-34B | 63.88 | 36.42 | 55.79 | 78.87 | 64.21 | 83.91 | 0.14 |
| GPT-4o-Mini-2024-07-18 (AlpacaEval)[†] | 63.66 | 44.85 | 50.53 | 83.10 | 65.26 | 84.51 | 0.14 |
| Athene-RM-8B | 62.85 | 42.56 | 61.05 | 83.10 | 67.37 | 85.56 | 0.12 |
| Gemini-1.5-Flash-002 (AlpacaEval)[†] | 62.70 | 41.05 | 47.90 | 74.65 | 66.32 | 83.91 | 0.11 |
| InternLM2-20B-Reward | 62.63 | 40.47 | 55.26 | 76.06 | 71.58 | 87.37 | 0.11 |
| Nemotron-4-340B-Reward | 61.60 | 48.64 | 59.47 | 87.32 | 77.89 | 93.23 | 0.09 |
| InternLM2-7B-Reward | 61.53 | 41.83 | 55.26 | 73.24 | 61.05 | 80.00 | 0.15 |
| Eurus-RM-7B | 61.31 | 35.08 | 54.21 | 57.75 | 47.37 | 64.06 | 0.22 |
| Skywork-Reward-Llama-3.1-8B | 60.65 | 43.03 | 53.16 | 77.46 | 63.16 | 81.65 | 0.14 |
| ArmoRM-Llama3-8B-v0.1 | 59.32 | 37.16 | 44.74 | 73.24 | 65.26 | 83.31 | 0.14 |
| Llama-3-OffsetBias-RM-8B | 58.96 | 31.99 | 50.00 | 70.42 | 54.74 | 71.88 | 0.20 |
| InternLM2-1.8B-Reward | 58.74 | 33.52 | 36.84 | 45.07 | 49.47 | 67.82 | 0.19 |
| Starling-RM-7B-Alpha | 58.08 | 26.79 | 38.95 | 56.34 | 54.74 | 74.59 | 0.18 |
| NaiveVerbosityModel | 57.49 | 27.69 | 60.00 | 49.30 | 30.53 | 41.05 | 0.31 |
| Skywork-Reward-Gemma-2-27B | 55.80 | 35.07 | 25.26 | 46.48 | 60.00 | 75.94 | 0.14 |

Table 9: Reward model and LLM judge performance on Math prompt subset of the human preference dataset. LLM-as-a-judge are labeled with system prompt source, and marked with †.

| Reward Model | Accuracy | R.W. Pearson | Separability | Conf. Agree. | Kendalltau | Spearmanr | Brier Score |
|---|---|---|---|---|---|---|---|
| Nemotron-4-340B-Reward | 62.65 | 56.88 | 58.95 | 62.28 | 51.58 | 68.42 | 0.19 |
| Gemini-1.5-Pro-002 (ArenaHard)[†] | 59.90 | 45.67 | 66.32 | 44.74 | 37.89 | 53.38 | 0.27 |
| Gemini-1.5-Pro-001 (ArenaHard)[†] | 58.01 | 36.29 | 52.63 | 42.11 | 41.05 | 53.23 | 0.27 |
| ArmoRM-Llama3-8B-v0.1 | 56.83 | 33.59 | 43.16 | 42.98 | 36.84 | 47.82 | 0.27 |
| Gemini-1.5-Pro-002 (AlpacaEval)[†] | 56.83 | 30.75 | 67.90 | 38.60 | 30.53 | 45.41 | 0.31 |
| Athene-RM-70B | 55.81 | 31.06 | 67.37 | 35.96 | 28.42 | 44.06 | 0.32 |
| Ensemble-Judges (ArenaHard)[†] | 55.27 | 36.57 | 66.32 | 42.11 | 37.89 | 53.68 | 0.27 |
| Skywork-Reward-Llama-3.1-8B | 54.67 | 24.79 | 55.26 | 36.84 | 29.47 | 41.50 | 0.33 |
| Skywork-Reward-Gemma-2-27B | 54.50 | 34.00 | 35.79 | 38.60 | 43.16 | 57.89 | 0.21 |
| Llama-3-OffsetBias-RM-8B | 54.04 | 30.51 | 41.58 | 42.11 | 34.74 | 49.77 | 0.26 |
| Athene-RM-8B | 54.04 | 23.29 | 64.74 | 32.46 | 25.26 | 39.85 | 0.34 |
| GPT-4o-2024-08-06 (ArenaHard)[†] | 52.74 | 29.48 | 58.95 | 40.35 | 34.74 | 53.38 | 0.29 |
| InternLM2-20B-Reward | 52.43 | 29.55 | 55.79 | 39.47 | 36.84 | 55.94 | 0.26 |
| Claude-3-5-Sonnet-20240620 (ArenaHard)[†] | 52.32 | 28.63 | 58.42 | 33.33 | 38.95 | 51.73 | 0.28 |
| Ensemble-Judges (AlpacaEval)[†] | 51.26 | 16.53 | 57.90 | 31.58 | 27.37 | 39.10 | 0.33 |
| GPT-4o-2024-08-06 (AlpacaEval)[†] | 50.18 | 12.95 | 51.05 | 31.58 | 33.68 | 50.08 | 0.30 |
| GPT-4o-Mini-2024-07-18 (ArenaHard)[†] | 50.06 | 15.15 | 51.58 | 30.70 | 28.42 | 45.71 | 0.30 |
| GPT-4o-Mini-2024-07-18 (AlpacaEval)[†] | 48.41 | -1.95 | 24.21 | 15.79 | 20.00 | 29.92 | 0.31 |
| InternLM2-1.8B-Reward | 47.86 | 2.97 | 36.32 | -3.51 | 9.47 | 20.75 | 0.37 |
| Gemini-1.5-Flash-002 (ArenaHard)[†] | 47.13 | 16.99 | 48.95 | 18.42 | 22.11 | 38.95 | 0.33 |
| Gemini-1.5-Flash-002 (AlpacaEval)[†] | 46.72 | 5.46 | 48.95 | 17.54 | 14.74 | 23.16 | 0.37 |
| InternLM2-7B-Reward | 45.77 | -3.02 | 42.63 | 9.65 | 14.74 | 21.80 | 0.36 |
| Claude-3-5-Sonnet-20240620 (AlpacaEval)[†] | 45.39 | 2.05 | 35.26 | 14.04 | 10.53 | 16.24 | 0.37 |
| Llama-3.1-70B-Instruct (AlpacaEval)[†] | 45.33 | -4.86 | 46.84 | 11.40 | 6.32 | 14.59 | 0.39 |
| Llama-3.1-70B-Instruct (ArenaHard)[†] | 45.27 | 7.88 | 45.26 | 18.42 | 20.00 | 31.88 | 0.34 |
| Eurus-RM-7B | 39.81 | -19.21 | 37.90 | -7.02 | -2.11 | -1.65 | 0.45 |
| Starling-RM-34B | 39.23 | -21.35 | 35.79 | -6.14 | 1.05 | 0.45 | 0.42 |
| Starling-RM-7B-Alpha | 38.59 | -25.59 | 32.63 | -12.28 | -3.16 | -5.41 | 0.44 |
| NaiveVerbosityModel | 6.10 | -93.99 | 52.63 | -75.44 | -94.74 | -99.10 | 0.85 |

Table 10: Reward model and LLM judge performance on Shorter won subset of the human preference dataset. LLM-as-a-judge are labeled with system prompt source, and marked with †.

| Reward Model | Accuracy | R.W. Pearson | Separability | Conf. Agree. | Kendalltau | Spearmanr | Brier Score |
|---|---|---|---|---|---|---|---|
| Ensemble-Judges (ArenaHard)[†] | 68.15 | 71.49 | 73.16 | 91.59 | 86.32 | 95.64 | 0.06 |
| Ensemble-Judges (AlpacaEval)[†] | 67.28 | 73.31 | 74.21 | 92.52 | 84.21 | 94.44 | 0.06 |
| GPT-4o-2024-08-06 (ArenaHard)[†] | 67.23 | 71.93 | 71.05 | 92.52 | 84.21 | 95.19 | 0.07 |
| Claude-3-5-Sonnet-20240620 (ArenaHard)[†] | 67.08 | 72.22 | 70.00 | 88.79 | 84.21 | 93.83 | 0.06 |
| GPT-4o-Mini-2024-07-18 (ArenaHard)[†] | 66.29 | 71.23 | 69.47 | 89.72 | 80.00 | 92.48 | 0.08 |
| Athene-RM-70B | 65.84 | 72.39 | 81.05 | 90.65 | 78.95 | 91.88 | 0.09 |
| Gemini-1.5-Pro-002 (AlpacaEval)[†] | 65.54 | 71.75 | 74.21 | 92.52 | 85.26 | 94.74 | 0.06 |
| GPT-4o-2024-08-06 (AlpacaEval)[†] | 65.45 | 71.06 | 68.42 | 88.79 | 82.11 | 93.68 | 0.07 |
| Gemini-1.5-Flash-002 (AlpacaEval)[†] | 64.88 | 66.90 | 66.84 | 88.79 | 74.74 | 88.87 | 0.10 |
| Llama-3.1-70B-Instruct (AlpacaEval)[†] | 64.86 | 71.92 | 75.26 | 88.79 | 71.58 | 86.47 | 0.11 |
| Gemini-1.5-Pro-002 (ArenaHard)[†] | 64.84 | 70.79 | 73.16 | 90.65 | 83.16 | 93.83 | 0.07 |
| Athene-RM-8B | 64.28 | 68.70 | 78.95 | 89.72 | 74.74 | 88.57 | 0.10 |
| Starling-RM-34B | 64.05 | 67.27 | 75.79 | 83.18 | 71.58 | 85.56 | 0.12 |
| Llama-3.1-70B-Instruct (ArenaHard)[†] | 63.96 | 66.05 | 68.95 | 85.98 | 72.63 | 87.52 | 0.12 |
| Claude-3-5-Sonnet-20240620 (AlpacaEval)[†] | 63.95 | 65.29 | 65.79 | 87.85 | 70.53 | 85.71 | 0.12 |
| Gemini-1.5-Flash-002 (ArenaHard)[†] | 63.26 | 66.65 | 72.63 | 88.79 | 74.74 | 89.47 | 0.10 |
| Skywork-Reward-Llama-3.1-8B | 62.83 | 71.83 | 73.68 | 97.20 | 81.05 | 92.18 | 0.08 |
| Gemini-1.5-Pro-001 (ArenaHard)[†] | 62.46 | 64.75 | 66.32 | 86.92 | 77.89 | 90.68 | 0.09 |
| Eurus-RM-7B | 62.07 | 56.73 | 68.95 | 73.83 | 57.89 | 72.03 | 0.20 |
| NaiveVerbosityModel | 61.30 | 40.25 | 68.95 | 53.27 | 34.74 | 49.92 | 0.30 |
| InternLM2-7B-Reward | 60.82 | 61.98 | 69.47 | 77.57 | 60.00 | 80.30 | 0.16 |
| GPT-4o-Mini-2024-07-18 (AlpacaEval)[†] | 60.59 | 60.26 | 57.90 | 87.85 | 75.79 | 88.87 | 0.10 |
| ArmoRM-Llama3-8B-v0.1 | 60.03 | 63.19 | 71.05 | 90.65 | 81.05 | 90.98 | 0.07 |
| Starling-RM-7B-Alpha | 59.01 | 54.50 | 64.21 | 64.49 | 49.47 | 70.83 | 0.20 |
| InternLM2-20B-Reward | 59.00 | 54.89 | 68.95 | 69.16 | 57.89 | 78.20 | 0.17 |
| Llama-3-OffsetBias-RM-8B | 58.58 | 57.04 | 58.95 | 71.96 | 64.21 | 81.80 | 0.14 |
| Nemotron-4-340B-Reward | 57.74 | 50.81 | 75.26 | 65.42 | 57.89 | 73.98 | 0.19 |
| Skywork-Reward-Gemma-2-27B | 55.93 | 54.08 | 51.58 | 76.64 | 75.79 | 90.68 | 0.10 |
| InternLM2-1.8B-Reward | 55.92 | 37.43 | 61.58 | 42.99 | 36.84 | 55.64 | 0.27 |

Table 11: Reward model and LLM judge performance on Similar response subset of the human preference dataset. LLM-as-a-judge are labeled with system prompt source, and marked with †.

| Reward Model | Accuracy | R.W. Pearson | Separability | Conf. Agree. | Kendalltau | Spearmanr | Brier Score |
|---|---|---|---|---|---|---|---|
| Ensemble-Judges (ArenaHard)[†] | 68.17 | 70.80 | 71.58 | 86.24 | 81.05 | 94.14 | 0.08 |
| GPT-4o-2024-08-06 (ArenaHard)[†] | 67.78 | 71.61 | 68.95 | 86.24 | 83.16 | 94.89 | 0.07 |
| Ensemble-Judges (AlpacaEval)[†] | 67.60 | 70.66 | 71.58 | 84.40 | 76.84 | 92.93 | 0.10 |
| GPT-4o-2024-08-06 (AlpacaEval)[†] | 66.70 | 63.51 | 66.32 | 80.73 | 76.84 | 91.73 | 0.09 |
| Claude-3-5-Sonnet-20240620 (ArenaHard)[†] | 66.42 | 68.25 | 70.53 | 86.24 | 78.95 | 93.68 | 0.08 |
| GPT-4o-Mini-2024-07-18 (ArenaHard)[†] | 66.39 | 66.39 | 67.37 | 81.65 | 78.95 | 92.03 | 0.09 |
| Athene-RM-70B | 65.53 | 68.75 | 79.47 | 83.49 | 73.68 | 90.98 | 0.12 |
| Gemini-1.5-Pro-002 (AlpacaEval)[†] | 65.37 | 70.68 | 74.74 | 87.16 | 76.84 | 91.88 | 0.10 |
| Llama-3.1-70B-Instruct (AlpacaEval)[†] | 64.79 | 65.74 | 72.11 | 78.90 | 66.32 | 85.56 | 0.13 |
| Gemini-1.5-Pro-002 (ArenaHard)[†] | 64.75 | 69.77 | 71.58 | 84.40 | 76.84 | 92.93 | 0.10 |
| Gemini-1.5-Flash-002 (AlpacaEval)[†] | 64.48 | 65.98 | 67.90 | 79.82 | 69.47 | 86.02 | 0.13 |
| Llama-3.1-70B-Instruct (ArenaHard)[†] | 64.31 | 63.74 | 67.90 | 82.57 | 70.53 | 88.87 | 0.12 |
| Claude-3-5-Sonnet-20240620 (AlpacaEval)[†] | 64.27 | 62.80 | 65.26 | 79.82 | 68.42 | 86.47 | 0.13 |
| Athene-RM-8B | 63.55 | 65.76 | 75.26 | 81.65 | 69.47 | 89.32 | 0.13 |
| Starling-RM-34B | 63.50 | 60.04 | 72.63 | 68.81 | 65.26 | 81.80 | 0.16 |
| Gemini-1.5-Flash-002 (ArenaHard)[†] | 62.97 | 64.16 | 66.84 | 77.98 | 70.53 | 88.12 | 0.12 |
| Skywork-Reward-Llama-3.1-8B | 62.94 | 68.77 | 70.53 | 87.16 | 75.79 | 90.98 | 0.10 |
| Gemini-1.5-Pro-001 (ArenaHard)[†] | 62.04 | 64.66 | 65.79 | 86.24 | 70.53 | 89.47 | 0.12 |
| Eurus-RM-7B | 61.78 | 51.70 | 71.58 | 58.72 | 52.63 | 65.86 | 0.20 |
| GPT-4o-Mini-2024-07-18 (AlpacaEval)[†] | 61.64 | 57.42 | 59.47 | 81.65 | 71.58 | 87.52 | 0.11 |
| NaiveVerbosityModel | 61.26 | 40.80 | 68.42 | 48.62 | 43.16 | 51.73 | 0.26 |
| InternLM2-7B-Reward | 61.01 | 53.18 | 66.84 | 70.64 | 58.95 | 80.30 | 0.18 |
| ArmoRM-Llama3-8B-v0.1 | 60.94 | 64.96 | 70.00 | 83.49 | 75.79 | 90.38 | 0.10 |
| Starling-RM-7B-Alpha | 59.55 | 50.50 | 67.90 | 53.21 | 55.79 | 71.43 | 0.21 |
| InternLM2-20B-Reward | 59.34 | 54.73 | 68.95 | 65.14 | 50.53 | 71.58 | 0.20 |
| Llama-3-OffsetBias-RM-8B | 59.06 | 54.04 | 55.26 | 66.06 | 54.74 | 69.47 | 0.20 |
| Nemotron-4-340B-Reward | 57.47 | 44.46 | 71.05 | 62.39 | 50.53 | 67.07 | 0.22 |
| InternLM2-1.8B-Reward | 56.17 | 41.19 | 61.58 | 38.53 | 32.63 | 50.23 | 0.28 |
| Skywork-Reward-Gemma-2-27B | 55.21 | 57.61 | 49.47 | 73.39 | 69.47 | 87.52 | 0.11 |

Table 12: Reward model and LLM judge performance on English prompt subset of the human preference dataset. LLM-as-a-judge are labeled with system prompt source, and marked with †.

| Reward Model | Accuracy | R.W. Pearson | Separability | Conf. Agree. | Kendalltau | Spearmanr | Brier Score |
|---|---|---|---|---|---|---|---|
| Ensemble-Judges (AlpacaEval)[†] | 69.68 | 73.76 | 74.21 | 94.31 | 90.53 | 97.74 | 0.03 |
| Ensemble-Judges (ArenaHard)[†] | 69.09 | 75.81 | 76.84 | 93.50 | 86.32 | 95.79 | 0.06 |
| Claude-3-5-Sonnet-20240620 (ArenaHard)[†] | 68.48 | 75.18 | 75.26 | 91.87 | 86.32 | 96.39 | 0.05 |
| Athene-RM-70B | 67.86 | 73.24 | 76.84 | 91.87 | 82.11 | 94.89 | 0.07 |
| GPT-4o-2024-08-06 (AlpacaEval)[†] | 67.66 | 72.18 | 72.63 | 98.37 | 93.68 | 98.65 | 0.03 |
| GPT-4o-2024-08-06 (ArenaHard)[†] | 67.63 | 71.24 | 73.16 | 91.87 | 82.11 | 94.74 | 0.07 |
| Gemini-1.5-Pro-002 (AlpacaEval)[†] | 67.01 | 73.72 | 80.00 | 94.31 | 88.42 | 97.14 | 0.05 |
| Gemini-1.5-Pro-002 (ArenaHard)[†] | 66.93 | 74.39 | 75.26 | 90.24 | 82.11 | 94.29 | 0.07 |
| Claude-3-5-Sonnet-20240620 (AlpacaEval)[†] | 66.68 | 67.72 | 60.53 | 80.49 | 81.05 | 94.14 | 0.07 |
| GPT-4o-Mini-2024-07-18 (ArenaHard)[†] | 66.55 | 71.23 | 72.63 | 90.24 | 82.11 | 94.44 | 0.07 |
| Athene-RM-8B | 65.91 | 70.37 | 80.53 | 92.68 | 82.11 | 95.04 | 0.07 |
| Llama-3.1-70B-Instruct (AlpacaEval)[†] | 65.87 | 65.70 | 68.95 | 83.74 | 75.79 | 90.53 | 0.09 |
| Gemini-1.5-Flash-002 (AlpacaEval)[†] | 65.75 | 70.61 | 67.90 | 86.99 | 87.37 | 96.84 | 0.06 |
| Llama-3.1-70B-Instruct (ArenaHard)[†] | 64.25 | 68.81 | 65.26 | 82.11 | 80.00 | 93.38 | 0.09 |
| GPT-4o-Mini-2024-07-18 (AlpacaEval)[†] | 64.17 | 62.56 | 54.74 | 78.05 | 83.16 | 94.44 | 0.06 |
| InternLM2-7B-Reward | 63.36 | 63.58 | 65.79 | 69.11 | 62.11 | 84.21 | 0.16 |
| Gemini-1.5-Pro-001 (ArenaHard)[†] | 63.24 | 70.19 | 70.53 | 87.80 | 80.00 | 94.14 | 0.08 |
| InternLM2-20B-Reward | 63.10 | 63.69 | 72.11 | 76.42 | 64.21 | 86.17 | 0.16 |
| Gemini-1.5-Flash-002 (ArenaHard)[†] | 63.06 | 68.96 | 71.05 | 86.18 | 77.89 | 93.38 | 0.08 |
| Eurus-RM-7B | 62.32 | 56.17 | 61.05 | 67.48 | 66.32 | 75.49 | 0.16 |
| Starling-RM-34B | 62.19 | 58.76 | 64.21 | 73.17 | 70.53 | 86.32 | 0.12 |
| Skywork-Reward-Llama-3.1-8B | 61.66 | 64.18 | 70.53 | 75.61 | 73.68 | 87.52 | 0.11 |
| Nemotron-4-340B-Reward | 61.57 | 67.30 | 72.63 | 83.74 | 76.84 | 90.53 | 0.10 |
| ArmoRM-Llama3-8B-v0.1 | 60.11 | 59.89 | 58.95 | 66.67 | 73.68 | 90.53 | 0.12 |
| Llama-3-OffsetBias-RM-8B | 59.20 | 48.58 | 55.79 | 52.85 | 53.68 | 69.17 | 0.19 |
| InternLM2-1.8B-Reward | 58.55 | 44.78 | 55.26 | 43.90 | 41.05 | 56.24 | 0.24 |
| Skywork-Reward-Gemma-2-27B | 58.40 | 58.79 | 61.05 | 83.74 | 83.16 | 95.19 | 0.06 |
| Starling-RM-7B-Alpha | 58.13 | 40.90 | 59.47 | 55.28 | 48.42 | 60.75 | 0.22 |
| NaiveVerbosityModel | 57.98 | 21.46 | 64.21 | 30.89 | 21.05 | 27.52 | 0.36 |

Table 13: Reward model and LLM judge performance on Non english prompt subset of the human preference dataset. LLM-as-a-judge are labeled with system prompt source, and marked with †.

| Reward Model | Accuracy | R.W. Pearson | Separability | Conf. Agree. | Kendalltau | Spearmanr | Brier Score |
|---|---|---|---|---|---|---|---|
| Ensemble-Judges (AlpacaEval)[†] | 67.91 | 52.67 | 54.21 | 93.33 | 80.00 | 94.14 | 0.07 |
| Claude-3-5-Sonnet-20240620 (ArenaHard)[†] | 67.03 | 50.91 | 48.42 | 90.00 | 78.95 | 93.38 | 0.08 |
| Athene-RM-70B | 66.39 | 45.24 | 61.05 | 90.00 | 83.16 | 93.83 | 0.07 |
| Gemini-1.5-Pro-002 (AlpacaEval)[†] | 66.27 | 49.83 | 58.42 | 93.33 | 82.11 | 93.38 | 0.08 |
| Ensemble-Judges (ArenaHard)[†] | 66.15 | 53.77 | 47.37 | 86.67 | 77.89 | 92.33 | 0.07 |
| GPT-4o-2024-08-06 (ArenaHard)[†] | 65.37 | 49.18 | 52.10 | 90.00 | 76.84 | 92.18 | 0.08 |
| GPT-4o-Mini-2024-07-18 (ArenaHard)[†] | 65.29 | 51.87 | 44.74 | 76.67 | 66.32 | 86.47 | 0.12 |
| Gemini-1.5-Flash-002 (AlpacaEval)[†] | 65.10 | 40.01 | 46.32 | 86.67 | 71.58 | 89.17 | 0.09 |
| Claude-3-5-Sonnet-20240620 (AlpacaEval)[†] | 64.89 | 47.98 | 43.16 | 88.33 | 69.47 | 87.52 | 0.11 |
| InternLM2-20B-Reward | 64.62 | 42.76 | 48.42 | 56.67 | 65.26 | 83.91 | 0.12 |
| Athene-RM-8B | 64.45 | 42.41 | 60.00 | 86.67 | 81.05 | 94.59 | 0.07 |
| Gemini-1.5-Pro-002 (ArenaHard)[†] | 64.16 | 49.86 | 51.05 | 80.00 | 76.84 | 91.88 | 0.08 |
| InternLM2-7B-Reward | 63.87 | 44.35 | 41.05 | 53.33 | 70.53 | 89.17 | 0.11 |
| GPT-4o-2024-08-06 (AlpacaEval)[†] | 63.53 | 43.47 | 51.58 | 90.00 | 83.16 | 94.89 | 0.06 |
| Llama-3.1-70B-Instruct (ArenaHard)[†] | 63.04 | 32.00 | 48.42 | 81.67 | 60.00 | 81.65 | 0.14 |
| Llama-3.1-70B-Instruct (AlpacaEval)[†] | 63.03 | 36.40 | 47.90 | 68.33 | 67.37 | 86.17 | 0.13 |
| Starling-RM-34B | 62.52 | 40.66 | 56.32 | 85.00 | 71.58 | 86.32 | 0.11 |
| Gemini-1.5-Flash-002 (ArenaHard)[†] | 62.48 | 43.33 | 46.32 | 83.33 | 73.68 | 89.02 | 0.09 |
| Gemini-1.5-Pro-001 (ArenaHard)[†] | 62.09 | 36.12 | 41.05 | 75.00 | 71.58 | 89.77 | 0.09 |
| GPT-4o-Mini-2024-07-18 (AlpacaEval)[†] | 61.43 | 38.81 | 23.68 | 55.00 | 63.16 | 83.01 | 0.14 |
| Eurus-RM-7B | 61.18 | 39.05 | 44.21 | 70.00 | 65.26 | 81.05 | 0.14 |
| InternLM2-1.8B-Reward | 60.08 | 38.02 | 42.63 | 40.00 | 51.58 | 70.83 | 0.20 |
| Skywork-Reward-Gemma-2-27B | 59.16 | 22.83 | 26.84 | 75.00 | 86.32 | 96.09 | 0.06 |
| Nemotron-4-340B-Reward | 58.07 | 28.62 | 32.63 | 45.00 | 52.63 | 72.33 | 0.18 |
| Llama-3-OffsetBias-RM-8B | 57.48 | 27.04 | 27.37 | 28.33 | 52.63 | 68.12 | 0.20 |
| Skywork-Reward-Llama-3.1-8B | 57.23 | 38.20 | 37.37 | 53.33 | 64.21 | 81.20 | 0.13 |
| ArmoRM-Llama3-8B-v0.1 | 56.64 | 18.09 | 26.84 | 28.33 | 46.32 | 59.40 | 0.21 |
| NaiveVerbosityModel | 56.55 | 19.66 | 48.95 | 11.67 | 14.74 | 21.05 | 0.36 |
| Starling-RM-7B-Alpha | 54.29 | 7.14 | 28.42 | 18.33 | 35.79 | 47.37 | 0.23 |

Table 14: Reward model and LLM judge performance on Chinese prompt subset of the human preference dataset. LLM-as-a-judge are labeled with system prompt source, and marked with †.

| Reward Model | Accuracy | R.W. Pearson | Separability | Conf. Agree. | Kendalltau | Spearmanr | Brier Score |
|---|---|---|---|---|---|---|---|
| Ensemble-Judges (ArenaHard)[†] | 70.37 | 50.61 | 53.16 | 92.86 | 77.89 | 92.63 | 0.09 |
| Ensemble-Judges (AlpacaEval)[†] | 69.43 | 51.76 | 57.90 | 92.86 | 80.00 | 94.44 | 0.06 |
| Claude-3-5-Sonnet-20240620 (ArenaHard)[†] | 68.63 | 44.71 | 50.53 | 85.71 | 70.53 | 87.97 | 0.09 |
| GPT-4o-2024-08-06 (AlpacaEval)[†] | 68.58 | 42.94 | 38.95 | 91.07 | 77.89 | 93.83 | 0.08 |
| GPT-4o-2024-08-06 (ArenaHard)[†] | 68.54 | 43.94 | 47.37 | 89.29 | 70.53 | 89.02 | 0.10 |
| Athene-RM-70B | 68.49 | 48.66 | 58.42 | 94.64 | 77.89 | 90.68 | 0.09 |
| Gemini-1.5-Pro-002 (ArenaHard)[†] | 67.23 | 49.82 | 53.68 | 87.50 | 73.68 | 89.32 | 0.10 |
| Gemini-1.5-Pro-002 (AlpacaEval)[†] | 66.20 | 50.01 | 58.42 | 92.86 | 78.95 | 93.38 | 0.07 |
| Claude-3-5-Sonnet-20240620 (AlpacaEval)[†] | 66.13 | 42.56 | 45.79 | 85.71 | 76.84 | 89.62 | 0.10 |
| Llama-3.1-70B-Instruct (AlpacaEval)[†] | 65.65 | 38.73 | 47.90 | 92.86 | 66.32 | 85.56 | 0.12 |
| GPT-4o-Mini-2024-07-18 (ArenaHard)[†] | 65.49 | 40.39 | 45.26 | 85.71 | 75.79 | 91.28 | 0.09 |
| Gemini-1.5-Flash-002 (AlpacaEval)[†] | 65.21 | 42.35 | 50.00 | 94.64 | 75.79 | 91.73 | 0.09 |
| Athene-RM-8B | 64.87 | 41.89 | 55.79 | 91.07 | 71.58 | 86.62 | 0.10 |
| Nemotron-4-340B-Reward | 63.86 | 41.06 | 52.10 | 87.50 | 72.63 | 87.07 | 0.10 |
| GPT-4o-Mini-2024-07-18 (AlpacaEval)[†] | 63.82 | 31.28 | 23.68 | 71.43 | 82.11 | 93.83 | 0.08 |
| Llama-3.1-70B-Instruct (ArenaHard)[†] | 63.37 | 28.42 | 40.53 | 69.64 | 64.21 | 81.80 | 0.14 |
| Gemini-1.5-Flash-002 (ArenaHard)[†] | 63.26 | 31.97 | 42.63 | 76.79 | 67.37 | 85.56 | 0.12 |
| Eurus-RM-7B | 62.84 | 33.63 | 43.68 | 76.79 | 56.84 | 73.38 | 0.16 |
| Gemini-1.5-Pro-001 (ArenaHard)[†] | 62.08 | 43.28 | 46.32 | 78.57 | 70.53 | 88.12 | 0.11 |
| Skywork-Reward-Llama-3.1-8B | 61.17 | 23.32 | 41.58 | 73.21 | 65.26 | 84.51 | 0.13 |
| InternLM2-7B-Reward | 61.08 | 30.92 | 41.58 | 46.43 | 58.95 | 78.05 | 0.15 |
| Starling-RM-34B | 60.98 | 36.02 | 36.32 | 73.21 | 63.16 | 80.00 | 0.13 |
| InternLM2-20B-Reward | 60.43 | 26.87 | 39.47 | 30.36 | 60.00 | 78.50 | 0.16 |
| ArmoRM-Llama3-8B-v0.1 | 60.33 | 38.52 | 35.26 | 83.93 | 74.74 | 90.23 | 0.09 |
| Starling-RM-7B-Alpha | 59.41 | 31.55 | 38.95 | 69.64 | 53.68 | 66.77 | 0.19 |
| Llama-3-OffsetBias-RM-8B | 59.04 | 25.82 | 30.53 | 50.00 | 48.42 | 68.27 | 0.19 |
| NaiveVerbosityModel | 59.04 | 10.26 | 34.21 | 33.93 | 29.47 | 38.95 | 0.29 |
| InternLM2-1.8B-Reward | 57.65 | 26.88 | 25.79 | 17.86 | 45.26 | 60.75 | 0.21 |
| Skywork-Reward-Gemma-2-27B | 56.26 | 29.71 | 23.68 | 50.00 | 64.21 | 82.86 | 0.14 |

Table 15: Reward model and LLM judge performance on Russian prompt subset of the human preference dataset. LLM-as-a-judge are labeled with system prompt source, and marked with †.

| Reward Model | Accuracy | R.W. Pearson | Separability | Conf. Agree. | Kendalltau | Spearmanr | Brier Score |
|---|---|---|---|---|---|---|---|
| Ensemble-Judges (ArenaHard)[†] | 75.16 | 38.73 | 38.42 | 84.62 | 73.68 | 88.42 | 0.10 |
| Claude-3-5-Sonnet-20240620 (ArenaHard)[†] | 72.49 | 30.32 | 23.16 | 66.67 | 65.26 | 81.50 | 0.12 |
| GPT-4o-2024-08-06 (ArenaHard)[†] | 71.03 | 31.32 | 24.74 | 84.62 | 72.63 | 85.86 | 0.10 |
| Gemini-1.5-Pro-002 (ArenaHard)[†] | 70.64 | 29.57 | 27.89 | 76.92 | 72.63 | 87.22 | 0.11 |
| GPT-4o-2024-08-06 (AlpacaEval)[†] | 69.71 | 21.47 | 21.05 | 74.36 | 72.63 | 88.27 | 0.10 |
| Ensemble-Judges (AlpacaEval)[†] | 68.88 | 15.78 | 27.37 | 71.79 | 60.00 | 78.05 | 0.14 |
| Athene-RM-70B | 67.71 | 11.39 | 33.68 | 76.92 | 65.26 | 84.21 | 0.13 |
| Nemotron-4-340B-Reward | 66.86 | 27.91 | 26.84 | 71.79 | 62.11 | 83.16 | 0.12 |
| Llama-3.1-70B-Instruct (AlpacaEval)[†] | 66.86 | 27.69 | 25.79 | 66.67 | 51.58 | 69.17 | 0.17 |
| Gemini-1.5-Flash-002 (AlpacaEval)[†] | 66.86 | 18.29 | 24.21 | 61.54 | 54.74 | 73.38 | 0.15 |
| Gemini-1.5-Pro-002 (AlpacaEval)[†] | 66.29 | 8.72 | 33.68 | 69.23 | 69.47 | 84.81 | 0.13 |
| GPT-4o-Mini-2024-07-18 (ArenaHard)[†] | 66.00 | 13.41 | 11.58 | 61.54 | 70.53 | 86.32 | 0.11 |
| Athene-RM-8B | 65.43 | 3.68 | 37.37 | 76.92 | 67.37 | 83.31 | 0.12 |
| Gemini-1.5-Flash-002 (ArenaHard)[†] | 65.32 | 19.95 | 15.79 | 43.59 | 57.89 | 75.64 | 0.16 |
| Llama-3.1-70B-Instruct (ArenaHard)[†] | 64.66 | 21.95 | 17.37 | 48.72 | 52.63 | 68.42 | 0.16 |
| Claude-3-5-Sonnet-20240620 (AlpacaEval)[†] | 63.69 | 11.97 | 7.37 | 20.51 | 46.32 | 61.65 | 0.20 |
| Starling-RM-34B | 63.43 | 11.24 | 11.58 | 46.15 | 49.47 | 64.81 | 0.19 |
| Gemini-1.5-Pro-001 (ArenaHard)[†] | 63.33 | 16.68 | 15.26 | 48.72 | 61.05 | 82.26 | 0.14 |
| Eurus-RM-7B | 62.57 | 14.76 | 8.95 | 41.03 | 44.21 | 56.54 | 0.22 |
| InternLM2-7B-Reward | 62.29 | 12.92 | 11.05 | 38.46 | 57.89 | 78.05 | 0.16 |
| GPT-4o-Mini-2024-07-18 (AlpacaEval)[†] | 62.29 | 14.84 | 10.00 | 33.33 | 48.42 | 66.17 | 0.18 |
| InternLM2-20B-Reward | 61.71 | 18.35 | 24.21 | 61.54 | 60.00 | 79.40 | 0.15 |
| ArmoRM-Llama3-8B-v0.1 | 60.86 | -8.08 | 19.47 | 46.15 | 57.89 | 71.73 | 0.16 |
| Skywork-Reward-Llama-3.1-8B | 59.71 | -4.01 | 20.00 | 53.85 | 57.89 | 72.03 | 0.16 |
| NaiveVerbosityModel | 56.86 | 17.14 | 8.42 | 12.82 | -2.11 | -4.36 | 0.36 |
| Llama-3-OffsetBias-RM-8B | 56.57 | -4.02 | 13.68 | 30.77 | 46.32 | 56.69 | 0.21 |
| Starling-RM-7B-Alpha | 56.29 | 6.70 | 7.89 | 23.08 | 34.74 | 47.67 | 0.24 |
| InternLM2-1.8B-Reward | 55.14 | 13.77 | 7.37 | 30.77 | 32.63 | 40.75 | 0.24 |
| Skywork-Reward-Gemma-2-27B | 54.57 | -11.99 | 6.84 | 23.08 | 45.26 | 60.45 | 0.19 |

Table 16: Reward model and LLM judge performance on German prompt subset of the human preference dataset. LLM-as-a-judge are labeled with system prompt source, and marked with †.

| Reward Model | Accuracy | R.W. Pearson | Separability | Conf. Agree. | Kendalltau | Spearmanr | Brier Score |
|---|---|---|---|---|---|---|---|
| Athene-RM-70B | 71.10 | 46.16 | 37.37 | 84.21 | 67.37 | 83.76 | 0.14 |
| Ensemble-Judges (AlpacaEval)[†] | 69.63 | 32.44 | 34.21 | 52.63 | 63.16 | 82.71 | 0.13 |
| Skywork-Reward-Llama-3.1-8B | 68.81 | 40.32 | 22.11 | 68.42 | 58.95 | 78.20 | 0.14 |
| Ensemble-Judges (ArenaHard)[†] | 68.45 | 33.85 | 25.79 | 65.79 | 61.05 | 78.50 | 0.14 |
| Gemini-1.5-Pro-002 (AlpacaEval)[†] | 68.06 | 28.63 | 28.42 | 50.00 | 66.32 | 84.36 | 0.12 |
| Claude-3-5-Sonnet-20240620 (AlpacaEval)[†] | 67.59 | 27.29 | 12.11 | 36.84 | 57.89 | 78.95 | 0.15 |
| Llama-3.1-70B-Instruct (AlpacaEval)[†] | 66.97 | 24.59 | 19.47 | 52.63 | 61.05 | 78.20 | 0.15 |
| GPT-4o-2024-08-06 (AlpacaEval)[†] | 66.97 | 34.79 | 27.37 | 44.74 | 66.32 | 86.32 | 0.13 |
| GPT-4o-2024-08-06 (ArenaHard)[†] | 66.67 | 30.49 | 25.26 | 63.16 | 63.16 | 81.05 | 0.13 |
| InternLM2-20B-Reward | 66.51 | 36.27 | 20.00 | 18.42 | 55.79 | 72.33 | 0.18 |
| Gemini-1.5-Pro-002 (ArenaHard)[†] | 66.36 | 29.17 | 21.05 | 73.68 | 61.05 | 79.85 | 0.14 |
| Athene-RM-8B | 65.60 | 31.00 | 32.63 | 63.16 | 60.00 | 78.65 | 0.14 |
| GPT-4o-Mini-2024-07-18 (ArenaHard)[†] | 65.14 | 29.31 | 25.79 | 73.68 | 74.74 | 89.92 | 0.12 |
| Gemini-1.5-Flash-002 (AlpacaEval)[†] | 64.81 | 21.30 | 18.42 | 50.00 | 66.32 | 84.36 | 0.13 |
| GPT-4o-Mini-2024-07-18 (AlpacaEval)[†] | 64.68 | 14.42 | 18.42 | 31.58 | 60.00 | 79.70 | 0.14 |
| Claude-3-5-Sonnet-20240620 (ArenaHard)[†] | 64.68 | 27.59 | 21.05 | 55.26 | 65.26 | 86.02 | 0.12 |
| Gemini-1.5-Flash-002 (ArenaHard)[†] | 63.68 | 20.76 | 24.21 | 65.79 | 64.21 | 80.30 | 0.14 |
| InternLM2-7B-Reward | 63.30 | 30.05 | 9.47 | -26.32 | 49.47 | 68.42 | 0.20 |
| Llama-3.1-70B-Instruct (ArenaHard)[†] | 63.13 | 10.68 | 17.89 | 73.68 | 57.89 | 78.80 | 0.15 |
| Llama-3-OffsetBias-RM-8B | 62.39 | 28.23 | 16.32 | 63.16 | 25.26 | 38.50 | 0.26 |
| ArmoRM-Llama3-8B-v0.1 | 62.39 | 29.54 | 23.16 | 60.53 | 43.16 | 58.65 | 0.20 |
| Gemini-1.5-Pro-001 (ArenaHard)[†] | 62.24 | 19.36 | 13.16 | 57.89 | 60.00 | 78.95 | 0.13 |
| Eurus-RM-7B | 61.47 | 30.57 | 15.79 | 44.74 | 50.53 | 71.43 | 0.17 |
| Nemotron-4-340B-Reward | 61.47 | 17.85 | 26.84 | 31.58 | 44.21 | 52.63 | 0.23 |
| Starling-RM-34B | 60.09 | 16.40 | 14.21 | 68.42 | 55.79 | 70.98 | 0.17 |
| InternLM2-1.8B-Reward | 57.34 | 19.72 | 6.32 | -7.89 | 38.95 | 54.59 | 0.21 |
| NaiveVerbosityModel | 56.88 | 9.00 | 8.42 | -28.95 | 15.79 | 20.90 | 0.25 |
| Starling-RM-7B-Alpha | 55.96 | 18.12 | 16.32 | 44.74 | 44.21 | 57.44 | 0.23 |
| Skywork-Reward-Gemma-2-27B | 55.05 | 8.51 | 20.53 | 55.26 | 42.11 | 56.54 | 0.20 |

Table 17: Reward model and LLM judge performance on Korean prompt subset of the human preference dataset. LLM-as-a-judge are labeled with system prompt source, and marked with †.

| Reward Model | Accuracy | R.W. Pearson | Separability | Conf. Agree. | Kendalltau | Spearmanr | Brier Score |
|---|---|---|---|---|---|---|---|
| Claude-3-5-Sonnet-20240620 (AlpacaEval)[†] | 73.36 | 37.78 | 6.32 | 58.33 | 69.47 | 87.22 | 0.11 |
| Athene-RM-8B | 71.89 | 39.72 | 14.21 | 54.17 | 67.37 | 87.07 | 0.10 |
| Ensemble-Judges (AlpacaEval)[†] | 71.36 | 36.61 | 11.05 | 70.83 | 71.58 | 86.62 | 0.11 |
| Llama-3.1-70B-Instruct (AlpacaEval)[†] | 70.05 | 37.95 | 6.32 | 62.50 | 62.11 | 81.50 | 0.11 |
| Claude-3-5-Sonnet-20240620 (ArenaHard)[†] | 68.52 | 33.33 | 14.74 | 75.00 | 72.63 | 89.62 | 0.10 |
| Athene-RM-70B | 68.20 | 33.11 | 18.42 | 50.00 | 72.63 | 87.82 | 0.13 |
| GPT-4o-Mini-2024-07-18 (ArenaHard)[†] | 68.20 | 41.02 | 8.95 | 58.33 | 62.11 | 80.75 | 0.13 |
| Gemini-1.5-Flash-002 (AlpacaEval)[†] | 67.44 | 35.21 | 14.21 | 66.67 | 62.11 | 81.20 | 0.13 |
| GPT-4o-Mini-2024-07-18 (AlpacaEval)[†] | 67.28 | 31.60 | 0.53 | 54.17 | 65.26 | 82.11 | 0.12 |
| Gemini-1.5-Pro-002 (AlpacaEval)[†] | 66.98 | 33.95 | 14.74 | 54.17 | 64.21 | 83.46 | 0.12 |
| Skywork-Reward-Llama-3.1-8B | 66.82 | 28.61 | 9.47 | 83.33 | 64.21 | 77.59 | 0.14 |
| InternLM2-7B-Reward | 66.36 | 19.15 | 16.32 | 25.00 | 53.68 | 70.53 | 0.16 |
| Ensemble-Judges (ArenaHard)[†] | 65.79 | 31.49 | 16.84 | 62.50 | 71.58 | 87.37 | 0.11 |
| Starling-RM-34B | 64.98 | 27.05 | 16.32 | 54.17 | 61.05 | 79.70 | 0.15 |
| GPT-4o-2024-08-06 (AlpacaEval)[†] | 64.52 | 29.56 | 5.79 | 37.50 | 64.21 | 82.11 | 0.13 |
| GPT-4o-2024-08-06 (ArenaHard)[†] | 64.10 | 28.43 | 15.26 | 58.33 | 69.47 | 86.47 | 0.12 |
| Llama-3.1-70B-Instruct (ArenaHard)[†] | 64.02 | 22.78 | 3.16 | 54.17 | 54.74 | 75.79 | 0.16 |
| Nemotron-4-340B-Reward | 63.59 | 28.08 | 8.95 | 37.50 | 67.37 | 83.46 | 0.13 |
| Skywork-Reward-Gemma-2-27B | 63.13 | 12.65 | 6.32 | 50.00 | 49.47 | 64.21 | 0.18 |
| InternLM2-20B-Reward | 63.13 | 21.49 | 9.47 | -4.17 | 58.95 | 80.15 | 0.16 |
| Gemini-1.5-Flash-002 (ArenaHard)[†] | 63.03 | 33.38 | 7.89 | 54.17 | 62.11 | 82.26 | 0.11 |
| Gemini-1.5-Pro-002 (ArenaHard)[†] | 62.91 | 22.44 | 15.79 | 62.50 | 60.00 | 79.85 | 0.14 |
| NaiveVerbosityModel | 62.21 | 18.81 | 5.26 | 4.17 | 27.37 | 29.92 | 0.27 |
| Eurus-RM-7B | 61.29 | 20.76 | 3.68 | 20.83 | 47.37 | 63.61 | 0.19 |
| ArmoRM-Llama3-8B-v0.1 | 60.37 | 12.93 | 9.47 | 75.00 | 22.11 | 33.08 | 0.24 |
| Llama-3-OffsetBias-RM-8B | 59.91 | 17.63 | 11.58 | 66.67 | 36.84 | 53.53 | 0.22 |
| Gemini-1.5-Pro-001 (ArenaHard)[†] | 59.51 | 15.30 | 3.16 | 66.67 | 51.58 | 70.38 | 0.15 |
| InternLM2-1.8B-Reward | 58.99 | 15.75 | 8.42 | -20.83 | 36.84 | 53.98 | 0.22 |
| Starling-RM-7B-Alpha | 58.06 | 23.72 | 8.42 | 54.17 | 10.53 | 14.14 | 0.32 |

Table 18: Reward model and LLM judge performance on Japanese prompt subset of the human preference dataset. LLM-as-a-judge are labeled with system prompt source, and marked with †.

| Reward Model | Accuracy | R.W. Pearson | Separability | Conf. Agree. | Kendalltau | Spearmanr | Brier Score |
|---|---|---|---|---|---|---|---|
| Ensemble-Judges (AlpacaEval)[†] | 72.11 | 31.81 | 5.79 | 36.84 | 20.00 | 30.53 | 0.28 |
| GPT-4o-2024-08-06 (AlpacaEval)[†] | 70.53 | 23.71 | 0.00 | 100.00 | 35.79 | 48.42 | 0.22 |
| GPT-4o-2024-08-06 (ArenaHard)[†] | 70.29 | 24.79 | 4.21 | 89.47 | 43.16 | 59.55 | 0.21 |
| Athene-RM-70B | 69.47 | 24.25 | 17.37 | 89.47 | 35.79 | 49.62 | 0.23 |
| Claude-3-5-Sonnet-20240620 (AlpacaEval)[†] | 68.42 | 28.53 | 1.58 | 100.00 | 20.00 | 33.83 | 0.26 |
| Llama-3.1-70B-Instruct (ArenaHard)[†] | 67.93 | 29.52 | 6.32 | 78.95 | 25.26 | 32.63 | 0.28 |
| Skywork-Reward-Llama-3.1-8B | 67.89 | 20.95 | 7.37 | 89.47 | 35.79 | 52.33 | 0.21 |
| Llama-3.1-70B-Instruct (AlpacaEval)[†] | 67.89 | 27.03 | 2.63 | 100.00 | 32.63 | 49.77 | 0.22 |
| NaiveVerbosityModel | 67.37 | 24.77 | 2.11 | 100.00 | 25.26 | 34.89 | 0.24 |
| Gemini-1.5-Flash-002 (AlpacaEval)[†] | 67.37 | 29.36 | 4.74 | 68.42 | 25.26 | 37.44 | 0.26 |
| InternLM2-7B-Reward | 67.37 | 23.65 | 2.63 | 78.95 | 23.16 | 34.89 | 0.24 |
| Starling-RM-34B | 66.84 | 23.40 | 2.11 | 78.95 | 13.68 | 20.30 | 0.30 |
| Ensemble-Judges (ArenaHard)[†] | 66.47 | 20.45 | 12.63 | 47.37 | 28.42 | 40.15 | 0.24 |
| Gemini-1.5-Pro-002 (AlpacaEval)[†] | 66.32 | 19.40 | 11.05 | 47.37 | 24.21 | 38.05 | 0.25 |
| Starling-RM-7B-Alpha | 65.79 | 32.43 | 1.58 | 68.42 | 6.32 | 6.02 | 0.30 |
| InternLM2-20B-Reward | 65.26 | 24.19 | 1.05 | 100.00 | 21.05 | 32.78 | 0.25 |
| GPT-4o-Mini-2024-07-18 (AlpacaEval)[†] | 64.74 | 22.02 | 0.00 | 100.00 | 11.58 | 14.89 | 0.27 |
| Claude-3-5-Sonnet-20240620 (ArenaHard)[†] | 64.74 | 21.07 | 8.95 | 5.26 | 24.21 | 36.54 | 0.26 |
| Athene-RM-8B | 64.21 | 23.88 | 9.47 | 68.42 | 27.37 | 40.45 | 0.26 |
| Gemini-1.5-Pro-001 (ArenaHard)[†] | 63.84 | 25.24 | 3.68 | 36.84 | 25.26 | 37.74 | 0.23 |
| GPT-4o-Mini-2024-07-18 (ArenaHard)[†] | 63.83 | 11.48 | 7.89 | 78.95 | 31.58 | 46.47 | 0.24 |
| Gemini-1.5-Pro-002 (ArenaHard)[†] | 63.64 | 15.85 | 11.05 | 36.84 | 32.63 | 46.02 | 0.23 |
| Eurus-RM-7B | 63.16 | 14.36 | 0.53 | 89.47 | 1.05 | 2.86 | 0.33 |
| Llama-3-OffsetBias-RM-8B | 61.05 | 20.44 | 1.58 | 100.00 | 42.11 | 53.68 | 0.21 |
| Gemini-1.5-Flash-002 (ArenaHard)[†] | 60.75 | 16.42 | 8.42 | 57.89 | 12.63 | 17.14 | 0.29 |
| Skywork-Reward-Gemma-2-27B | 60.00 | 30.32 | 0.53 | 89.47 | 22.11 | 31.58 | 0.27 |
| ArmoRM-Llama3-8B-v0.1 | 59.47 | 15.07 | 3.16 | 100.00 | 33.68 | 47.07 | 0.23 |
| InternLM2-1.8B-Reward | 59.47 | 17.02 | 2.63 | 47.37 | 8.42 | 10.53 | 0.32 |
| Nemotron-4-340B-Reward | 58.42 | 10.01 | 6.32 | 89.47 | 20.00 | 29.17 | 0.28 |

Table 19: Reward model and LLM judge performance on Spanish prompt subset of the human preference dataset. LLM-as-a-judge are labeled with system prompt source, and marked with †.

| Reward Model | Accuracy | R.W. Pearson | Separability | Conf. Agree. | Kendalltau | Spearmanr | Brier Score |
|---|---|---|---|---|---|---|---|
| Gemini-1.5-Pro-002 (ArenaHard)[†] | 69.57 | 14.77 | 14.74 | 54.17 | 63.16 | 82.41 | 0.14 |
| GPT-4o-Mini-2024-07-18 (ArenaHard)[†] | 68.45 | 25.12 | 4.21 | 75.00 | 54.74 | 73.08 | 0.17 |
| Ensemble-Judges (ArenaHard)[†] | 68.24 | 21.05 | 17.37 | 66.67 | 62.11 | 80.90 | 0.13 |
| Ensemble-Judges (AlpacaEval)[†] | 67.74 | 27.12 | 4.21 | 79.17 | 46.32 | 65.71 | 0.19 |
| Gemini-1.5-Pro-002 (AlpacaEval)[†] | 67.38 | 26.42 | 8.95 | 79.17 | 47.37 | 65.26 | 0.18 |
| Athene-RM-8B | 67.38 | 26.84 | 18.95 | 45.83 | 45.26 | 64.81 | 0.17 |
| InternLM2-7B-Reward | 66.31 | 20.42 | 11.05 | 45.83 | 43.16 | 62.41 | 0.19 |
| Claude-3-5-Sonnet-20240620 (ArenaHard)[†] | 66.31 | 24.02 | 5.79 | 45.83 | 55.79 | 73.53 | 0.15 |
| Athene-RM-70B | 65.78 | 22.45 | 17.89 | 54.17 | 45.26 | 65.86 | 0.18 |
| InternLM2-20B-Reward | 65.24 | 26.25 | 13.16 | 29.17 | 58.95 | 79.55 | 0.15 |
| ArmoRM-Llama3-8B-v0.1 | 65.24 | 21.41 | 5.79 | 45.83 | 33.68 | 55.19 | 0.23 |
| Llama-3-OffsetBias-RM-8B | 64.71 | 13.13 | 2.11 | 79.17 | 27.37 | 41.80 | 0.23 |
| GPT-4o-2024-08-06 (AlpacaEval)[†] | 64.71 | 20.04 | 4.21 | 58.33 | 52.63 | 72.33 | 0.16 |
| Llama-3.1-70B-Instruct (AlpacaEval)[†] | 64.17 | 20.26 | 3.68 | 70.83 | 43.16 | 61.65 | 0.19 |
| Claude-3-5-Sonnet-20240620 (AlpacaEval)[†] | 63.98 | 27.44 | 2.11 | 79.17 | 36.84 | 51.73 | 0.21 |
| Starling-RM-7B-Alpha | 63.10 | 22.33 | 9.47 | 54.17 | 34.74 | 47.67 | 0.20 |
| GPT-4o-Mini-2024-07-18 (AlpacaEval)[†] | 62.57 | 30.14 | 1.05 | 70.83 | 25.26 | 38.50 | 0.24 |
| GPT-4o-2024-08-06 (ArenaHard)[†] | 62.43 | 15.80 | 8.95 | 70.83 | 49.47 | 65.56 | 0.18 |
| Gemini-1.5-Flash-002 (ArenaHard)[†] | 62.37 | 22.71 | 13.16 | 62.50 | 36.84 | 55.19 | 0.21 |
| Eurus-RM-7B | 62.03 | 14.76 | 8.42 | 37.50 | 17.89 | 26.17 | 0.29 |
| Nemotron-4-340B-Reward | 62.03 | 11.19 | 18.95 | 29.17 | 49.47 | 66.92 | 0.19 |
| Gemini-1.5-Flash-002 (AlpacaEval)[†] | 62.03 | 20.24 | 2.11 | 79.17 | 37.89 | 54.59 | 0.20 |
| Llama-3.1-70B-Instruct (ArenaHard)[†] | 61.62 | 20.93 | 3.68 | 70.83 | 46.32 | 69.17 | 0.17 |
| Gemini-1.5-Pro-001 (ArenaHard)[†] | 61.11 | 12.74 | 5.79 | 58.33 | 47.37 | 59.55 | 0.17 |
| Skywork-Reward-Llama-3.1-8B | 60.96 | 9.19 | 10.53 | 70.83 | 28.42 | 40.00 | 0.26 |
| Starling-RM-34B | 59.36 | 11.68 | 0.53 | 79.17 | 38.95 | 54.44 | 0.22 |
| InternLM2-1.8B-Reward | 58.82 | 21.97 | 4.21 | 12.50 | 36.84 | 46.47 | 0.21 |
| Skywork-Reward-Gemma-2-27B | 57.75 | 3.40 | 8.42 | 87.50 | 48.42 | 63.46 | 0.20 |
| NaiveVerbosityModel | 54.01 | 9.52 | 10.00 | 62.50 | -2.11 | -3.16 | 0.35 |

Table 20: Reward model and LLM judge performance on French prompt subset of the human preference dataset. LLM-as-a-judge are labeled with system prompt source, and marked with †.

| Reward Model | Accuracy | R.W. Pearson | Separability | Conf. Agree. | Kendalltau | Spearmanr | Brier Score |
|---|---|---|---|---|---|---|---|
| GPT-4o-Mini-2024-07-18 (AlpacaEval)[†] | 71.84 | 31.95 | 2.11 | 100.00 | 49.47 | 67.82 | 0.18 |
| Claude-3-5-Sonnet-20240620 (AlpacaEval)[†] | 68.93 | 27.08 | 7.37 | 100.00 | 48.42 | 67.97 | 0.22 |
| InternLM2-7B-Reward | 68.93 | 25.47 | 1.05 | 100.00 | 49.47 | 68.12 | 0.18 |
| Claude-3-5-Sonnet-20240620 (ArenaHard)[†] | 68.63 | 20.55 | 3.68 | 100.00 | 60.00 | 77.74 | 0.18 |
| Ensemble-Judges (AlpacaEval)[†] | 67.96 | 17.35 | 7.37 | 100.00 | 57.89 | 79.25 | 0.22 |
| Ensemble-Judges (ArenaHard)[†] | 67.02 | 20.72 | 10.53 | 100.00 | 62.11 | 76.39 | 0.17 |
| GPT-4o-2024-08-06 (AlpacaEval)[†] | 66.99 | 16.25 | 3.68 | 100.00 | 50.53 | 69.47 | 0.18 |
| Skywork-Reward-Gemma-2-27B | 66.02 | 21.16 | 4.74 | 100.00 | 58.95 | 77.29 | 0.20 |
| Athene-RM-8B | 66.02 | 20.34 | 8.42 | 89.47 | 54.74 | 75.49 | 0.16 |
| Eurus-RM-7B | 65.05 | 26.36 | 3.16 | 78.95 | 30.53 | 39.55 | 0.21 |
| Athene-RM-70B | 65.05 | 10.12 | 7.89 | 89.47 | 50.53 | 72.33 | 0.18 |
| GPT-4o-Mini-2024-07-18 (ArenaHard)[†] | 64.08 | 12.29 | 13.68 | 89.47 | 61.05 | 81.35 | 0.15 |
| Gemini-1.5-Pro-002 (AlpacaEval)[†] | 64.08 | 14.69 | 3.16 | 100.00 | 54.74 | 72.03 | 0.18 |
| Gemini-1.5-Flash-002 (AlpacaEval)[†] | 64.08 | 21.03 | 3.68 | 100.00 | 41.05 | 58.05 | 0.21 |
| Llama-3-OffsetBias-RM-8B | 64.08 | 28.73 | 11.05 | 100.00 | 27.37 | 40.15 | 0.21 |
| InternLM2-20B-Reward | 64.08 | 8.68 | 2.63 | 100.00 | 53.68 | 75.49 | 0.19 |
| Gemini-1.5-Pro-002 (ArenaHard)[†] | 64.00 | 12.53 | 12.63 | 89.47 | 48.42 | 65.56 | 0.19 |
| GPT-4o-2024-08-06 (ArenaHard)[†] | 63.27 | 18.86 | 5.26 | 89.47 | 56.84 | 72.63 | 0.16 |
| Starling-RM-34B | 63.11 | 14.73 | 2.63 | 89.47 | 42.11 | 58.20 | 0.18 |
| Llama-3.1-70B-Instruct (AlpacaEval)[†] | 62.14 | 19.12 | 1.05 | 100.00 | 63.16 | 78.05 | 0.15 |
| Skywork-Reward-Llama-3.1-8B | 62.14 | 25.10 | 6.32 | 100.00 | 36.84 | 54.59 | 0.21 |
| Llama-3.1-70B-Instruct (ArenaHard)[†] | 61.39 | -2.36 | 3.68 | 100.00 | 55.79 | 76.09 | 0.18 |
| ArmoRM-Llama3-8B-v0.1 | 60.19 | 19.66 | 2.11 | 100.00 | 18.95 | 32.18 | 0.25 |
| InternLM2-1.8B-Reward | 59.22 | 11.84 | 2.11 | 57.89 | 27.37 | 33.38 | 0.24 |
| Starling-RM-7B-Alpha | 59.22 | 10.16 | 1.05 | 100.00 | 35.79 | 47.52 | 0.21 |
| NaiveVerbosityModel | 58.25 | 11.49 | 2.63 | 100.00 | 20.00 | 32.78 | 0.22 |
| Nemotron-4-340B-Reward | 58.25 | 7.87 | 3.16 | 100.00 | 40.00 | 55.94 | 0.20 |
| Gemini-1.5-Pro-001 (ArenaHard)[†] | 57.58 | -1.56 | 4.21 | 100.00 | 48.42 | 66.77 | 0.18 |
| Gemini-1.5-Flash-002 (ArenaHard)[†] | 51.96 | -0.90 | 1.05 | 78.95 | 37.89 | 62.11 | 0.19 |

Table 21: Reward model and LLM judge performance on Portuguese prompt subset of the human preference dataset. LLM-as-a-judge are labeled with system prompt source, and marked with †.

| Reward Model | Accuracy | R.W. Pearson | Separability | Conf. Agree. | Kendalltau | Spearmanr | Brier Score |
|---|---|---|---|---|---|---|---|
| Gemini-1.5-Pro-002 (AlpacaEval)[†] | 81.40 | 51.04 | 3.16 | 100.00 | 50.53 | 74.14 | 0.17 |
| Ensemble-Judges (AlpacaEval)[†] | 75.58 | 44.04 | 6.84 | 100.00 | 45.26 | 66.47 | 0.18 |
| Gemini-1.5-Pro-002 (ArenaHard)[†] | 74.42 | 40.23 | 3.16 | 57.89 | 52.63 | 70.83 | 0.18 |
| Athene-RM-70B | 74.42 | 42.65 | 4.74 | 100.00 | 43.16 | 61.05 | 0.20 |
| Claude-3-5-Sonnet-20240620 (ArenaHard)[†] | 73.26 | 42.33 | 1.58 | 100.00 | 47.37 | 58.80 | 0.20 |
| Athene-RM-8B | 73.26 | 43.29 | 8.42 | 78.95 | 43.16 | 60.45 | 0.19 |
| Ensemble-Judges (ArenaHard)[†] | 71.25 | 44.59 | 1.58 | 89.47 | 36.84 | 51.88 | 0.20 |
| Claude-3-5-Sonnet-20240620 (AlpacaEval)[†] | 69.77 | 28.35 | 5.79 | 100.00 | 40.00 | 52.03 | 0.22 |
| Gemini-1.5-Pro-001 (ArenaHard)[†] | 69.23 | 35.18 | 2.63 | 100.00 | 40.00 | 55.94 | 0.19 |
| GPT-4o-2024-08-06 (AlpacaEval)[†] | 68.60 | 39.33 | 5.79 | 100.00 | 40.00 | 53.53 | 0.19 |
| Eurus-RM-7B | 67.44 | 25.34 | 2.63 | 89.47 | -2.11 | -1.95 | 0.29 |
| Skywork-Reward-Llama-3.1-8B | 66.28 | 27.43 | 1.58 | 100.00 | 37.89 | 47.82 | 0.21 |
| ArmoRM-Llama3-8B-v0.1 | 66.28 | 28.46 | 5.79 | 100.00 | 42.11 | 57.14 | 0.19 |
| Gemini-1.5-Flash-002 (AlpacaEval)[†] | 66.28 | 33.17 | 1.05 | 89.47 | 30.53 | 44.81 | 0.22 |
| GPT-4o-2024-08-06 (ArenaHard)[†] | 66.25 | 39.65 | 6.32 | 100.00 | 34.74 | 51.88 | 0.20 |
| GPT-4o-Mini-2024-07-18 (ArenaHard)[†] | 64.71 | 31.59 | 1.05 | 100.00 | 34.74 | 55.64 | 0.20 |
| Llama-3.1-70B-Instruct (ArenaHard)[†] | 64.63 | 27.88 | 1.58 | 89.47 | 38.95 | 54.59 | 0.20 |
| InternLM2-7B-Reward | 63.95 | 26.87 | 3.16 | 36.84 | 12.63 | 15.49 | 0.25 |
| InternLM2-20B-Reward | 63.95 | 19.03 | 0.00 | 100.00 | 29.47 | 46.32 | 0.20 |
| Gemini-1.5-Flash-002 (ArenaHard)[†] | 63.10 | 24.42 | 4.21 | 89.47 | 27.37 | 44.96 | 0.22 |
| Starling-RM-34B | 62.79 | 13.29 | 1.58 | 100.00 | 10.53 | 10.23 | 0.28 |
| Skywork-Reward-Gemma-2-27B | 61.63 | 19.87 | 0.00 | 100.00 | 41.05 | 56.84 | 0.21 |
| Llama-3.1-70B-Instruct (AlpacaEval)[†] | 61.63 | 19.26 | 2.11 | 100.00 | 16.84 | 21.50 | 0.24 |
| Nemotron-4-340B-Reward | 60.47 | 19.10 | 13.16 | 5.26 | 53.68 | 75.34 | 0.18 |
| InternLM2-1.8B-Reward | 59.30 | 16.29 | 0.53 | 89.47 | 2.11 | 0.00 | 0.27 |
| GPT-4o-Mini-2024-07-18 (AlpacaEval)[†] | 58.14 | 14.03 | 1.05 | 100.00 | 24.21 | 33.98 | 0.23 |
| Llama-3-OffsetBias-RM-8B | 58.14 | 2.76 | 1.05 | 100.00 | 45.26 | 61.95 | 0.20 |
| Starling-RM-7B-Alpha | 56.98 | 12.63 | 3.68 | 89.47 | 2.11 | -2.86 | 0.30 |
| NaiveVerbosityModel | 50.00 | -0.20 | 2.63 | 100.00 | -7.37 | -13.68 | 0.31 |

Table 22: Reward model and LLM judge performance on Italian prompt subset of the human preference dataset. LLM-as-a-judge are labeled with system prompt source, and marked with †.

### A.2.1 Score Distribution Statistics of Human Preference Metrics

| subset | mean | std | min | 25% | 50% | 75% | max |
|---|---|---|---|---|---|---|---|
| overall | 0.6341 | 0.0337 | 0.5662 | 0.6100 | 0.6301 | 0.6609 | 0.6859 |
| hard_prompt | 0.6351 | 0.0353 | 0.5621 | 0.6115 | 0.6414 | 0.6630 | 0.6946 |
| easy_prompt | 0.6375 | 0.0412 | 0.5554 | 0.6037 | 0.6410 | 0.6698 | 0.7015 |
| if_prompt | 0.6306 | 0.0369 | 0.5571 | 0.6110 | 0.6304 | 0.6609 | 0.6977 |
| code_prompt | 0.6336 | 0.0316 | 0.5641 | 0.6139 | 0.6418 | 0.6557 | 0.6806 |
| math_prompt | 0.6449 | 0.0483 | 0.5580 | 0.6131 | 0.6388 | 0.6858 | 0.7358 |
| similar_response | 0.6287 | 0.0342 | 0.5592 | 0.6059 | 0.6395 | 0.6545 | 0.6815 |

Table 23: Human Preference V1 Accuracy Metric Statistics

| subset | mean | std | min | 25% | 50% | 75% | max |
|---|---|---|---|---|---|---|---|
| overall | 0.7135 | 0.1133 | 0.3203 | 0.6803 | 0.7477 | 0.7842 | 0.8263 |
| hard_prompt | 0.6623 | 0.0842 | 0.4218 | 0.6303 | 0.6890 | 0.7138 | 0.7637 |
| easy_prompt | 0.5070 | 0.1438 | 0.0761 | 0.4327 | 0.5342 | 0.6105 | 0.7266 |
| if_prompt | 0.6355 | 0.1040 | 0.3583 | 0.5848 | 0.6647 | 0.7011 | 0.7646 |
| is_code | 0.5871 | 0.0857 | 0.3950 | 0.5392 | 0.5971 | 0.6331 | 0.7311 |
| math_prompt | 0.5381 | 0.0876 | 0.3010 | 0.4862 | 0.5668 | 0.6085 | 0.6540 |
| similar_response | 0.6609 | 0.0951 | 0.3456 | 0.6155 | 0.6755 | 0.7270 | 0.7682 |

Table 24: Human Preference V1 Row-wise Pearson Metric Statistics

| subset | mean | std | min | 25% | 50% | 75% | max |
|---|---|---|---|---|---|---|---|
| overall | 0.8244 | 0.1540 | 0.3643 | 0.7571 | 0.8786 | 0.9357 | 0.9714 |
| hard_prompt | 0.7978 | 0.0998 | 0.5000 | 0.7786 | 0.8286 | 0.8643 | 0.9071 |
| easy_prompt | 0.6071 | 0.2054 | 0.0643 | 0.4500 | 0.6571 | 0.7571 | 0.8500 |
| if_prompt | 0.7759 | 0.1362 | 0.4571 | 0.6929 | 0.8357 | 0.8714 | 0.9214 |
| is_code | 0.7355 | 0.0993 | 0.5143 | 0.6929 | 0.7500 | 0.8071 | 0.8571 |
| math_prompt | 0.6527 | 0.1360 | 0.3000 | 0.6143 | 0.6929 | 0.7571 | 0.8071 |
| similar_response | 0.7798 | 0.1296 | 0.3500 | 0.7429 | 0.7929 | 0.8643 | 0.9214 |

Table 25: Human Preference V1 Confidence Agreement Metric Statistics

| subset | mean | std | min | 25% | 50% | 75% | max |
|---|---|---|---|---|---|---|---|
| overall | 82.1779 | 3.9903 | 73.1580 | 80.0000 | 82.6320 | 84.2110 | 91.5790 |
| hard_prompt | 73.3031 | 5.7412 | 51.0530 | 71.5790 | 73.6840 | 76.8420 | 81.5790 |
| easy_prompt | 55.7350 | 8.2493 | 34.7370 | 50.5260 | 55.2630 | 61.5790 | 67.8950 |
| if_prompt | 68.8929 | 7.6624 | 45.7890 | 67.3680 | 69.4740 | 74.7370 | 80.0000 |
| is_code | 66.8239 | 7.0225 | 39.4740 | 65.2630 | 68.4210 | 70.0000 | 76.8420 |
| math_prompt | 61.1070 | 8.9433 | 27.8950 | 58.4210 | 62.6320 | 66.3160 | 72.1050 |
| similar_response | 76.5153 | 4.8143 | 64.7370 | 74.2110 | 77.3680 | 78.9470 | 83.6840 |

Table 26: Human Preference V1 Separability Metric Statistics

| subset | mean | std | min | 25% | 50% | 75% | max |
|---|---|---|---|---|---|---|---|
| overall | 0.8432 | 0.1473 | 0.3353 | 0.8120 | 0.8872 | 0.9444 | 0.9699 |
| hard_prompt | 0.8637 | 0.0911 | 0.5955 | 0.8541 | 0.8812 | 0.9218 | 0.9474 |
| easy_prompt | 0.7673 | 0.2210 | 0.0737 | 0.6421 | 0.8451 | 0.9128 | 0.9624 |
| if_prompt | 0.8709 | 0.1176 | 0.5504 | 0.8662 | 0.9113 | 0.9429 | 0.9699 |
| is_code | 0.8405 | 0.0873 | 0.6015 | 0.8331 | 0.8752 | 0.8902 | 0.9429 |
| math_prompt | 0.8096 | 0.1062 | 0.3895 | 0.8075 | 0.8316 | 0.8737 | 0.9203 |
| similar_response | 0.8299 | 0.1208 | 0.4195 | 0.8015 | 0.8586 | 0.9098 | 0.9489 |

Table 27: Human Preference V1 Spearman Metric Statistics

| subset | mean | std | min | 25% | 50% | 75% | max |
|---|---|---|---|---|---|---|---|
| overall | 0.7172 | 0.1593 | 0.2947 | 0.6737 | 0.7579 | 0.8421 | 0.9053 |
| hard_prompt | 0.7227 | 0.1083 | 0.4211 | 0.6737 | 0.7474 | 0.7895 | 0.8421 |
| easy_prompt | 0.6203 | 0.2009 | 0.0737 | 0.4737 | 0.6737 | 0.7474 | 0.8632 |
| if_prompt | 0.7397 | 0.1338 | 0.3895 | 0.7158 | 0.7895 | 0.8316 | 0.8737 |
| is_code | 0.6897 | 0.0939 | 0.4632 | 0.6737 | 0.7053 | 0.7579 | 0.8211 |
| math_prompt | 0.6570 | 0.1080 | 0.3053 | 0.6421 | 0.6737 | 0.7158 | 0.8000 |
| similar_response | 0.6860 | 0.1345 | 0.3053 | 0.6211 | 0.7053 | 0.7789 | 0.8421 |

Table 28: Human Preference V1 Kendalltau Metric Statistics

| subset | mean | std | min | 25% | 50% | 75% | max |
|---|---|---|---|---|---|---|---|
| overall | 0.1276 | 0.0750 | 0.0454 | 0.0700 | 0.1055 | 0.1532 | 0.3344 |
| hard_prompt | 0.1169 | 0.0481 | 0.0680 | 0.0857 | 0.1047 | 0.1353 | 0.2507 |
| easy_prompt | 0.1549 | 0.0906 | 0.0562 | 0.0946 | 0.1201 | 0.2273 | 0.4132 |
| if_prompt | 0.1062 | 0.0591 | 0.0442 | 0.0685 | 0.0851 | 0.1148 | 0.2579 |
| is_code | 0.1289 | 0.0447 | 0.0721 | 0.0986 | 0.1176 | 0.1396 | 0.2421 |
| math_prompt | 0.1408 | 0.0507 | 0.0857 | 0.1064 | 0.1285 | 0.1473 | 0.3191 |
| similar_response | 0.1378 | 0.0628 | 0.0705 | 0.0934 | 0.1259 | 0.1577 | 0.3299 |

Table 29: Human Preference V1 Brier Metric Statistics

## A.3 Details on Curation and Scores for Correctness Preference Evaluation Dataset

### A.3.1 Small Benchmark Modifications

To ensure more natural responses that better reflect real-world use cases, we modified each verifiable benchmark's canonical prompt to encourage Chain of Thought (CoT) thinking (citation). This approach both increases the diversity of sampled responses and enhances the task difficulty for the human preference proxy by incorporating additional signals beyond final answer correctness. The specific instructions for each benchmark are detailed below.

For the MATH benchmark, we implemented a new system prompt to facilitate zero-shot CoT behavior. Additionally, we converted the parsed answer to its symbolic representation and utilized a symbolic solver to evaluate true equality instead of relying on raw string matching. This refinement of the correctness signal ensures that trivial answer differences, such as $1\frac{3}{4}$ vs $\frac{7}{4}$ or $\frac{4i+\sqrt{5}}{2}$ vs $\frac{\sqrt{5}}{2}+2i$, are marked as equivalent, with either answer accepted if correct.

In practice, we observed that the sampled MBPP-Plus generations from some models were almost all identical. Models also generally failed to follow instructions to "think step-by-step" before providing their final answers, suppressing answer diversity. To address this issue, we prompted the models to "write comments clearly explaining each part of the code," thereby lengthening trajectories and yielding greater exploration of the answer spaces. We also observed some ambiguity in MBPP-Plus intructions. To mitigate this, we added standard MBPP test cases into the function docstring as examples, and used the more extensive remaining MBPP-Plus test cases as the real tests.

Lastly, for IFEval, we prefixed the prompts with "It is extremely important that you follow all instructions exactly." This addition emphasizes the necessity of precise instruction following in these tasks and ensures that the human preference proxy implicitly recognizes this as a significant evaluation criterion.

The prompt template for MMLU-Pro and GPQA were adaption from Gao et al. (2021)'s Language Model Evaluation Harness. The MATH template was generated with the assistance of Anthropic's prompt generator.

The prompt templates for each benchmark are detailed below. Note that {{var}} indicates a field to be filled by prompt data or metadata.

---

**MMLU Prompt Template:**

```
The following are multiple choice questions (with answers) about {{domain}}.  Think step
by step and then finish your answer with "the answer is (X)" where X is the correct
letter choice.

Question: {{question}}
Options:
{{letter}}. {{choice}}
{{letter}}. {{choice}}
{{letter}}. {{choice}}
...
```

**MATH Prompt Template:**

```
You are a highly skilled mathematician tasked with solving complex math problems.
Your goal is to provide clear, step-by-step solutions that can be easily parsed and
evaluated.

Here is the math problem you need to solve:

<problem>
{{MATH_PROBLEM}}
</problem>

Box your final answer using LaTeX, for example: $x = \\boxed{[Your final numerical or
algebraic answer]}$.

Now, please solve the given math problem and provide your solution in the specified format.
```

**GPQA Prompt Template:**

```
The following is a {{domain}} multiple choice question.  Think step by step and then
finish your answer with "the answer is (X)" where X is the correct letter choice.

Question: {{question}}
Choices:
(A) {{choice1}}
(B) {{choice2}}
(C) {{choice3}}
(D) {{choice4}}
```

**MBPP-Plus Prompt Template:**

```
Below will be an instruction to write a python function that accomplishes a task.
You will also be given starter code with a function definition and any required imports.
Think step-by-step, write comments clearly explaining each part of the code, and make sure
your code solution is enclosed in markdown ticks (``` [your code here] ```).

<instruction>
{{instruction}}
</instruction>

<starter_code>
```
{{starter_code}}
    pass
```
</starter_code>
```

**IFEval Prompt Template:**

```
It is extemely important that you follow all instructions exactly:
{{prompt}}
```

## A.3.2   MORE ON BEST OF K CURVES

These curves represent how much the reward model can differentiate the LLM's generations whilst picking from examples drawn from the same distribution. The simple intuition here is that as K increases, the "exploration" of the LLM is expanded, thereby increasing the likelihood that a correct

answer lies within the K different samples. However, as exploration increases, the likelihood that a response that exploits the reward model is present also increases. In all best of K metrics, we use $K = 32$, providing both reasonable inference costs balanced with a significant enough exploration space to test the reward model's capabilities.

In order to distill the curves into interpretable numbers, we propose several metrics:

1. **Maximum Achieved Performance:** the maximum score achieved by the reward model at any point on the best of K curve. Note that the maximum achieved performance is relatively agnostic to over-optimization.

2. **Error With Respect to Ground Truth:** the expected squared error between the score of the reward model's selected response against the ground truth best response. Once again, let $S_K$ be a size $K$ random sample of responses from a model, $g : S_K \to \{0, 1\}$ be the ground truth scoring function, and $\hat{R} : S_K \to \mathbb{R}$ be the reward model proxy score. Then, the error with respect to ground truth is $\frac{1}{32} \sum_{K=1}^{32} \mathbb{E}_{S_K}[(g(\arg\max_{s \in S_K} \hat{R}(s)) - \max_{s \in S_K} g(s))^2]$

3. **End Score:** We also look at the final score achieved by the reward model at $K = 32$. If no over-optimization has occurred this should also be the maximum achieved performance.

### A.3.3 DETAILED SCORES

| Reward Model | MMLU Pro | Math | GPQA | MBPP Plus | IF Eval | Mean |
|---|---|---|---|---|---|---|
| Athene-RM-70B | 0.761 | 0.607 | 0.499 | 0.748 | 0.633 | 0.650 |
| InternLM2-20B-Reward | 0.673 | 0.538 | 0.471 | 0.654 | 0.652 | 0.598 |
| Llama-3-Offsetbias-RM-8B | 0.590 | 0.481 | 0.450 | 0.819 | 0.646 | 0.597 |
| Athene-RM-8B | 0.656 | 0.517 | 0.459 | 0.675 | 0.586 | 0.579 |
| Nemotron-4-340B-Reward | 0.697 | 0.499 | 0.484 | 0.567 | 0.623 | 0.574 |
| InternLm2-7B-Reward | 0.638 | 0.552 | 0.457 | 0.562 | 0.658 | 0.573 |
| ArmoRM-Llama3-8B-v0.1 | 0.654 | 0.508 | 0.470 | 0.602 | 0.601 | 0.567 |
| Skywork-Reward-Llama-3.1-8B | 0.641 | 0.500 | 0.468 | 0.581 | 0.639 | 0.566 |
| Starling-RM-34B | 0.651 | 0.476 | 0.453 | 0.634 | 0.569 | 0.557 |
| Eurus-RM-7B | 0.607 | 0.516 | 0.438 | 0.590 | 0.594 | 0.549 |
| Skywork-Reward-Gemma-2-27B | 0.550 | 0.462 | 0.447 | 0.691 | 0.583 | 0.547 |
| InternLM2-1-8B-Reward | 0.538 | 0.411 | 0.451 | 0.572 | 0.581 | 0.510 |
| Starling-RM-7B-Alpha | 0.562 | 0.409 | 0.433 | 0.559 | 0.564 | 0.505 |
| NaiveVerbosityModel | 0.487 | 0.349 | 0.420 | 0.568 | 0.539 | 0.473 |

Table 30: Reward Model Best of K Performance Across Benchmarks

| Reward Model | MMLU Pro | Math | GPQA | MBPP Plus | IF Eval | Mean |
|---|---|---|---|---|---|---|
| Athene-RM-70B | 0.792 | 0.760 | 0.603 | 0.661 | 0.594 | 0.682 |
| Internlm2-20B-reward | 0.677 | 0.691 | 0.562 | 0.574 | 0.595 | 0.620 |
| Llama-3-offsetbias-RM-8B | 0.631 | 0.617 | 0.541 | 0.710 | 0.594 | 0.619 |
| Athene-RM-8B | 0.683 | 0.673 | 0.560 | 0.602 | 0.556 | 0.615 |
| Nemotron-4-340B-Reward | 0.704 | 0.660 | 0.570 | 0.506 | 0.587 | 0.605 |
| Skywork-Reward-Llama-3.1-8B | 0.663 | 0.678 | 0.560 | 0.523 | 0.586 | 0.602 |
| Internlm2-7B-Reward | 0.665 | 0.718 | 0.558 | 0.464 | 0.605 | 0.602 |
| ArmoRM-Llama3-8B-v0.1 | 0.678 | 0.659 | 0.549 | 0.538 | 0.573 | 0.599 |
| Starling-RM-34B | 0.683 | 0.621 | 0.547 | 0.534 | 0.536 | 0.584 |
| Eurus-RM-7B | 0.627 | 0.665 | 0.521 | 0.537 | 0.554 | 0.581 |
| Skywork-Reward-Gemma-2-27B | 0.542 | 0.582 | 0.506 | 0.572 | 0.536 | 0.547 |
| Internlm2-1-8B-Reward | 0.561 | 0.587 | 0.538 | 0.462 | 0.538 | 0.537 |
| Starling-RM-7B-Alpha | 0.547 | 0.527 | 0.506 | 0.400 | 0.519 | 0.500 |
| NaiveVerbosityModel | 0.495 | 0.528 | 0.506 | 0.330 | 0.511 | 0.474 |

Table 31: Area Under ROC Curve for Reward Models across Benchmarks

| Reward Model | gemma-2-9b-it | | | gpt-4o-mini | | | Llama-3-8B | | | claude-3-haiku | | |
|---|---|---|---|---|---|---|---|---|---|---|---|---|
| | Loss | Max | End | Loss | Max | End | Loss | Max | End | Loss | Max | End |
| athene-rm-70b | 0.093 | 0.702 | 0.681 | 0.110 | 0.678 | 0.629 | 0.113 | 0.669 | 0.653 | 0.131 | 0.633 | 0.605 |
| armorm-llama3-8b-v0.1 | 0.119 | 0.657 | 0.636 | 0.147 | 0.620 | 0.580 | 0.179 | 0.576 | 0.537 | 0.194 | 0.564 | 0.512 |
| naiveverbositymodel | 0.241 | 0.508 | 0.463 | 0.250 | 0.554 | 0.425 | 0.358 | 0.448 | 0.317 | 0.337 | 0.467 | 0.355 |
| eurus-rm-7b | 0.143 | 0.627 | 0.597 | 0.158 | 0.613 | 0.562 | 0.187 | 0.562 | 0.512 | 0.228 | 0.531 | 0.452 |
| skywork-reward-gemma-2-27b | 0.169 | 0.583 | 0.543 | 0.175 | 0.590 | 0.549 | 0.209 | 0.534 | 0.494 | 0.190 | 0.558 | 0.529 |
| skywork-reward-llama-3.1-8b | 0.126 | 0.643 | 0.612 | 0.136 | 0.633 | 0.597 | 0.189 | 0.565 | 0.527 | 0.216 | 0.561 | 0.491 |
| llama-3-offsetbias-rm-8b | 0.133 | 0.653 | 0.629 | 0.146 | 0.629 | 0.585 | 0.210 | 0.542 | 0.502 | 0.151 | 0.620 | 0.592 |
| nemotron-4-340b-reward | 0.129 | 0.641 | 0.617 | 0.128 | 0.644 | 0.618 | 0.159 | 0.610 | 0.583 | 0.232 | 0.565 | 0.485 |
| starling-rm-34b | 0.157 | 0.602 | 0.570 | 0.151 | 0.622 | 0.563 | 0.183 | 0.562 | 0.528 | 0.209 | 0.545 | 0.487 |
| athene-rm-8b | 0.142 | 0.621 | 0.584 | 0.133 | 0.636 | 0.600 | 0.175 | 0.589 | 0.543 | 0.183 | 0.560 | 0.531 |
| internlm2-7b-reward | 0.138 | 0.630 | 0.588 | 0.147 | 0.633 | 0.581 | 0.155 | 0.608 | 0.581 | 0.253 | 0.565 | 0.462 |
| starling-rm-7b-alpha | 0.183 | 0.569 | 0.535 | 0.199 | 0.578 | 0.516 | 0.238 | 0.508 | 0.476 | 0.319 | 0.486 | 0.378 |
| internlm2-1-8b-reward | 0.193 | 0.566 | 0.501 | 0.191 | 0.583 | 0.506 | 0.218 | 0.526 | 0.480 | 0.256 | 0.503 | 0.448 |
| internlm2-20b-reward | 0.124 | 0.648 | 0.626 | 0.130 | 0.646 | 0.607 | 0.159 | 0.602 | 0.570 | 0.166 | 0.586 | 0.570 |

Table 32: Average Best of K per Sample Model across MMLU Pro, Math, GPQA, MBPP Plus, and IF Eval

| Reward Model | gemma-2-9b-it | gpt-4o-mini | Llama-3-8B | claude-3-haiku |
|---|---|---|---|---|
| athene-rm-70b | 0.710 | 0.648 | 0.710 | 0.674 |
| armorm-llama3-8b-v0.1 | 0.655 | 0.577 | 0.616 | 0.591 |
| naiveverbositymodel | 0.515 | 0.491 | 0.487 | 0.433 |
| eurus-rm-7b | 0.620 | 0.546 | 0.621 | 0.562 |
| skywork-reward-gemma-2-27b | 0.553 | 0.519 | 0.562 | 0.550 |
| skywork-reward-llama-3.1-8b | 0.639 | 0.594 | 0.619 | 0.578 |
| llama-3-offsetbias-rm-8b | 0.628 | 0.574 | 0.583 | 0.650 |
| nemotron-4-340b-reward | 0.639 | 0.586 | 0.658 | 0.561 |
| starling-rm-34b | 0.602 | 0.571 | 0.604 | 0.574 |
| athene-rm-8b | 0.640 | 0.592 | 0.635 | 0.601 |
| internlm2-7b-reward | 0.657 | 0.573 | 0.655 | 0.569 |
| starling-rm-7b-alpha | 0.544 | 0.499 | 0.525 | 0.475 |
| internlm2-1-8b-reward | 0.581 | 0.536 | 0.570 | 0.504 |
| internlm2-20b-reward | 0.629 | 0.603 | 0.650 | 0.603 |

Table 33: Average AUC per sample model across MMLU Pro, Math, GPQA, MBPP Plus, and IF Eval

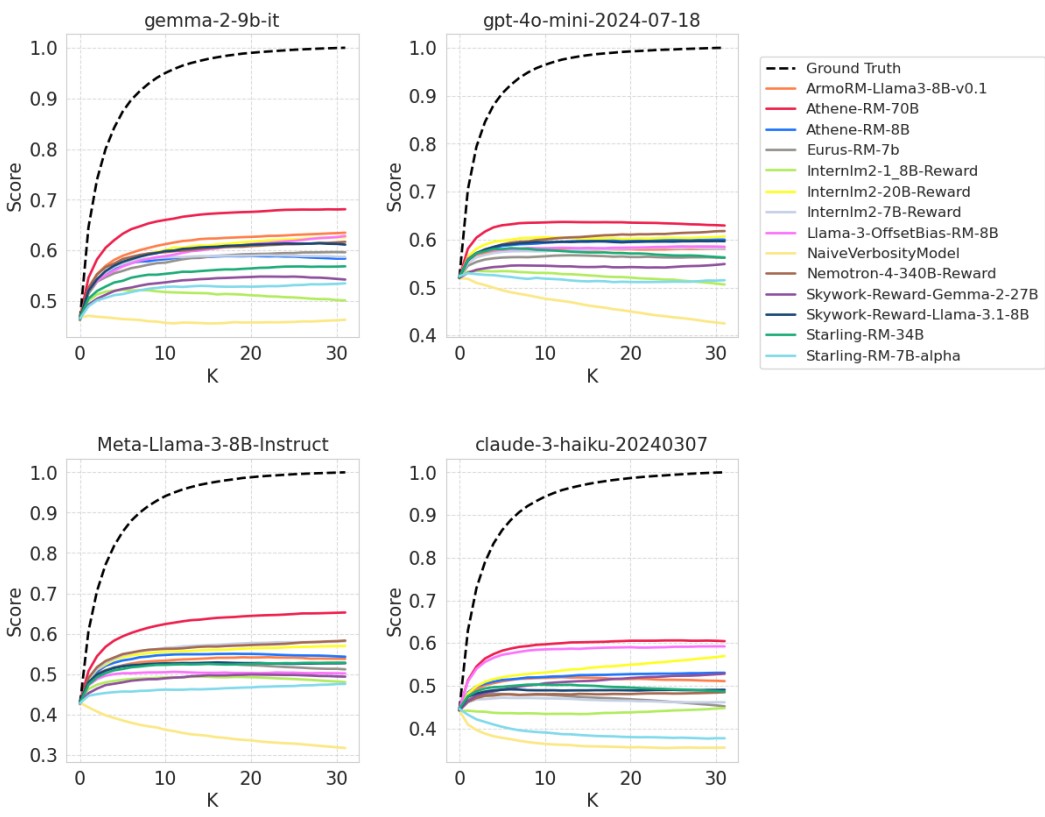

Figure 8: Performance average across all benchmarks, conditioned on each sample model

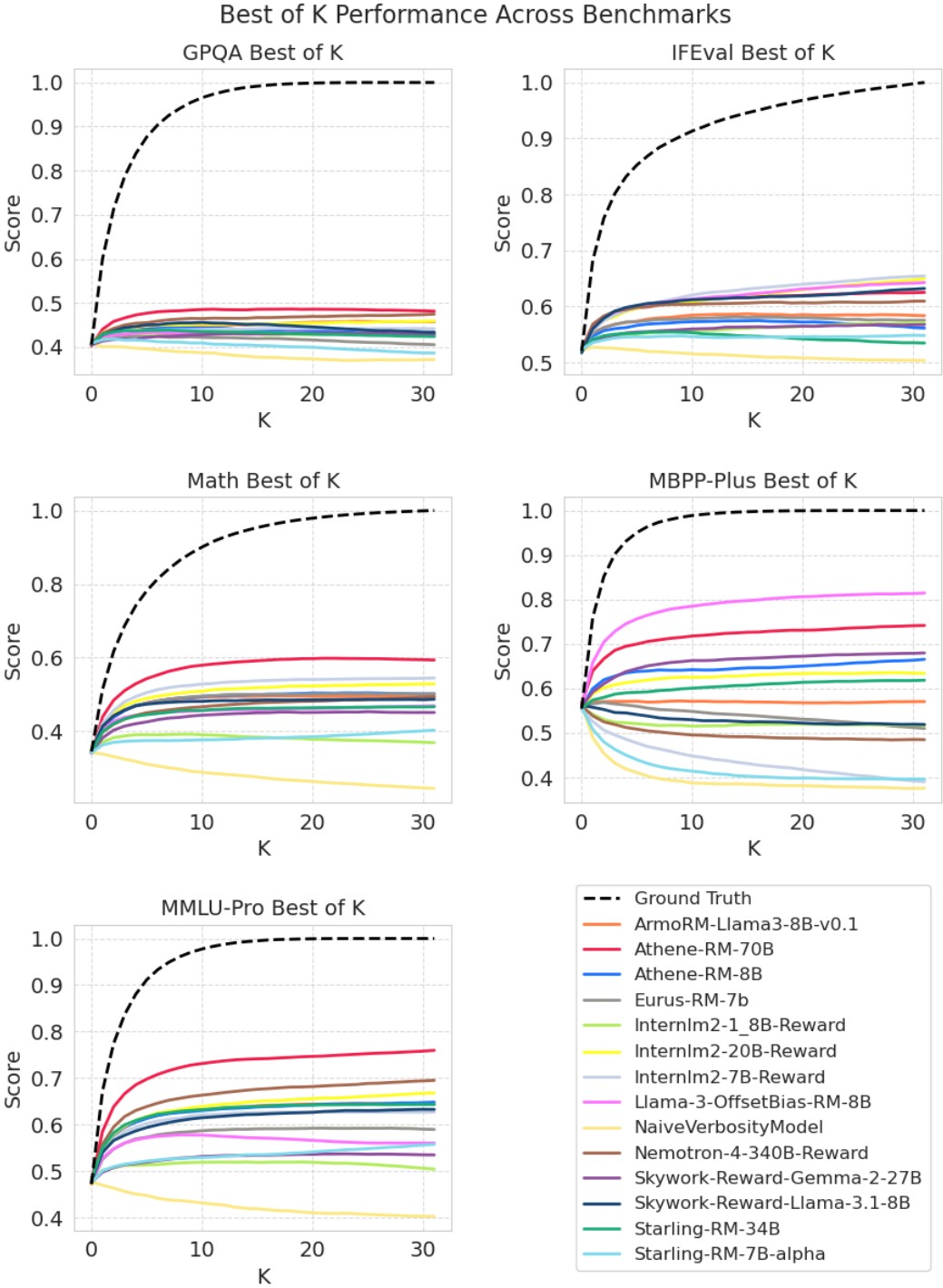

Figure 9: Performance comparison across all benchmarks

A.3.4 SCORE DISTRIBUTION STATISTICS OF CORRECTNESS BENCHMARKS

|  | mean | std | min | 25% | 50% | 75% | max |
|---|---|---|---|---|---|---|---|
| accuracy | 0.557 | 0.031 | 0.477 | 0.544 | 0.561 | 0.570 | 0.632 |
| area_under_curve | 0.545 | 0.028 | 0.506 | 0.525 | 0.548 | 0.560 | 0.603 |
| loss | 0.265 | 0.026 | 0.219 | 0.251 | 0.265 | 0.270 | 0.322 |
| mean_max_score | 0.458 | 0.020 | 0.424 | 0.449 | 0.455 | 0.469 | 0.498 |
| mean_end_score | 0.432 | 0.031 | 0.372 | 0.423 | 0.431 | 0.453 | 0.481 |

Table 34: GPQA Benchmark Score Distribution Information

|  | mean | std | min | 25% | 50% | 75% | max |
|---|---|---|---|---|---|---|---|
| accuracy | 0.581 | 0.035 | 0.517 | 0.560 | 0.576 | 0.617 | 0.640 |
| area_under_curve | 0.563 | 0.031 | 0.511 | 0.536 | 0.565 | 0.593 | 0.605 |
| loss | 0.121 | 0.025 | 0.090 | 0.097 | 0.122 | 0.135 | 0.173 |
| mean_max_score | 0.605 | 0.037 | 0.540 | 0.581 | 0.599 | 0.638 | 0.658 |
| mean_end_score | 0.590 | 0.047 | 0.503 | 0.563 | 0.579 | 0.631 | 0.654 |

Table 35: IFEVAL Benchmark Score Distribution Information

|  | mean | std | min | 25% | 50% | 75% | max |
|---|---|---|---|---|---|---|---|
| accuracy | 0.693 | 0.091 | 0.498 | 0.645 | 0.693 | 0.726 | 0.866 |
| area_under_curve | 0.656 | 0.089 | 0.527 | 0.602 | 0.660 | 0.684 | 0.878 |
| loss | 0.199 | 0.080 | 0.047 | 0.169 | 0.189 | 0.214 | 0.401 |
| mean_max_score | 0.504 | 0.091 | 0.348 | 0.470 | 0.500 | 0.527 | 0.741 |
| mean_end_score | 0.486 | 0.107 | 0.245 | 0.459 | 0.494 | 0.516 | 0.736 |

Table 36: Math Benchmark Score Distribution Information

|  | mean | std | min | 25% | 50% | 75% | max |
|---|---|---|---|---|---|---|---|
| accuracy | 0.533 | 0.095 | 0.312 | 0.510 | 0.538 | 0.580 | 0.743 |
| area_under_curve | 0.530 | 0.098 | 0.330 | 0.474 | 0.536 | 0.573 | 0.710 |
| loss | 0.177 | 0.092 | 0.035 | 0.110 | 0.176 | 0.221 | 0.337 |
| mean_max_score | 0.631 | 0.078 | 0.557 | 0.577 | 0.596 | 0.668 | 0.818 |
| mean_end_score | 0.565 | 0.134 | 0.376 | 0.491 | 0.544 | 0.658 | 0.815 |

Table 37: MBPP Plus Benchmark Score Distribution Information

|  | mean | std | min | 25% | 50% | 75% | max |
|---|---|---|---|---|---|---|---|
| accuracy | 0.654 | 0.078 | 0.479 | 0.615 | 0.662 | 0.684 | 0.814 |
| area_under_curve | 0.639 | 0.079 | 0.495 | 0.578 | 0.664 | 0.682 | 0.792 |
| loss | 0.139 | 0.059 | 0.053 | 0.109 | 0.118 | 0.172 | 0.291 |
| mean_max_score | 0.622 | 0.073 | 0.483 | 0.570 | 0.640 | 0.655 | 0.762 |
| mean_end_score | 0.605 | 0.089 | 0.403 | 0.559 | 0.630 | 0.647 | 0.760 |

Table 38: MMLU Pro Benchmark Score Distribution Information

## A.4 DPO CONFIGURATION

| DPO Configuration | |
| --- | --- |
| Base Model | Meta-Llama-3.1-8B-Instruct |
| $\tau$ | 0.1 |
| Learning Rate | $2.00 \times 10^{-0.6}$ |
| LR Schedule | Constant |
| Global Batch Size | 64 |
| Max Length | 8192 |
| Max Prompt Length | 4096 |
| Implementation | TRL DPOTrainer (von Werra et al., 2020) |
| Optimizer | AdamW, $\beta_1 = 0.9$, $\beta_2 = 0.999$ |
| Space Optimization | Deepspeed Zero2 |

## A.5 CROWDSOURCED HUMAN PREFERENCE VOTE DETAILS

| #Votes | Est. Unique Users | Mean Votes/User | Median Votes/User | Mean Battles/Pair | Mean Votes/Model |
| --- | --- | --- | --- | --- | --- |
| 12190 | 6120 | 1.99 | 1.00 | 190.47 | 2031.67 |

Table 39: Statistics on vote participation and distribution for crowdsourced human preference labels.

## A.6 ADDITIONAL ANALYSIS ON DOWNSTREAM PERFORMANCE

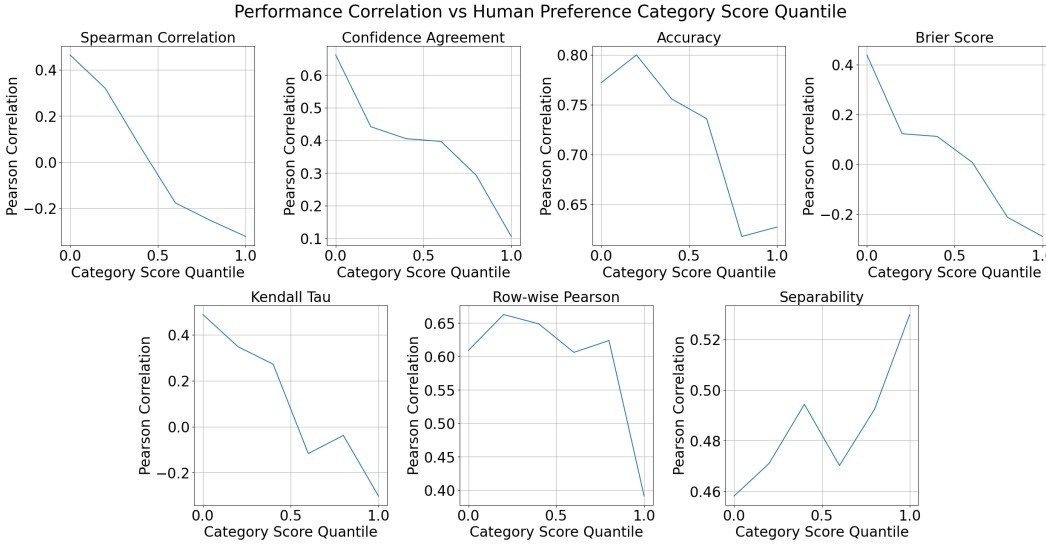

Figure 10: The graphs show all metrics for the human preference dataset. For each metric, the six benchmarks (Hard, Easy, Instruction Following, Coding, Math, and Similar Responses Prompts) (all mean and SD normalized) aggregated into final score by quantile (x-axis). The Pearson Correlation between the aggregated scores are calculated relative to Post-RLHF Human Preference ratings for each aggregation level. Notice that for all metrics except Separability, decreasing quantile increases correlation.

One possible cause of the pattern seen in Figure 10 is that low quantile aggregation better measures robustness. Intuitively, any single weakness within some input domain could be exploited by the policy model during RL training, thus damaging the model. Another reasonable explanation is that a reward model's weakness in one area may yield noisy signals during training, causing the policy model's rather fragile parameters to be disrupted— a possibly unrecoverable degradation in what

we may consider an instance of "catastrophic forgetting". Ultimately, the underlying mechanisms are complex; we do not expect to answer this question in its entirety. However, we believe that our end-to-end experiment provides the first step to understanding how reward model behaviors relate to downstream performance.

### A.6.1 COMMENTS ON REWARDBENCH CORRELATIONS

Commenting on Figure 6; while our work's focus was not to prove or disprove RewardBench, we can provide the following hypothesis for context and clarity: we hypothesize that the reward models tested may have over-optimized for RewardBench's specific preference distribution rather than capturing broader human preferences, potentially exceeding RewardBench's measurement capabilities. However, we note that initial improvements in RewardBench score may still correlate well to real post RLHF human preference outcomes. Ultimately, these insights are only possible through our end-to-end experiments, which enable the research community to further investigate and discuss the true correlations between benchmark metrics and downstream performance. We believe this highlights the value of comprehensive evaluation approaches like ours in understanding real-world model behaviors.

### A.6.2 STYLE-CONTROLLED DOWNSTREAM PERFORMANCE

| Model | Elo | 95% CI Lower | 95% CI Upper |
|---|---|---|---|
| Meta-Llama-3.1-70B-Instruct[*] | 1229 | 1218 | 1239 |
| Athene-RM-70B | **1209** | 1201 | 1218 |
| Athene-RM-8B | 1203 | 1194 | 1211 |
| internlm2-7b-reward | 1201 | 1192 | 1210 |
| Llama-3-OffsetBias-RM-8B | 1197 | 1188 | 1204 |
| ArmoRM-Llama3-8B-v0.1 | 1185 | 1175 | 1191 |
| Meta-Llama-3.1-8B-Instruct[*] | 1177 | 1168 | 1186 |
| Skywork-Reward-Llama-3.1-8B | 1171 | 1163 | 1182 |
| Nemotron-4-340B-Reward | 1170 | 1161 | 1180 |
| internlm2-20b-reward | 1170 | 1159 | 1179 |
| Skywork-Reward-Gemma-2-27B | 1170 | 1160 | 1180 |
| Meta-Llama-3-8B-Instruct[*] | 1152 | 1142 | 1160 |

Table 40: Post DPO performance on real human preference Overall Category after applying style-control. "Model" is the reward model used to train the base model. Models marked with "*" are baseline unaltered models. The best non-base model elo is bolded.

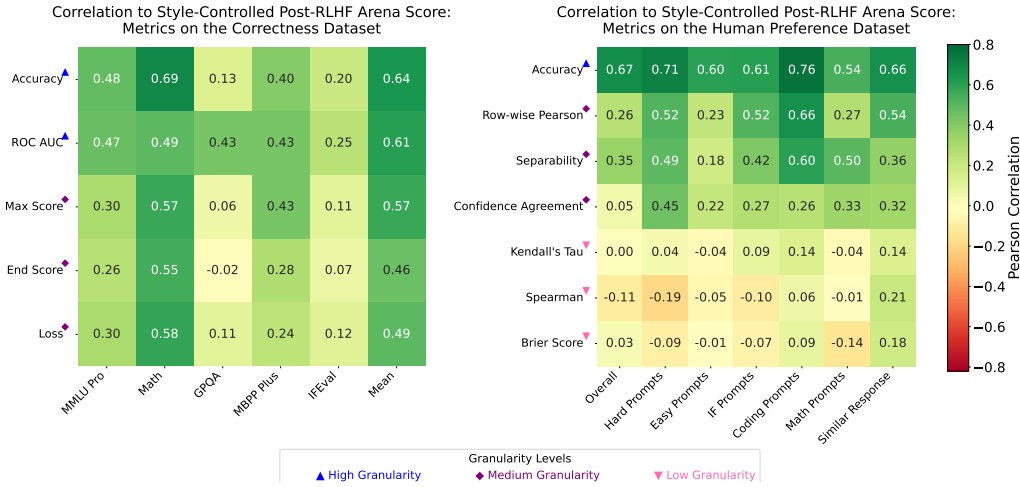

Figure 11: Pearson correlations between various metrics and styled-controlled human preference scores. Left: Correlations between metrics on the Correctness Dataset and Post-RLHF human preference rating. Right: Correlations between metrics on the Human Preference Dataset and Post-RLHF human preference rating.

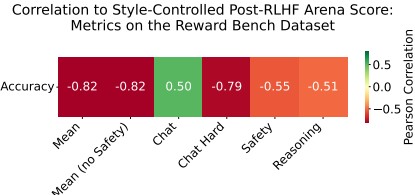

Figure 12: Pearson correlation between the ranking of models in RewardBench and their respective style-controlled Post-DPO rankings on real human preference.

As an ablation, we calculate style-controlled human preference ratings. Style-controlled ratings fit the Bradley Terry model with style elements as features of the regression. These features are used to decouple style from model ratings; this process yields score estimates, style *aside*. The full process for style control is detailed in Li et al. (2024a). For maximum coverage, we control for length and markdown.

### A.6.3 Correlation vs. K

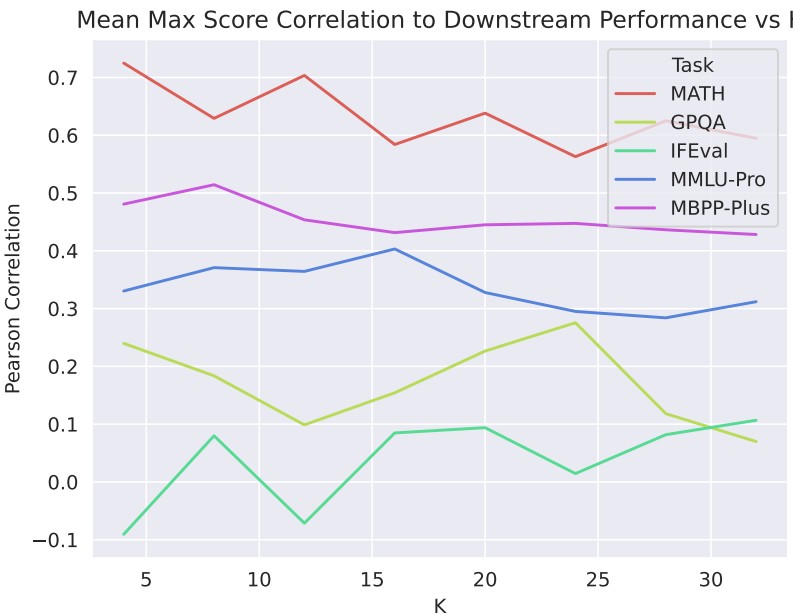

Figure 13: Pearson correlation to downstream human preference performance of mean max score best of $K$ metric vs $K$.

Figure 13 shows that increasing the value of $K$ for best of K metrics does not increase benchmark predictive power. We note that the most predictive correctness metrics is the accuracy metric detailed in subsubsection 5.2.3 which is inherently $K = 2$. Therefore, the predictive power of PPE can be retained without running full $K = 32$, which is more compute heavy.

### A.7 Recommendations for PPE and Future Reward Model Benchmarks

Based on this end-to-end study results detailed in section 7 and Appendix Figure 13, we recommend those seeking the most predictive power from PPE run the human preference set as well as the MATH accuracy metric. We suggest that users pay particular attention to the lower bound accuracy across the main human preference set categories (easy, hard, instruction following, coding, math, and similar). Considering our findings, this configuration likely maintains full predictive power of PPE with less than half of the runtime. Future reward benchmarks may find it helpful to attend to these particular design patterns.

### A.8 Runtimes and Costs for PPE

| Benchmark Set | Time | Cost |
|---|---|---|
| Optimized (Human Preference V1 + Math Accuracy) | < 42 minutes | < $1.50 |
| Full Benchmark | < 120 minutes | < $3.50 |
| End-to-end RLHF pipeline | > 1 week | $1000 or more |

Table 41: Benchmark runtimes and costs. Costs are calculated from RunPod's hourly GPU pricing, which puts an NVIDIA A100 80GB PCIe instance at $1.64 per hour. Costs could fluctuate between GPU providers. Runtimes are estimated assuming an 8B reward model.

