# OpenReview forum: "How to Evaluate Reward Models for RLHF"
_ICLR.cc/2025/Conference — ICLR 2025 Poster_

### Official Review · Reviewer_xNaA · 2024-10-24

**Soundness:** 3
**Presentation:** 3
**Contribution:** 3
**Rating:** 8
**Confidence:** 2

**Summary:**

This paper presents PPE, a reward model benchmark explicitly tied to post-RLHF outcomes based on real human preferences. Previous reward model benchmarks focused solely on evaluating the reward models themselves, without assessing their post-RLHF performance. PPE fills this gap. More importantly, the paper analyzes which metrics best reflect post-RLHF performance, providing valuable insights for improving reward model evaluation methods.

**Strengths:**

- Compared to previous reward model benchmarks, PPE directly evaluates the post-RLHF performance of models, which more accurately reflects the positive impact of reward models on LLM tuning.
- PPE offers more diverse and larger-scale datasets than earlier benchmarks, making it a highly robust benchmark.
- The paper is very detailed and easy to follow, allowing readers to clearly understand the evaluation process. The explanations for each metric and evaluation method are convincing.
- I personally appreciate Section 7 of the paper. The analyses can help researchers predict the performance of fine-tuned LLMs using high-confidence metrics when they lack the resources to conduct full post-RLHF evaluations.

**Weaknesses:**

- I believe this work fills a significant gap in reward model evaluation by addressing post-RLHF performance testing. The effort is substantial and thorough. I do not observe any obvious weaknesses.

**Questions:**

N/A

---

> ### Author Response · Authors · 2024-11-19
> **Official Response to Reviewer**
>
> We thank the reviewer for their comments. We appreciate that the reviewer recognizes the importance of PPE, especially our in-depth analysis of downstream performance. Please let us know if there are any questions.

---

> > ### Comment · Reviewer_xNaA · 2024-12-02
> > **Thank you for the revision**
> >
> > Thank you for your revision and additions during rebuttal. I still believe that the PPE's direct evaluation on the post-RLHF performance of LLM is very much appreciated. Therefore, I maintain the current score.

---

### Official Review · Reviewer_PDs3 · 2024-10-24

**Soundness:** 3
**Presentation:** 2
**Contribution:** 3
**Rating:** 6
**Confidence:** 3

**Summary:**

The paper tackles the problem of evaluating reward models without having to run the expensive evaluation of downstream LLM performance. The authors compare metrics that are predictive of actual human evaluations on downstream performance, when employed directly at the reward model stage. Different metrics across preference and correctness are compared to down-stream performance, and rew. model accuracy is identified as the best metric predicting downstream human preferences. Correctness is default benchmarks is also a reasonable indicator of LLM performance.

**Strengths:**

Strengths:
- The problem is highly relevant, the evaluation of reward models and RLHF workflows is a challenging problem
- The collection of human preferences and release as open-source is a valuable contribution to the community, the amount of collected human data is impressive (especially including multi-lingual queries)
- The approach of grounding metrics with human data is highly appreciated, and can be a valuable tool for future reward model development
- The selection of models, prompts and correctness metrics is well motivated

**Weaknesses:**

Weaknesses
- Some of the results seem pretty surprising (e.g. the low correlation with reward bench), for me it’s a bit unclear why we observe them. It would be great if the authors could try to reason more about these observations (as outlined below)
- Clarity of writing and presentation could be improved
- "Reward Model Accuracy" as the best metric is not a novel contribution in itself, you may argue that it’s good to have it validated towards human preferences
- Overall, i find the findings a bit weak, i.e. i don't find it easy to draw clear lessons for reward model design (larger vs. smaller models), i think it would have been very helpful to discuss these results more
- I would have liked some more information about variance/stds., e.g. spread of accuracy across sub datasets, etc., best-of-k curves, etc.

Minor
- Some formulations feel a bit clunky, e.g. P6, L.318. “We use the reward models to RLHF Llama-1.1-8B using..”, P8, L.327: “Mean across all metrics is well correlated across all metrics”  I think the paper could profit from one additional pass of copy editing
- I think adding some figures/plots would help readability, e.g. turning table 3 into a chart

I think there are multiple ways to improve the paper, but the main content is generally convincing to me. If the authors can address some of the points for revision, I am ready to raise my score.

**Questions:**

Needs clarification
- Section 4 felt a bit out of place (so the first section), especially “The human preference dataset contains human-labeled preferences for 16,038 pairwise comparisons between 20 selected top models”: How is this dataset used?  In the paper, there are references to multiple datasets and collections, and I find it a bit confusing to follow how many datasets exist, and for which experiment they are used (e.g. 16,038 preferences in section 4, 12,190 votes in section 6, 50.000 votes in sec 6.1.
- I think the structuring of sections 4 and 5 could be improved a bit, in particular I was a bit confused about the content of section 4, I think a figure showing the workflow/process would help to clarify this
- I would like to ask the authors opinion on using already RLHF fine-tuned models as the source of data. Does it introduce its own biases, in particular to the distribution of responses? Doesn’t it implicitly reward similarity to already existing RLHF strategies?
- Can the caption of Figure 2 be improved? I’m not sure what I see in the right plot compared to the left one, how does the message differ?
- The inverse correlation between reward bench and human elo (Figure 4) is interesting, I would really like the authors to provide a hypothesis for this

---

> ### Author Response · Authors · 2024-11-19
> **Official Response to Reviewer [1/4]**
>
> We thank the reviewer for their helpful suggestions. We address comments below:
>
>
> > “Some of the results seem pretty surprising (e.g. the low correlation with reward bench), for me it’s a bit unclear why we observe them. It would be great if the authors could try to reason more about these observations (as outlined below)”
>
> We agree that the discovered discrepancy between RewardBench metrics and post-RLHF preference performance warrants further study. However, our work's focus was not to disprove RewardBench. We can provide the following hypothesis at the requests of the reviewer: we hypothesize that these reward models may have over-optimized for RewardBench's specific preference distribution rather than capturing broader human preferences, potentially exceeding RewardBench's measurement capabilities. However, we note that initial improvements in RewardBench score may still correlate well to real post RLHF human preference outcomes. Ultimately, these insights were only possible through our end-to-end experiments, which enables the research community to further investigate and discuss the true correlations between benchmark metrics and downstream performance. We believe this highlights the value of comprehensive evaluation approaches like ours in understanding real-world model behaviors.
> We will include these explanations in the final version to give readers better context on Figure 4 and its implications.
>
> ***
>
> >“‘Reward Model Accuracy’ as the best metric is not a novel contribution in itself, you may argue that it’s good to have it validated towards human preferences”
>
> We assert that no previous work has explicitly modeled the correlation between reward model behavior and downstream LLM performance. In particular, we find it surprising, and a significant contribution, that correlative measures, such as Spearman correlation, are not predictive of downstream outcomes. Moreover, it was counter intuitive that best of K metrics were overall less predictive than accuracy metrics. We believe there is inherent value in validating in-depth how reward models should behave for optimal downstream performance. We hope the review will consider the novelty of our end-to-end experiment.
>
> >“Overall, i find the findings a bit weak, i.e. i don't find it easy to draw clear lessons for reward model design (larger vs. smaller models), i think it would have been very helpful to discuss these results more”
>
> The main goal of our research is to understand how we should *measure* reward models. Only once we know how to best measure reward models can we then iterate quickly to understand how we should design them. Nevertheless, our research comes with several lessons on how good reward models need to *behave*.
> Reward models should be accurate on an individual response level, while overall biases (eg. favoring OpenAI models over Anthropic models) are not particularly important as long as they are within reason.
> Reward models should show good performance in every subdomain, that is the lower bound performance must be high. This suggests that reward models should be trained to be exceptionally robust. We feel reward model training aimed at robustness may be an interesting future research direction based on our work in this paper. This phenomenon has been observed in theory ([1], [2]) but had not been investigated in empirical studies before our work.
> We hope the reviewer sees the value in these contributions, and recognizes the potential for PPE to provide a foundation for iterating and understanding reward model design moving forward.
>
> ***

---

> > ### Author Response · Authors · 2024-11-19
> > **Official Response to Reviewer [2/4]**
> >
> > >“I would have liked some more information about variance/stds., e.g. spread of accuracy across sub datasets, etc., best-of-k curves, etc.”
> >
> > We thank the reviewer for this suggestion, this analysis will give valuable context on our benchmarks and metrics. We will ensure to provide these tables in the final version. We show tables contains the mean, std, min, quartiles, and max for each sub dataset below. We show 2 examples for the sake of space, but have 12 tables total which we will add to our final paper version.
> >
> > Human Preference Set Accuracy Metric Statistics:
> >
> > | subset           |   mean |    std |    min |    25% |    50% |    75% |    max |
> > |:-----------------|-------:|-------:|-------:|-------:|-------:|-------:|-------:|
> > | overall          | 0.6341 | 0.0337 | 0.5662 | 0.6100 | 0.6301 | 0.6609 | 0.6859 |
> > | hard_prompt      | 0.6351 | 0.0353 | 0.5621 | 0.6115 | 0.6414 | 0.6630 | 0.6946 |
> > | easy_prompt      | 0.6375 | 0.0412 | 0.5554 | 0.6037 | 0.6410 | 0.6698 | 0.7015 |
> > | if_prompt        | 0.6306 | 0.0369 | 0.5571 | 0.6110 | 0.6304 | 0.6609 | 0.6977 |
> > | is_code          | 0.6336 | 0.0316 | 0.5641 | 0.6139 | 0.6418 | 0.6557 | 0.6806 |
> > | math_prompt      | 0.6449 | 0.0483 | 0.5580 | 0.6131 | 0.6388 | 0.6858 | 0.7358 |
> > | similar_response | 0.6287 | 0.0342 | 0.5592 | 0.6059 | 0.6395 | 0.6545 | 0.6815 |
> >
> > MATH correctness set statistics:
> >
> > | metric             |   mean |   std |   min |   25% |   50% |   75% |   max |
> > |:-----------------|-------:|------:|------:|------:|------:|------:|------:|
> > | accuracy         |  0.693 | 0.091 | 0.498 | 0.645 | 0.693 | 0.726 | 0.866 |
> > | area_under_curve |  0.656 | 0.089 | 0.527 | 0.602 | 0.660 | 0.684 | 0.878 |
> > | loss             |  0.199 | 0.080 | 0.047 | 0.169 | 0.189 | 0.214 | 0.401 |
> > | mean_max_score   |  0.504 | 0.091 | 0.348 | 0.470 | 0.500 | 0.527 | 0.741 |
> > | mean_end_score   |  0.486 | 0.107 | 0.245 | 0.459 | 0.494 | 0.516 | 0.736 |
> >
> > ***
> >
> > Minor:
> >
> > >“Some formulations feel a bit clunky, e.g. P6, L.318. “We use the reward models to RLHF Llama-1.1-8B using..”, P8, L.327: “Mean across all metrics is well correlated across all metrics” I think the paper could profit from one additional pass of copy editing”
> >
> > >“I think adding some figures/plots would help readability, e.g. turning table 3 into a chart”
> >
> > We thank the reviewer for pointing out these clunky sentences. We will ensure to clean this up for the final version.
> >
> > We agree that table 3 works better as a chart. We have created a barplot, which the reviewer can view and this anonymized imgur link: https://imgur.com/a/HcOSHLx
> >
> > We also include a graph of human preference accuracies for various models for section 4, which we believe might help readability. We provide an anonymized imgur link here: https://imgur.com/a/1VLXWnQ
> >
> > In addition, we will move table 2 and figure 2 to page 6, which is closer to the relevant text.
> >
> > ***

---

> > > ### Author Response · Authors · 2024-11-19
> > > **Official Response to Reviewer [3/4]**
> > >
> > > Clarifications:
> > >
> > > >“Section 4 felt a bit out of place (so the first section), especially “The human preference dataset contains human-labeled preferences for 16,038 pairwise comparisons between 20 selected top models”: How is this dataset used?  In the paper, there are references to multiple datasets and collections, and I find it a bit confusing to follow how many datasets exist, and for which experiment they are used (e.g. 16,038 preferences in section 4, 12,190 votes in section 6, 50.000 votes in sec 6.1.”
> > >
> > > We apologize for the confusion surrounding Section 4, we will ensure to improve clarity in the section before the final version. We make the following clarifications below:
> > >
> > > 1. The 16,038 human label preferences is the benchmark used to evaluate the reward models on real human preference.
> > > 2. Section 6.1 details the creation of the training set, which samples 7,000 prompts from a set of 50,000, and undergoes deduplication and PII removal as standard practice.
> > > 3. 12,190 downstream human preference labels which were collected live to rank the RLHFed LLMs, and are not part of PPE.
> > >
> > > In summary, there is one human preference dataset used in PPE for the benchmarking of reward models, one dataset of prompts used for training the RLHFed models, and one set of collected downstream human preference labels used to rank the RLFHed models.
> > >
> > > >“I think the structuring of sections 4 and 5 could be improved a bit, in particular I was a bit confused about the content of section 4, I think a figure showing the workflow/process would help to clarify this”
> > >
> > > We thank the reviewer for their feedback on the clarity of these sections. We hope the above clarification of section 4 helps. We will ensure that we clearly explain each dataset’s purpose in the final version. Additionally, we will add the following table to the appendix detailing exactly the subsets of PPE.
> > >
> > > Datasets in PPE:
> > >
> > > | Name                | Num Prompts | Response per Prompt | Preference Type |
> > > |---------------------|-------------|---------------------|-----------------|
> > > | Human Preference V1 | 16,038      | 2                   | Real Human      |
> > > | MMLU Pro            | 512         | 32                  | Correctness     |
> > > | MATH                | 512         | 32                  | Correctness     |
> > > | GPQA                | 512         | 32                  | Correctness     |
> > > | IFEval              | 512         | 32                  | Correctness     |
> > > | MBPP Plus           | 507         | 32                  | Correctness     |
> > >
> > > ***
> > >
> > > >“I would like to ask the authors opinion on using already RLHF fine-tuned models as the source of data. Does it introduce its own biases, in particular to the distribution of responses? Doesn’t it implicitly reward similarity to already existing RLHF strategies?”
> > >
> > > We appreciate the reviewer’s question on this matter. All ground truth preferences in our benchmarks are either sourced from real human preference or objective correctness labels. Therefore, we only reward similarity to these unbiased ground truth preference labels, regardless of response distribution. Additionally, we note that nearly every modern base chat model has some degree of RLHF tuning, therefore a good reward model must be able to correctly label preference when responses are generated from an RLHFed model in order to be relevant for real-world training tasks.
> > >
> > > ***
> > >
> > > >“Can the caption of Figure 2 be improved? I’m not sure what I see in the right plot compared to the left one, how does the message differ?”
> > >
> > > Thank you for the feedback on the figure caption. The left graph shows the best of K curves averaging across all benchmarks. The right shows the best of K curves on a single dataset (MBPP-Plus). We chose the right as an example of over-optimization behavior. We will update the caption to be the following:
> > >
> > > Figure 2: Best of K curves showing reward model score vs K. The blacked dashed line is the theoretical optimal curve, closer to this curve implies a better score. The left graph shows each reward model’s curve averaged across all correctness PPE benchmarks. The right graph shows each reward model’s curve on just the MBPP-Plus set where over-optimization behavior is seen in some reward models, characterized by curves that decrease with respect to increases in K.
> > >
> > > We hope this clarifies the figure for future readers.
> > >
> > > ***

---

> > > > ### Author Response · Authors · 2024-11-19
> > > > **Official Response to Reviewer [4/4]**
> > > >
> > > > >“The inverse correlation between reward bench and human elo (Figure 4) is interesting, I would really like the authors to provide a hypothesis for this”
> > > >
> > > > We agree that the discovered discrepancy between RewardBench metrics and post-RLHF preference performance warrants further study. However, our work's focus was not to disprove RewardBench. We can provide the following hypothesis at the requests of the reviewer: we hypothesize that these reward models may have over-optimized for RewardBench's specific preference distribution rather than capturing broader human preferences, potentially exceeding RewardBench's measurement capabilities. However, we note that initial improvements in RewardBench score may still correlate well to real post RLHF human preference outcomes. Ultimately, these insights were only possible through our end-to-end experiments, which enables the research community to further investigate and discuss the true correlations between benchmark metrics and downstream performance. We believe this highlights the value of comprehensive evaluation approaches like ours in understanding real-world model behaviors.
> > > >
> > > > We will include these explanations in the final version to give readers better context on Figure 4 and its implications.
> > > >
> > > >
> > > > ***
> > > >
> > > > [1] Zhu et al. Principled Reinforcement Learning with Human Feedback from Pairwise or $K$-wise Comparisons, 2024, https://arxiv.org/abs/2301.11270
> > > >
> > > > [2] Li et al. Reinforcement Learning with Human Feedback: Learning Dynamic Choices via Pessimism, 2024, https://arxiv.org/abs/2305.18438
> > > >
> > > > ***
> > > >
> > > > We thank the reviewer for their invaluable feedback, and we sincerely hope that the reviewer would reconsider their rating in light of all our points listed in the rebuttal.

---

> > > > > ### Author Response · Authors · 2024-11-23
> > > > > **Sincere Request for Review of Our Responses**
> > > > >
> > > > > As we approach the end of the discussion period, we want to ensure that our response has adequately addressed all of your concerns given the extensive nature of our materials. Please advise if further clarification is needed or if there are additional questions. We are keen to address any remaining issues and hope you might reconsider your rating based on the information provided.
> > > > >
> > > > > Thank you for your time and consideration.
> > > > >
> > > > > Sincerely,
> > > > >
> > > > > Authors

---

> > > > > > ### Comment · Reviewer_PDs3 · 2024-11-26
> > > > > >
> > > > > > I thank the reviewers for their efforts in answering the raised questions.
> > > > > > I think the proposed changes to the manuscript will improve its quality.
> > > > > >
> > > > > > The hypothesis on reward bench correlations is plausible, and the work opens up further investigation.
> > > > > >
> > > > > > I am looking forward to a revision incorporating the proposed changes (please correct me in case i am missing it, but i do not see a revision at the current point). I am ready to reevaluate my score accordingly. However, i will not change my score at the current time.

---

> ### Author Response · Authors · 2024-11-27
> **Official Comment by Authors**
>
> We thank the reviewer for their comments. We are glad to hear that the reviewer feels the proposed changes will improve quality.
>
> We have since uploaded a revision to paper where proposed changes from the rebuttal have been fully integrated. A detailed list of all changes is in the official comment above.
>
> With regards to changes specific to reviewer PDs3:
> 1. We've added a new figure detailing model accuracies on page 4 (Figure 2).
> 2. Figure 3 (best of k curves) caption was re-written.
> 3. Figure 4 showing model performance is now a bar graph instead of a table for clearer visualization.
> 4. Appendix A1 contains a table of the datasets in PPE (page 15).
> 5. Appendix A.2.1 and A.3.4 contain the various summary statistics for different benchmarks and metrics. (page 25, page 32)
> 6. Appendix A.6.1 (page 34) details the comments and context regarding RewardBench correlations as an extension to Figure 6.
> 7. Minor changes for clarity are spread throughout.
>
> In light of these new revisions, we hope the reviewer considers reevaluating their score.
>
> Once again, we thank the reviewer for their detailed feedback.

---

> > ### Comment · Reviewer_PDs3 · 2024-11-27
> >
> > I thank the authors for making these changes for the revision, i think it helps readability, and i like the added discussion in the appendix. I have increased my score in response.

---

### Official Review · Reviewer_Ub3i · 2024-11-04

**Soundness:** 3
**Presentation:** 3
**Contribution:** 2
**Rating:** 5
**Confidence:** 4

**Summary:**

It proposes Preference Proxy Evaluations (PPE), a novel benchmark designed to evaluate reward models based on their ability to predict post-RLHF outcomes. They use a crowdsourced human preferences and a verifiable correctness dataset for the RM evaluation.

**Strengths:**

- Achieves 77% Pearson correlation with human preference ELO.
- Well motivates why best-of-K scores, row-wise Pearson correlation, & accuracy are relevant for downstream RLHF success.
- Human preference dataset is large and useful in producing statistically significant outcomes (16,038 labeled preference pairs).

**Weaknesses:**

- Underdiscusses the possibility that the RM evals are computationally very expensive to run in the the PPE framework. Best-of-K performance curves and pairwise accuracy have a huge computational burden, especially as K increases. Please provide runtime estimates for different K values or discuss strategies for making the evaluations more computationally efficient.
- Does not discusses the findings (e.g. low quantile aggregation correlation) in any depth. Please explore potential explanations for this correlation or discuss its implications for reward model design and training.

**Questions:**

- Why, intuitively or theoretically, is low quantile aggregation correlation correlatedmore strongly with downstream RLHF performance? Please provide hypotheses or conduct additional analyses to investigate this relationship further.
- How does inference cost scale with K, and what's a feasible value of K for training a large language model today? Please provide a table or graph showing inference costs for different K values, and discuss how these costs might impact real-world applications of the method.
- Why do pairwise accuracy-like metrics suffer from overfitting? Discuss this in more detail and provide empirical evidence of overfitting in pairwise accuracy metrics if possible.
- Is there a predictive accuracy plateau after a certain K-value? Is it possible to include a graph demonstrating diminishing returns/values of predictive accuracy with K?

---

> ### Author Response · Authors · 2024-11-19
> **Official Response to Reviewer [1/2]**
>
> We thank the reviewer for their insightful suggestions and questions. We address this feedback below:
>
> ***
>
> >“Is there a predictive accuracy plateau after a certain K-value? Is it possible to include a graph demonstrating diminishing returns/values of predictive accuracy with K?”
>
> We thank the reviewer for suggesting this additional analysis on K. Note that our paper finds that best of K metrics are overall less predictive than accuracy metrics which are inherently K=2 (Figure 3, left side). As suggested by the reviewer, we create an additional graph showing how as K increases, the predictive power (correlation) decreases slight.. We provide an anonymous imgur link showing the graph here: https://imgur.com/a/EaX9OcX. We will attach this into the appendix along with analysis of its implications. We believe this is an excellent extension of our study which aims to understand which metrics are and are not predictive. That being said, we still release all 32 responses per prompt in PPE such that users can choose to run at whatever K they wish.
>
> We will include recommendations on which metrics to run for maximum efficiency and predictive power in section 7 for the final version. For reference, we copy this new content below:
>
> *“Based on our end-to-end study, we recommend those seeking the most predictive power from PPE run the human preference set as well as the MATH accuracy metric. We suggest that users pay particular attention to the lower bound accuracy across the main human preference set categories (easy, hard, instruction following, coding, math, and similar). Considering our findings, this configuration likely maintains full predictive power of PPE with less than half of the runtime.”*
>
>
> >“Underdiscusses the possibility that the RM evals are computationally very expensive to run in the the PPE framework. Best-of-K performance curves and pairwise accuracy have a huge computational burden, especially as K increases. Please provide runtime estimates for different K values or discuss strategies for making the evaluations more computationally efficient.”
>
> We detail the runtime estimate below for an 8B reward model inference on a single standard Nvidia A100 80GB GPU:
>
> Following the findings of our results section, we find the most predictive power can be achieved by running the human preference set and accuracy metrics on MATH. This runtime is **under 42 minutes** and achieves 77% Pearson correlation with downstream performance.
>
> For comparison, the time required to run the end-to-end validation experiment to observe ground truth reward model rankings would be over a week between model training and collecting enough real human votes.
>
> We document more runtime details below:
>
> Human Preference Set (most correlated to downstream) runtime: 32 minutes
> Each Correctness Preference Set: 1 minute when K=2. Increases in K increase runtime linearly. K=8 would take around 4 minutes, and, if desired, the full K=32 would take about 16 minutes. There are a total of 5 correctness sets.
> Expected runtime if running ALL metrics at full K=32: < 2 hrs
>
> We thank the reviewer for this suggestion and will detail our benchmark runtimes and highlight our improvement from a full end-to-end validation. We construct a table containing this information below the next comment, which we will add to our final paper version.

---

> > ### Author Response · Authors · 2024-11-19
> > **Official Response to Reviewer [2/2]**
> >
> > >“How does inference cost scale with K, and what's a feasible value of K for training a large language model today? Please provide a table or graph showing inference costs for different K values, and discuss how these costs might impact real-world applications of the method.”
> >
> > Reward model inference cost and time scales linearly with K. When training an LLM, a K=16 could be a reasonable number, which is what we chose for our end-to-end training experiments. Ultimately, it is dependent on the model developer preference.
> >
> > When inferencing reward models for PPE, the estimated cost of running an 8B model on an A100 80GB PCIe on full K=32 would be less than \\$3.50. Running the recommended optimized version from above would cost less than \\$1.50.
> >
> > We note that running and testing the full RLHF pipeline by training an LLM with inference K=16 responses score by a reward model using DPO, then deploying the RLHFed LLM to collect human votes would cost around on the order of hundreds of dollars in compute alone. Human preference vote costs could bring the total to be over $1000 per model tested. Therefore, we believe PPE is a significantly cheaper alternative that retains predictive power.
> >
> > Costs are calculated from RunPod’s hourly GPU pricing, which puts an A100 80GB PCIe instance at $1.64 per hour. Costs could fluctuate between GPU providers.
> >
> > We create the following table as a summary of runtime and cost, with the end-to-end numbers for comparison.
> >
> > | Benchmark Set                                   | Time          | Cost           |
> > |-------------------------------------------------|---------------|----------------|
> > | Optimized (Human Preference V1 + Math Accuracy) | < 42 minutes  | < \\$1.50          |
> > | Full Benchmark                                  | < 120 minutes | < \\$3.50          |
> > | End-to-end RLHF pipeline                        | > 1 week      | ~\\$1000 or more |
> >
> > We will make sure to add time and cost estimates to the final version of the paper, including the table above.
> >
> > ***
> >
> > >“Does not discusses the findings (e.g. low quantile aggregation correlation) in any depth. Please explore potential explanations for this correlation or discuss its implications for reward model design and training.”
> >
> > >“Why, intuitively or theoretically, is low quantile aggregation correlation correlatedmore strongly with downstream RLHF performance? Please provide hypotheses or conduct additional analyses to investigate this relationship further.”
> >
> > We posit that the increase in correlation to downstream when using low quantile aggregation across metrics is because this strategy closer measures the robustness of the reward model. Previous theoretical work has suggest that pessmistic measures on reward model perfromance may be useful ([1], [2]). Intuitively, any single weakness within some input domain could be exploited by the policy model during RL training, thus damaging the model. Another reasonable explanation is that a reward model’s weakness in one area may yield noisy signals during training, causing the policy model’s rather fragile parameters to be disrupted— a possibly unrecoverable degradation in what we may consider an instance of “catastrophic forgetting”. Ultimately, the underlying mechanisms are complex; we do not expect to answer this question in its entirety. However, we believe that our end-to-end experiment provides the first step to understanding how reward model behaviors relate to downstream performance.
> >
> > We will add the above more detailed hypothesis to line 505 where these low quantile aggregation results are discussed, as suggested by the reviewer.
> >
> > ***
> >
> > >“Why do pairwise accuracy-like metrics suffer from overfitting? Discuss this in more detail and provide empirical evidence of overfitting in pairwise accuracy metrics if possible”
> >
> > We apologize for the confusion. On line 290 we say “over-fitting” instead of “over-optimization” which may be confusing to readers. We will fix this before the final version. Note, we only suggest that pairwise accuracy metrics may not detect well if a reward model is easily over-optimized, not that the metric itself can be over-optimized. The right graph of Figure 2, shows distinct over-optimization behavior detected by the best of K metrics, marked by the decreasing score as K increases. With this being said, we note that **pairwise accuracy is still the most predictive metric for downstream performance** post-RLHF, as indicated by our findings in Figure 3.
> >
> > ***
> >
> > [1] Zhu et al. Principled Reinforcement Learning with Human Feedback from Pairwise or $K$-wise Comparisons, 2024, https://arxiv.org/abs/2301.11270
> >
> > [2] Li et al. Reinforcement Learning with Human Feedback: Learning Dynamic Choices via Pessimism, 2024, https://arxiv.org/abs/2305.18438
> >
> > ***
> >
> > We thank the reviewer for their invaluable feedback, and we sincerely hope that the reviewer would reconsider their rating in light of all our points listed in the rebuttal.

---

> > > ### Author Response · Authors · 2024-11-23
> > > **Sincere Request for Review of Our Responses**
> > >
> > > As we approach the end of the discussion period, we want to ensure that our response has adequately addressed all of your concerns given the extensive nature of our materials. Please advise if further clarification is needed or if there are additional questions. We are keen to address any remaining issues and hope you might reconsider your rating based on the information provided.
> > >
> > > Thank you for your time and consideration.
> > >
> > > Sincerely,
> > > Authors

---

### Official Review · Reviewer_nK4S · 2024-11-04

**Soundness:** 3
**Presentation:** 2
**Contribution:** 3
**Rating:** 6
**Confidence:** 3

**Summary:**

This paper introduces Preference Proxy Evaluations (PPE), the first reward model benchmark explicitly connected to real-world human preference performance after RLHF. Unlike benchmarks like RewardBench that primarily perform reward evaluations, PPE facilitates investigation into which reward model metrics correlate most closely with RLHF outcomes.

Studying the impact of reward models on post-RLHF model performance is important. This paper involves significant work on this topic. Although I am inclined to accept it, there are still some important shortcomings (see the **Weaknesses**). I hope the authors can address my concerns.

**Strengths:**

- This paper addresses an important topic: the effectiveness of a reward model should ultimately be assessed by the performance of LLMs after RLHF. The study offers valuable insights into the evaluation of existing reward models.
- The paper includes extensive experiments and evaluates models using real-world human preferences.

**Weaknesses:**

- The study employs DPO as the RLHF algorithm to evaluate post-RLHF performance. However, the offline DPO algorithm may face generalization issues. Utilizing an online PPO algorithm to obtain post-RLHF models and assess their performance would offer a more comprehensive evaluation of various reward models. This additional experimentation is essential to support the authors’ core claim of being “the first reward model benchmark explicitly linked to post-RLHF performance.”
- The paper utilizes Llama-3.1-8B-Instruct as the base model for RLHF, which presents two potential concerns: 1) The paper should explore a broader range of base model scales rather than exclusively using the 8B model to fully assess the new benchmark’s validity. 2) Llama-3.1-8B-Instruct itself has undergone DPO RLHF training. I recommend that the authors use a purely SFT-trained Llama-SFT model as the base model.
- The paper would benefit from more detailed descriptions and discussions regarding Figure 4. For example, it should provide intuitive explanations for why metrics from RewardBench fail to reflect post-RLHF preference performance.

**Questions:**

- I am unclear about why the Pearson correlation for “chat hard” in Figure 4 is so low. Does this imply that most RewardBench metrics do not reflect post-RLHF performance?

---

> ### Author Response · Authors · 2024-11-19
> **Official Response to Reviewer [1/3]**
>
> We thank the reviewer for their insightful feedback. We address their comments below:
>
> ***
>
> >“The study employs DPO as the RLHF algorithm to evaluate post-RLHF performance. However, the offline DPO algorithm may face generalization issues. Utilizing an online PPO algorithm to obtain post-RLHF models and assess their performance would offer a more comprehensive evaluation of various reward models. This additional experimentation is essential to support the authors’ core claim of being “the first reward model benchmark explicitly linked to post-RLHF performance.”
>
> We fundamentally agree with the reviewer: PPO would be the preferred algorithm for this experiment, cost and   implementation aside. Unfortunately, the compute requirements of PPO are much more difficult to manage. We would love to explore PPO in future research, but costs and time make it infeasible to do so in this rebuttal period. We ended up choosing DPO for cost, complexity, and stability. A nice feature of DPO is that the experiment is very controlled: the supplied responses are identical, exactly only the reward is different with no randomness from sampling. Ultimately, we still believe our end-to-end experiment is informative and helpful to the research community. We are happy to alter any specific claims the reviewer feels are too strong given the inherent constraints of the RLHF process.
> However, we understand that this direction is fundamentally an important area of exploration. Therefore, we will add additional comments in the limitation section to address this, as well as soften claims made in the results section that may be read as too broad.

---

> > ### Author Response · Authors · 2024-11-19
> > **Official Response to Reviewer [2/3]**
> >
> > >“The paper utilizes Llama-3.1-8B-Instruct as the base model for RLHF, which presents two potential concerns: 1) The paper should explore a broader range of base model scales rather than exclusively using the 8B model to fully assess the new benchmark’s validity. 2) Llama-3.1-8B-Instruct itself has undergone DPO RLHF training. I recommend that the authors use a purely SFT-trained Llama-SFT model as the base model.”
> >
> > For point (1): We agree with the reviewer that a more diverse range of base models would increase our claims of validity— especially across larger model sizes. Unfortunately, the full RLHF verification process is extremely time consuming and expensive. As a university research lab, our financial limitations make it difficult to repeat full RLHF pipelines multiple times. For reference, the full end-to-end experiment was roughly $15,000 (compute, model hosting, human votes), repeating is simply not feasible.
> > This being said, we recognize that this running additional experimentation on more base models would be ideal. We will comment on this in the limitations section. We will alter any claims the reviewer feels are too strong given experimental constraints as well.
> >
> > For point (2): Unfortunately, Meta has not released pure SFT-Llama models within the past year. In order to run an experiment that is representative of a real RLHF procedure, we found it necessary to use these newer, widely used models. We have also observed that RLHFing a model that has already undergone some form of RLHF is quite common, see Starling-7B [1], gemma-2-9b-it-SimPO [2], Llama-3.1-Nemotron-70B-Instruct [3].
> > We concur with the reviewer that an ideal RLHF experiment would include training on all possible types of base models, RLHF-ed or not. However, DPOing an already RLHFed base model means the reward model needs to continue to improve the policy model despite the policy model already optimizing for some previous human preference signals. We believe this makes the training procedure slightly closer to an online procedure like PPO, where the reward model must continuously improve the policy model as it becomes more and more RLHFed. Considering over half the reward models tested were able to prove the human preference performance of the base model, we believe further RLHFing this base model was a task of reasonable difficulty and sufficient for the purpose of comparing reward models.
> >
> > Based on the reviewers above two comments, we augment the limitations section 8.2 to be as follows:
> >
> > *“Unfortunately, end-to-end evaluation of reward models via post-RLHF LLM performance on human preference is extremely expensive and time-consuming. As such, we are limited to testing the performance of nine select models, rather than all reward models. In addition, we use DPO, an offline RL algorithm over PPO, an online algorithm, which may play more into over-optimization issues or may have different reward model requirements altogether. We encourage future work to study downstream outcomes under online RL algorithms. Moreover, we note that resource constraints necessitated experimenting with just Llama-3.1-8B-Instruct as the base policy model. We believe additional exploration on a diverse set of base models, especially larger base models or pure SFT-ed based models, may yield additional novel insights. With these considerations, we note that the downstream performance measured in our work is in the context of the base model and RLHF learning algorithm used, and is not a unilateral measurement of downstream outcomes in all possible configurations. Future work should experimentally verify the desired reward model behavior of other RLHF configurations.”*

---

> > > ### Author Response · Authors · 2024-11-19
> > > **Official Response to Reviewer [3/3]**
> > >
> > > >“The paper would benefit from more detailed descriptions and discussions regarding Figure 4. For example, it should provide intuitive explanations for why metrics from RewardBench fail to reflect post-RLHF preference performance.”
> > >
> > > We agree that the discovered discrepancy between RewardBench metrics and post-RLHF preference performance warrants further study. However, our work's focus was not to disprove RewardBench. We can provide the following hypothesis at the requests of the reviewer: we hypothesize that these reward models may have over-optimized for RewardBench's specific preference distribution rather than capturing broader human preferences, potentially exceeding RewardBench's measurement capabilities. However, we note that initial improvements in RewardBench score may still correlate well to real post RLHF human preference outcomes. Ultimately, these insights were only possible through our end-to-end experiments, which enables the research community to further investigate and discuss the true correlations between benchmark metrics and downstream performance. We believe this highlights the value of comprehensive evaluation approaches like ours in understanding real-world model behaviors.
> > >
> > > We will include these explanations in the final version to give readers better context on Figure 4 and its implications.
> > >
> > > >“I am unclear about why the Pearson correlation for “chat hard” in Figure 4 is so low. Does this imply that most RewardBench metrics do not reflect post-RLHF performance?”
> > >
> > > We thank the reviewer for bringing up this question. While our end-to-end experiments suggest that the “Chat-Hard” category has lower correlation in particular, claiming RewardBench metrics does not reflect post-RLHF performance would require a dedicated study in that area. One possible explanation for the lower correlation in “Chat-Hard” is that it contains adversarial preference pairs from the LLMBar dataset generated through a synthetic process, which may not be representative of real-world human preference. For instance, we noticed roughly 72% of the rejected responses in “Chat Hard” are the longer response, meaning this category encourages very anti-verbose preferences, which could favor reward models overfitted toward this direction. Ultimately, these are speculative observations. We believe end-to-end experiments like the one in our paper are the first step to understanding underlying behavior.
> > >
> > > ***
> > >
> > > [1] Zhu et al. Starling-7B: Improving Helpfulness and Harmlessness with RLAIF, COLM 2024, https://openreview.net/forum?id=GqDntYTTbk
> > >
> > > [2] Meng et al. SimPO: Simple Preference Optimization with a Reference-Free Reward, 2024, https://arxiv.org/abs/2405.14734
> > >
> > > [3] Wang et al. HelpSteer2-Preference: Complementing Ratings with Preferences, https://arxiv.org/abs/2410.01257
> > >
> > > ***
> > >
> > > We thank the reviewer for their invaluable feedback, and we sincerely hope that the reviewer would reconsider their rating in light of all our points listed in the rebuttal.

---

> > > > ### Author Response · Authors · 2024-11-23
> > > > **Sincere Request for Review of Our Responses**
> > > >
> > > > As we approach the end of the discussion period, we want to ensure that our response has adequately addressed all of your concerns given the extensive nature of our materials. Please advise if further clarification is needed or if there are additional questions. We are keen to address any remaining issues and hope you might reconsider your rating based on the information provided.
> > > >
> > > > Thank you for your time and consideration.
> > > >
> > > > Sincerely,
> > > > Authors

---

> ### Comment · Reviewer_nK4S · 2024-11-29
>
> Thanks for the detailed responses.
>
> If possible in the future, I suggest that the authors include experiments with various types and scales of base models, along with PPO experiments, to better support the benchmarks and arguments presented in this paper. I am inclined to maintain my current score.

---

### Author Response · Authors · 2024-11-27
**Official Comment by the Authors**

We thank the reviewers for the valuable feedback. We appreciate that the reviewers see the value in our work to evaluate reward models with respect to their downstream LLM performance (nK4S, xNaA) as well as our usage and release of real human preferences (nK4S, Ub3i, PDs3).

In our revised version, changes as suggested by the reviewers are shown in blue text. Below we detail revisions made to the paper.


### Major Additions

The major additions are as follows, in rough order of appearance:

1. Figure 2 showing accuracies of various preference models has been added, as suggested by reviewer PDs3.
2. Figure 3 caption is re-written as suggested by reviewer PDs3.
3. Figure 4 showing model performance as a graph instead of a table has been added, as suggested by reviewer PDs3.
4. The last paragraph on lines 519-525 has been slightly expanded upon, with links to deeper additional comments in the appendix, as suggested by reviewer Ub3i.
5. Limitations section 8.2 has been updated based on reviewer nK4S's comments and suggestions.
6. Appendix A1 has been added, which shows an overview of PPE's various datasets, as suggested by reviewer PDs3. Additionally, it is referenced at the end of section 3 on line 136.
7. Appendix A.2.1 and A.3.4 have been added. These section contain 12 new tables detailing distribution statistics for each benchmark/metric/domain in PPE, as suggested by reviewer PDs3
8. Lines 1778-1785 have been added to Appendix A.6, laying out deeper possible explanations for why low quantile aggregation is helpful, as suggested by reviewer Ub3i.
9. As suggested by reviewers nK4S, Ub3i, and PDs3, Appendix section A.6.1 has been added, which comments on the low correlation seen in RewardBench. This section is referenced in the original figure showing RewardBench correlations in section 7.
10. As suggested by reviewer Ub3i, Appendix A.6.3 and Figure 13 has been added, showing predictive power vs K, along with lines 1918-1921 providing additional explanation.
11. Based on reviewer Ub3i's suggestions, Appendix A.7 has been added, detailed recommended settings to run PPE based on the end-to-end experiment and Appendix A.6.3.
12. Appendix A.8 has been added, which contains a table detailing PPE's efficient runtimes and costs, as suggested by reviewer Ub3i's.

### Other Changes

Other minor changes based on reviewer suggestions are written in blue.

### Reorganization

Additionally, in order to integrate reviewer feedback while adhering to the 10 page limit, we reorganized some content based on importance:
Details on best of K scoring and metrics has been moved to Appendix A.3.2. The Abstract and Introduction have been modified to be slightly shorter and clearer; the underlying semantics of these sections are unaltered. (Note these structural changes are not highlighted in blue.) The main figure has been slightly updated showing PPE's role in the RLHF pipeline for increased clarity.

Once again, we thank the reviewers for their questions, feedback, and suggestions.

---

### Meta-Review · Area_Chair_hpwL · 2024-12-21

**Metareview:**

This paper addresses an important problem: which reward model to use for RLHF? Running the full pipeline is expensive and take days. This paper addresses it proposing a new large benchmark for reward model along with how to use it to get a score that correlates well with downstream RLHF performance.

Strengths:
1. Addresses a very important practical problem facing any team that does RLHF
2. Reviewers found the new benchmark useful
3. Achieves a decent 77% Pearson correlation with downstream performance

Weakness:
1. Reviewers found that the paper is lacking in explanations. E.g., about why low quantile aggregation correlation used by reviewer Ub3i.
2. Also, due to budget reasons, authors were not able to train on more LLMs. Nevertheless, this limits how much we can read from the results. Someone else will have to run these results more broadly before we can trust the generality.

Overall, I like this paper. I think this new benchmark will be useful. More investigation is certainly needed but I think this paper will have an impact even in its current form so I recommend acceptance.

**Additional Comments On Reviewer Discussion:**

Reviewers pointed certain concerns:

1. How expensive it is to run the benchmark by reviewer Ub3i. Authors presented number that shows that it is much cheaper to run the metrics than the full RLHF pipeline.

2. Lacking explanation for important observations. E.g., why is low quantile aggregation correlation better (reviewer Ub3i) and why there is low correlation with reward bench (reviewer PDs3). Author have added explanation that this measures robustness of the reward model and that we could be over-optimizing on reward bench. These seem reasonable.

3. Lack of experiments on more LLMs. Authors mentioned that they have budgetary constraints. This is understandable but does take away from the generality of the study.

Additionally, authors have also made other changes to improve the presentation.

Overall my main concern is generality but I think the benchmark will be used by the community given the importance of this problem.

---

### Decision · Program_Chairs · 2025-01-22

Accept (Poster)